# Ultraviolet Photodetectors: From Photocathodes to Low-Dimensional Solids

**DOI:** 10.3390/s23094452

**Published:** 2023-05-02

**Authors:** Antoni Rogalski, Zbigniew Bielecki, Janusz Mikołajczyk, Jacek Wojtas

**Affiliations:** 1Institute of Applied Physics, Military University of Technology, 2 Kaliskiego Str., 00-908 Warsaw, Poland; 2Institute of Optoelectronics, Military University of Technology, 2 Kaliskiego Str., 00-908 Warsaw, Poland

**Keywords:** ultraviolet photodetectors, photoemissive UV photodetectors, silicon-based UV detectors, wide bandgap UV photodetectors, 2D nanostructures

## Abstract

The paper presents the long-term evolution and recent development of ultraviolet photodetectors. First, the general theory of ultraviolet (UV) photodetectors is briefly described. Then the different types of detectors are presented, starting with the older photoemission detectors through photomultipliers and image intensifiers. More attention is paid to silicon and different types of wide band gap semiconductor photodetectors such as AlGaN, SiC-based, and diamond detectors. Additionally, Ga_2_O_3_ is considered a promising material for solar-blind photodetectors due to its excellent electrical properties and a large bandgap energy. The last part of the paper deals with new UV photodetector concepts inspired by new device architectures based on low-dimensional solid materials. It is shown that the evolution of the architecture has shifted device performance toward higher sensitivity, higher frequency response, lower noise, and higher gain-bandwidth products.

## 1. Introduction

Ultraviolet (UV) radiation is an important component of solar radiation (Figure 1a), which has a significant impact on the development and survival of mankind. Extreme UV radiation can cause various diseases, including cataracts and skin cancer, and accelerate the aging process. This radiation also has a significant impact on crop yields and the vitality of infrastructure. Recent studies have shown that a 1% reduction in the thickness of the ozone layer will increase UV radiation near the earth’s surface by 2%, resulting in a 3% increase in melanoma incidence [1,2]. Accordingly, the search for UV photodetection has attracted the interest of scientists in related fields. The International Commission on Non-Ionizing Radiation Protection regulates UV radiation limits for humans, as shown in Figure 1b.

Photodetectors are one of the popular types of technology used in ultraviolet radiation research. They are widely used in the industrial area (flame detectors, fire alarm systems, extreme UV lithography), national security (missile defense, military recognition, explosives detection, forensic analysis, secure communications), in fields such as medicine (UV imaging, protein analysis, and DNA sequencing) or biology (biological agent detection), and when dealing with environmental issues (ozone detection, air pollution determination, disinfection, and decontamination).

Ultraviolet radiation is electromagnetic radiation from 10 nm to 400 nm (Figure 1c). It is shorter than radiation in the visible spectrum but longer than X-rays. The ultraviolet spectrum is also subdivided due to each band’s effect on the biosphere [3]:UV-A—wavelengths: 315–400 nm;UV-B—wavelengths: 280–315 nm;UV-C—wavelengths: 100–280 nm.

The sun emits the entire range of UV radiation; therefore, the radiation of the UV-A range and part of the UV-B radiation range reaches the Earth’s surface (Figure 2). The ozone (triatomic oxygen) layer absorbs most of the UV-B, while the diatomic oxygen in the atmosphere absorbs all radiation below 200 nm [4].

UV radiation detection systems use photographic films (nowadays very rarely), thermal, and photon detectors. Historically, the first image-capturing method is one of the oldest and most efficient in the UV range for wavelengths above 200 nm, where gelatine shows noticeable absorption [5]. A single exposure allows for the capture of large amounts of data on photographic films. However, they have serious disadvantages: their sensitivity is lower than that of photon detectors, their responsivity has a non-linear dependence on the incident photon flux at a given wavelength, and their spectral response is broad. Although photographic films are historically significant, we will not deal with them.

On the other hand, the sensitivity of thermal detectors is much lower than that of photon detectors, so in this chapter, we will focus on the analysis of photon detectors. They are absolute radiometric standards in the ultraviolet range of the spectrum. Photon detectors based on external or internal photoelectric phenomena overcome the abovementioned limitations and offer superior performance. The main objective is to develop surveillance imaging systems for military and civilian applications with the following criteria:Not light-sensitive (solar blind);High quantum efficiency;High dynamic range of operation;Low background and floor noise often dominate observations of the weak UV.

Figure 3 shows the classification of ultraviolet detectors.

There are two main groups of photon detectors. The first includes those with the external photoelectric phenomenon, which means that photons excite the photocathode to generate photoelectrons, and the external anode registers these electrons. The procedure is used inside the vacuum tube photodetectors, particularly photomultipliers and amplifiers (microchannel plates—MCP).

The second group includes detectors based on the internal photoelectric phenomenon, in which absorbed electromagnetic radiation releases electrically charged particles (electrons and holes) within a material. In this group are included solid-state devices based on silicon (including CCD and CMOS) or detectors made of wide bandgap semiconductors such as AlGaN and SiC. In photovoltaic detectors, the electrical field (p–n junctions, Schottky barriers, or metal-insulator-semiconductor structures) separates the electron-hole pairs, generating a photocurrent. The photocurrent level increases proportionally to the intensity of the incident radiation.

The photodiodes become a more significant type of detector over time. They have many advantages, such as low power dissipation, inherently high impedance, and negligible 1/*f* noise. Due to the fact that they can be easily multiplexed using the ROIC, the number of pixels in two-dimensional (2D) arrays is limited by existing technologies. Due to the higher doping levels in the absorber of the photodiode and the fast collection of the carriers, the photodiode output signal remains linear for substantially higher photon flux compared to photoconductors.

Table 1 lists different types of photon detectors with their brief characterization. In this paper, we describe the state of the art of UV range detector technology, including standard UV detectors dominating the global market and new conceptual designs exploring new materials generally referred to as low-dimensional solids.

## 2. General Theory of Photodetectors

Photon detectors exhibit a selective sensitivity depending on the incident radiation wavelength related to their unit power (Figure 4a). This is because, ideally, each photon with an energy higher than the bandgap of the semiconductor generates an electron-hole pair (e-h pairs). For a constant radiation power, the number of photons reaching the detector is directly proportional to the wavelength:(1)$$N_{photons}=\frac{\Phi }{E}=\frac{\Phi \lambda }{hc},$$
where *Φ* is the photon flux, *E* is the photon energy, *h* is the Planck constant, *c* is the light speed, and *λ* is the wavelength.

Thus, the current responsivity is also directly proportional to the wavelength:(2)$$R_{i}\left(\lambda \right)=\frac{I_{ph}}{\Phi }=\frac{q\lambda }{hc},$$
where *q* is the electron charge.

Optical radiation irradiated onto the active surface of the photodetector generates a photocurrent. For a small radiation signal, this current proportionally increases with the radiation power:(3)$$I_{ph}=q\eta A\Phi g,$$
where *η* is the quantum efficiency and *g* is the photoelectric current gain.

The quantum efficiency *η* of a detector is defined by the number of e-h pairs generated by the photodetector per incident photon. Its standardized value is generally less than unity, expressed in percentage. Some absorbed photons may not be registered based on the collected free e-h pairs because some carriers recombine or become immediately trapped. If the radiation penetration depth 1/*α* (where *α* is the absorption coefficient) is comparable with the thickness of the semiconductor layer, all photons will not be absorbed. The reflectance of the detector surface and device structure also influences quantum efficiency. That is why reducing the reflections on the semiconductor surface, increasing absorption, and preventing carrier recombination or trapping increase the photodetector’s quantum efficiency.

The number of carriers passing through the contacts per generated pair determines the photoconductive gain. The gain quantitively describes the photodetector’s current response. For example, the photoelectric current gain equals unity for a typical photovoltaic detector.

Finally, the spectral current responsivity can be expressed by:(4)$$R_{i}\left(\lambda \right)=\frac{\eta q\lambda }{hc}g.$$

The output signal of infrared detectors can be presented as a function of wavelength [6]. In Figure 4a, photon detectors are characterized by wavelength-selective response dependence, and the signal is proportional to the photon arrival rate. Since energy per photon is inversely proportional to the wavelength, the spectral response increases linearly with the wavelength. This growth is possible at the so-called cut-off wavelength, which is determined by the material. It is usually defined when the detector’s response is reduced by 50% of the peak value.

Thermal detectors have a constant responsivity value for analyzing radiation power. However, their photon flux response decreases with increasing wavelength (a decrease in absorbed energy) (Figure 4b). Photon detectors have a higher responsivity and response rate than thermal detectors.

Assuming the same gain values for photocurrent and noise current, the noise current caused by statistical processes of carrier generation and recombination equals [6]:(5)$$I_{n}^{2}=2(G+R)A_{e}t\Delta fq^{2}g^{2},$$
where *G* and *R* are generation and recombination rates, respectively; Δ*f* is the frequency band; and *t* is the detector thickness.

Detectivity *D** is the most important parameter of the detector, and it is usually used to determine the detector’s signal-to-noise ratio. Its value is normalized to the detector area and the noise bandwidth and is described by:(6)$$D^{*}=\frac{{\left(A_{d}\Delta f\right)}^{1/2}}{V_{n}}R_{v}=\frac{{\left(A_{d}\Delta f\right)}^{1/2}}{I_{n}}R_{i}=\frac{{\left(A_{d}\Delta f\right)}^{1/2}}{\Phi_{e}}\left(SNR\right).$$

Therefore, according to Equations (4) and (5), we reach the following:(7)$$D^{*}=\frac{\lambda }{2^{1/2}hc}{\left(\frac{A_{o}}{A_{e}}\right)}^{1/2}\frac{\eta }{t^{1/2}}\frac{1}{{\left(G+R\right)}^{1/2}}.$$

For a given wavelength and operating temperature, the best detector performance has a maximum value of *η*/[*t*(*G* + *R*)]^1/2^ term, corresponding to the highest quantum efficiency in the thinnest detector. In addition, the total number of generation and recombination acts per time unit [equal to (*G* + *R*) (*A_e_t*)] should be as small as possible. At equilibrium, the generation and recombination rates are equal (*G* = *R*), and for *A_o_* = *A_e,_* we obtain:(8)$$D^{*}=\frac{\lambda }{2hc}\frac{\eta }{t^{1/2}}\frac{1}{{\left(G\right)}^{1/2}}.$$

The noise performance of the detector can also be described by a noise equivalent power (*NEP*). *NEP* corresponds to the incident power on the detector that gives a signal-to-noise ratio (*SNR*) of unity. In terms of responsivity, *NEP* equals:(9)$$NEP=\frac{I_{n}}{R_{i}}=\frac{V_{n}}{R_{v}},$$
where *I_n_* and *V_n_* are the current and voltage noises.

Since the noise voltage root mean square (RMS) value is proportional to the square root of the bandwidth, the *NEP* uses a spectral density with a unit of W/Hz^1/2^. The lowest values of *NEP* (*NEP_min_*) are obtained at the wavelength *λ* with the maximum detector responsivity *R_max(_*(*λ*). Thus:(10)$${NEP}_{\min}=NEP\left(\lambda \right)\frac{R}{R_{max}}.$$

The minimum detectable power *P_min_* can be calculated using the following equation:(11)$$P_{min}=NEP\left(\lambda \right){∆f}^{1/2},$$
where Δ*f* is the measurement bandwidth.

The best performance of the photodetector can be achieved when its noise is lower than photon noise. This type of noise is essential because of the discrete nature of the optical radiation. The radiation that illuminates the detector includes the radiation of the object and the background. The limits of most photodetectors’ operation can be practically described by the signal fluctuation limit (SFL) and the background fluctuation limit, also known as the background limited infrared photodetector (BLIP).

The values of *NEP* and *D** in the SFL operation equal [7,8]:(12)$${NEP}_{SFL}=\frac{2^{3/2}hc∆f}{\eta \lambda },$$
(13)$$D_{SFL}^{*}=\frac{\eta \lambda }{2^{3/2}hc}\sqrt{\frac{A}{\Delta f}}.$$

For the application of infrared detectors, the signal radiation is usually lower than the thermal background. If the thermal generation is reduced much below the background level, the background radiation determines the device’s performance. Thus, the mean square value of the noise current is:(14)$$I_{n}^{2}=2q^{2}g^{2}\eta A\Phi_{B}∆f,$$
where $\Phi_{B}$ is the flux density of the background photon reaching the detector determined by its field of view (FOV) and the temperature of the background.

Using Equations (4), (6) and (14), the *NEP* is given by [9]:(15)$${NEP}_{BLIP}=\frac{hc}{\lambda }{\left(\frac{A\Phi_{B}\Delta f}{\eta }\right)}^{1/2}.$$

The expression for the BLIP detectivity limited by shot noise is equal to [8,10,11,12]:(16)$$D_{BLIP}^{*}=\frac{\lambda }{hc}{\left(\frac{\eta }{\Phi_{B}}\right)}^{1/2}.$$

The exemplary graph of the spectral detectivity for a background temperature of 290 K and a 2π steradian detector FOV is presented in Figure 5 [6,13]. It can be seen that the SFL curve crosses the BLIP line near 1.2 μm. From this, it follows that SFL dominates at wavelengths below 1.2 μm, and that the wavelength dependence of detectivity is weak. However, above 1.2 μm, the influence of background radiation is more significant, and detectivity strongly depends on the wavelength.

The influence of background noise on UV detector performance is related to the solar spectral irradiance and a declining tail of the detector’s responsivity characteristics. The solar background noise for solar-blind UV detection is as important as BLIP for infrared photodetectors. The detectivity of background-limited UV photodetectors (BLUP) describes maximum values in the presence of solar irradiance background [13]. Figure 6 shows an example of the spectral performance of the UV photodetectors with various cut-off wavelengths, where the SFL curve presents their ultimate performance and *N_Sun_* denotes the total rate of incident solar photons. The so-called lowpass background limit (LBL) performance is calculated for an ideal solar-blind UV detector assuming negligible dark current and an ideal sawtooth-like output signal curve (Figure 4a).

For photons below 285 nm, *N_Sun_* is very low because of the ozone absorption in the atmosphere. An exponential decrease in BLUP and LBL detectivities from 9.5 × 10^17^ cmHz^1/2^W^−1^ at 285 nm to ca. 10^12^ cmHz^1/2^W^−1^ at 300 nm can be noticed due to the increase in solar background noise. They approach a plateau above 300 nm, corresponding to solar spectral irradiance. Moreover, the LBL detectivity becomes lower than the BLUP, which can explain the accumulation of solar irradiance leakage in the range of low-pass responsivity. What is more, AlGaN photodiodes have the highest detectivity (close to the SFL) at 260 nm [13], but filters reducing solar-irradiance leakage are required to obtain such high values. The papers signaling the highest detectivities (data highlighted in pink) do not indicate that optical filters were used in the detectors whose characteristics were measured. In addition, some of these papers reported detectivity values exceeding the physical limit of the SFL. On this basis, it can be thought that these published parameters of UV detectors are unreliable or overestimated.

It is believed that these overestimations of detector parameters are a consequence of the:Inadequacy of detectivity specifications based on responsivity and noise (shot noise and generation-recombination noise);Lack of properly refined protocols for accurately determining figure-of-merit for photodetectors (especially for the new-generation of two-dimensional (2D) and quasi-2D material ones).

The correct formulas for shot noise and g-r nose include internal gain, *g*:(17)$$I_{sh}=\sqrt{2qI_{d}g∆f},$$
(18)$$I_{gr}=\frac{4qI_{d}g∆f}{1+\omega^{2}\tau^{2}},$$
where *I_d_* is the dark current and *τ* is the carrier lifetime. The g-r noise is frequency-related, *ω*.

In the literature, one encounters the use of an incorrect expression for shot noise ($I_{sh}=\sqrt{2qI_{d}∆f}$) which leads to a fictitious improvement in SNR by a factor of $\sqrt{g}$. It follows that the error in shot noise estimation increases with higher gain and is particularly significant for photodetectors with very high internal gain.

The highest *D**-values, including unrealistic ones (exceeding SFL limits), are marked for Ga_2_O_3_ FET phototransistors and a new generation of photodetectors with active areas containing low-dimensional solids, as shown in Figure 5.

The last figure of Table 1 shows a diagram of how a phototransistor works. Its operation is similar to that of a photoconductor. In a photoconductor, the signal is caused by the generation of an electron-hole pair when one type of carrier is trapped by localized states (nanoparticles and defects). The photoconductive gain can be easily determined by the ratio of the lifetime of the free carriers to their transit time between the electrodes of the detector. If the drift length of the carriers is greater than the inter-electrode distance, the free charge swept from one electrode is immediately replaced by the injection of an equivalent free charge on the opposite electrode to maintain charge neutrality in the detector. In this way, the free charge will circulate in the detector circuit until recombination causes signal amplification—it is the so-called photoelectric gain. However, in the case of a phototransistor, the active region of the detector is separated from the substrate by an insulator, which allows the application of a gate voltage, *V_G_*, to tune carrier transport in the active region. The active region is more susceptible to the local electric field than conventional bulk materials, and then the photogeneration effect can strongly modulate the conductivity of the channel by the external gate voltage, *V_G_*. Under such conditions, much higher optical gain can be achieved.

## 3. Photoemissive Detectors

Photoemissive detectors are detectors based on the external photoelectric phenomenon. It involves the emission of an electron from a material (a photocathode) due to its excitation by an incident photon. The process of photoemission from any material consists of the excitation of a photoelectron, its diffusion to the emitting surface, and its escape into the surrounding vacuum (Figure 7).

Ionization energy binds electrons to the lattice in any material. All electrons excited to an energy higher than the ionization energy have a significant probability of escaping. As for metal photocathodes, whose reflectivity is high in the visible range (90–99%) and near-infrared region, the energy loss of excited carriers is fast, but quantum efficiency is low. Semiconductor photocathodes have better quantum efficiency due to higher absorption efficiency and a longer relaxation time for any energy losses. Their lower ionization potential also provides broader spectral operation.

Electron scattering is a dominant process in metals. Therefore, carriers’ average free path is short, and only electrons created close to the surface (within a few atomic layers) can escape. This loss mechanism is negligible for semiconductors, and photoelectrons lose energy due to electron-hole pair production (impact ionization) and lattice scattering (phonon creation). Minor energy loss per scattering (~0.01 eV) enables an average free path of 30–40 Å. In the depths of several hundred Å, adequate energy for photoemission is feasible in the absence of electron pair generation. Although electron energy is higher than the threshold limit (*E_th_*), it can be lost by creating an electron-hole pair. Generally, *E_th_* is several times higher than the energy gap *E_g_*.

Lastly, the created photoelectron needs sufficient energy to surpass the surface potential when it approaches the interface. For metals, it is a work function described by the difference between the Fermi level energy and the minimum energy of a free electron in a vacuum. Metal photocathodes have relatively high work functions, varying most in the 4 to 5 eV range. This range corresponds to wavelengths from 311 to 249 nm. As for the semiconductors, doping modifies the position of the Fermi level. The electron affinity *E_A_* is a better factor for describing the energy gap between the vacuum level and the bottom of the conduction band.

Since the first demonstration of the principle of photoemission in 1887 by Hertz and after Einstein’s clarification of the effect in 1905, many distinct materials and their spectral response characteristics as well as threshold wavelengths have been studied. A significant motivator for the use of the photoemission effect in practical systems was the discovery of the Ag-O-Cs photocathode by Koller (1929) [14] and Campbell (1931) [15]. It provided two orders of magnitude higher quantum efficiency than anything previously investigated. Each photocathode can be characterized using the so-called S number, defined by the Electronic Industries Association (1954). It refers to the total spectral response considering the input window’s impact but does not identify specific types of cathodes, their materials, or absolute responsivities. Typical spectral characteristics of various photocathodes are presented in Figure 8 [16]. Numerous types of photocathodes enable the detection of UV radiation, but only those made with CsTe are “blind” to sunlight. In others, optical filters blocking solar radiation can be used.

A new generation of different types of photocathodes has recently been developed. They are characterized by low electron affinity, high chemical stability, and a direct bandgap, making ternary AlGaN alloys superb materials for photocathodes. The planar p-type AlGaN photocathodes are characterized by a 70–80% quantum efficiency at a wavelength of 122 nm. Although it is reduced by approx. At 10–20% at 360 nm, a sizable improvement compared to typical CsI and CsTe photocathodes is observed (Figure 9) [17].

Photocathodes determine the spectral sensitivity of photomultiplier tubes or photoemission array detectors, such as microchannel plates. The work function of the photoemission material determines the maximum possible optical wavelength at which the external photoelectric effect can occur. Detailed information on reflection and transmission mode photocathodes and their spectral characteristics is widely available in the study/scientific literature [18].

The operation of the photomultiplier tube (PMT) is schematically shown in Table 1. It generally contains between 10 and 15 dynodes, and a voltage of about 100 V is maintained between successive dynode plates. The result is an amplification of the generated electric current by a factor as high as 10^9^. To meet market demands, different PMTs have been developed. Hamamatsu has developed hundreds of different photomultiplier tubes (Figure 10a,b), and their representative selection is shown in Figure 10c [19,20].

A schematic of a microchannel plate (MCP) is shown in Figure 11. The primary photoelectrons produced by the photocathodes are multiplied by secondary emission. As a result, large “clouds” of electrons are released, which, on reaching the anodes, generate a current that conventional electronics can easily detect. The resulting electron clouds can also be accelerated to high energy, which produces visible radiation when hitting the phosphor screen. The multiplied visible photons irradiate the CCD array through a fiber-optic beam.

MCPs ensure high responsivity and radiometric stability, as well as high-resolution imaging ability. Nevertheless, performance improvements are still being made, especially in terms of spatial resolution and detector size, dynamic range, and quantum efficiency, along with a reduction in background noise, power consumption, dimensions, and mass.

The next MCP generation’s development was possible thanks to photolithographic silicon-based microchannel plates, and large-format (scaled to 6″) position sensing readouts ensured high-resolution imaging. Silicon-based MCPs have become preferable to glass MCPs due to their low intrinsic background and low fixed pattern noise.

The instruments designed for space applications were faced with new challenges related to implementing this kind of detector [21,22]. For instance, the Cosmic Origins Spectrograph (COS), which provided spatial resolution of ~25 µm and was mounted on the Hubble Space Telescope (HST) in May 2009, had two segments of far-ultraviolet detectors, the active area of which was 85 × 10 mm (Figure 12a). It is important to mention that astounding progress has been made in decreasing the solid noise created in the “dead zones” of the MCP fiber beam boundaries (Figure 12b).

Figure 13 presents the working principle of photon-counting UV-MCP detectors for space missions, sensitive in the wavelength range from far UV to near UV. These detectors have a Cs-activated GaN photocathode, operating in semi-transparent mode on (001)-MgF_2_. The UV-MCP detector comprises a stack of two MCPs and a coplanar cross-strip anode with an advanced electronic readout.

The real-scale detector body has a diameter of 7.9 cm. The numbers in Figure 13 indicate the stages of the detector operation:Incoming UV radiation passes through a window (fused silica/MgF_2_/LiF);The photocathode converts the radiation into photoelectrons; the degree of conversion depends on the photocathode material and the wavelength of the incident radiation (estimated to reach as high as 70% at 230 nm for GaN);A high voltage of the order of 2 kV accelerates the incident photoelectrons in microchannel plates (MCPs), resulting in a charge of, e.g., 10^5^ to 10^6^ electrons on the bottom side of the MCP stack;The electron cloud is accelerated towards the anode with a cross stripe (64 strips in the X direction and 64 strips in the Y direction).

Photomultiplier tubes are relatively expensive, bulky, sensitive to magnetic fields, and require high voltage and power. They have been used for a long time for UV detection and are still commonly used in laboratories. However, the practical need to construct portable UV detection systems requires the development of miniaturized semiconductor-based UV photodetectors. This scenario mainly uses photoconductors, Schottky junction photodiodes, p-i-n photodiodes, or metal-semiconductor-metal photodiodes.

## 4. Silicon-Based Photodetectors

Semiconductors such as silicon and some III-V compounds (e.g., GaP, GaAsP) were first considered to realize UV detection, although expensive optical filters are required in such detectors to tune the photoreceiver to the appropriate spectral range. However, their use results in a significant attenuation of the signal reaching the detection structure. Therefore, wide bandgap semiconductors such as silicon carbide (SiC), diamond and gallium nitride (GaN), or AlGaN alloys are used instead. Table 2 provides the physical properties of important semiconductors used to fabricate UV photodetectors. Silicon, SiC-based materials, and semiconductors such as GaAs, GaAsP, and GaP materials are mainly used in commercially available UV photodetector constructions.

In this section, we will first focus briefly on Si, and in the next sections, on some III-nitride-based UV photodetectors, SiC-based UV photodetectors, and diamond.

Silicon photodiodes are widely used for detection in a spectral range below 1.1 µm, including X-ray and gamma-ray. Various constructions characterize them, but the most typical are p–n and p–i–n junction, UV/blue-enhanced and APD. A p-n type photodiodes are usually fabricated by diffusion or ion implantation. Interestingly, the p-i-n photodiodes with thicker active regions provide a broader near-IR spectral responsivity.

The spectral characteristics of typical planar diffusion silicon photodiodes are presented in Figure 14. The time constant of p-n photodiodes is limited to the range of microseconds, mainly due to the RC constant rather than the inherent speed of operation mechanism (drift and/or diffusion). In small-area detectors, the detectivity reaches a value from mid-10^12^ to 10^13^ cmHz^1/2^/W and is generally amplifier-limited.

The p-i-n detector is faster but less sensitive than a conventional p–n junction detector and has a slightly prolonged red response. It is a consequence of the increased width of the depletion layer since longer-wavelength photons are absorbed in the device’s active region. The inclusion of a very lightly doped region between the p- and n-regions and a moderate reverse bias create a depletion region of the total material thickness (about 500 µm for a typical silicon wafer). Higher dark currents collected from generation in the wider depletion layer result in lower sensitivity.

The limited radiation penetration depth is shown in Figure 15a, notably in the 100–300 nm region. In p-n and p-i-n photodiodes, the strongly doped p^+^ (or n^+^) contact surface generates carriers due to silicon’s high UV and blue spectral absorption coefficients. A short lifetime and rapid degradation of quantum efficiency are also caused by high and/or surface recombination. The response of blue- and UV-enhanced photodiodes at short wavelengths is optimized by a decreasing carrier recombination rate close to the surface. The use of extremely thin and highly graded p^+^ contacts (or n^+^ or Schottky metal), lateral collection reducing the part of the severely doped surface, and/or its passivation with a fixed surface charge to remove minority carriers from the surface provides the best response. The ideal lossless Si detector’s (i.e., lacking an additional absorbing top layer) wavelength dependence on a current responsivity for the UV spectral range is depicted in Figure 15b. The solid red line delineates the silicon internal quantum efficiency and defines the upper limit. The long wavelength limit for which one absorbed photon produces one electron is indicated by the red dashed line. Creating one electron-hole pair requires a constant energy of 3.66 eV in the short-wavelength range (red dotted line), while the silicon detector responsivity has a constant value of 0.273 A/W. The spectral responsivities taking into account the external reflection on the silicon surface are shown by the solid blue line [23].

Both silicon diffusion photodiodes (n-on-p and p-on-n) and Schottky barrier diodes are widely used for UV detection. Figure 16 shows vertical cross-sections of the general construction of silicon photodiodes. The n-on-p structure contains a nitrided passivating SiO_2_ surface layer. In addition, the positive charge in the oxide improves the surface charge collection efficiency. The disadvantages of this type of n-on-p structure, however, are reduced extreme UV (EUV) responsivity due to the optical properties of the oxide layer and the need for a critical processing step to nitridate the Si-SiO_2_ interface (without this last process, the photodiodes exhibit low radiation hardness).

Because the capping silicon oxide layer has a naturally positive charge, typical p-on-n junction photodiodes have poor responsivity and stability. The photogenerated electrons (minority carriers) cannot be collected since they are directed away from the p-on-n junction by the electric field created by this positive charge. Nevertheless, the p^+^-active surface in p-on-n type photodiodes is formed by, e.g., pure boron using chemical vapor deposition (CVD). The extremely shallow junctions are produced via the formation of nanometer-thick, amorphous B and delta-like B-doped layers on the Si substrate, considerably enhancing sensitivity to UV radiation.

In addition to Schottky barriers, p–n junction photodiodes appear to have appreciable advantages. The thermionic emission process in the diffusion one is vastly less efficient than in a Schottky barrier, in which the built-in voltage tends to be lower than in the case of a p–n junction implemented in the same semiconductor material. Consequently, the saturation current in the first type of photodiode is several orders of magnitude higher than in the second one for a given built-in voltage.

Figure 17a compares the spectral responsivity of silicon p–n, n–p, and Figure 17b shows Schottky photodiodes (SPDs) in which the PtSi platinum silicide layer (Schottky contact) is placed on an n-type silicon substrate. It is shown that the PtSi photodiode responsivity is lower than the n–p junction one. PtSi-n-Si devices are characterized by relatively small breakdown voltages and large dark currents. However, the advantages of this type of design include resistance to long-term vacuum ultraviolet (VUV) radiation.

Traditional UV photodiodes typically use a fixed doping profile near the semiconductor surface, which results in a negligible electric field, limiting the collection efficiency of the photogenerated carrier. Kuroda et al. recently developed a UV-light-robust Si photodiode technology compatible with a CMOS image sensor [25,26]. Figure 18a shows the cross-sectional view of PD. An annealing process in ultrapure Ar was carried out to flatten the Si surface atomically. The ion implantation process buried the p-layer, while oxygen radical oxidation formed a surface n^+^-layer through the oxide film. It was found that photodiode sensitivity changes occur due to the charging of the SiO_2_ layer over the photodiode during exposure to UV light, which induces a difference in the Si potential near the surface. That is why it is necessary to use a steep doping profile that induces a photogenerated carrier drift field, with the dopant concentration at the interface high enough to suppress the change in the drift field caused by a fixed charge. The proposed strategy for designing the photodiode doping profile is important for developing sensors with high UV sensitivity and stability. The surface atomic flatness is one of the most critical parameters for forming a uniform dopant profile. At the p–n junction, dopant concentrations should be low to suppress the dark current generated in the depletion region. Figure 18b summarizes the PDs structure, achieving high sensitivity to UV light and high stability to intense UV irradiation.

The aforementioned results were reported by Xia et al. [27]. The carrier collection efficiency of the APD and its responsivity were enhanced thanks to a built-in electric field close to the surface and optimized gradient boron doping. It is characterized by a responsivity of about 0.1 A/W in the UV range and a very low capacitance of about 0.129 pF. Such a type of APD can be used in single-photon photoreceivers thanks to a gain of about 2800 and a reverse bias of 10.6 V.

The quantum efficiency of modern silicon APDs is moderate in the ultraviolet range and requires expensive optical filters to ensure solar photon rejection. They can operate in Geiger (GM) and linear (LIN) modes. In the case of LIN mode, the reverse-bias voltage of APD is below the critical value, and carriers are picked up faster than they are created, resulting in the avalanche process being terminated. The output photocurrent, produced by finite gain, is linearly proportional to the intensity of the optical radiation. The APD begins operating in GM when the biasing voltage increases and exceeds the breakdown point. A single carrier photoexcitation can cause self-sustaining avalanches, creating a macroscopic current pulse. It can be noiselessly detected using a threshold detection circuit because such a process is inherently digital thanks to the current pulse being large enough.

The GM provides a high output signal even in the case of single photon detection. In order to count the individual arrival of photons, the avalanche current needs to be passively or actively quenched between successive photons. For example, the passive circuit can stop the avalanche multiplication process thanks to the so-called quenching resistor connected in series with the APD (Figure 19a). It causes that photocurrent to flow and the operating voltage of an APD to be reduced [28].

GM-APDs are relatively mature and have two main constructions: single large-area GM-APDs and multi-pixel photon counters (MPPC). Many of these constructions are arranged in two-dimensional, electrically connected pixels. It is worth noting that the MPPC has excellent photon-counting capability, high gain, fast response, excellent time resolution, low noise, high photon detection efficiency, and a high S/N ratio. Figure 19b,c show the image and pulse waveforms of MPPC photon counting observed on an oscilloscope. Si-based APDs operated at liquid nitrogen temperatures with very low dark counts were demonstrated by T. Isoshima et al. [29].

**Figure 19 sensors-23-04452-f019:**
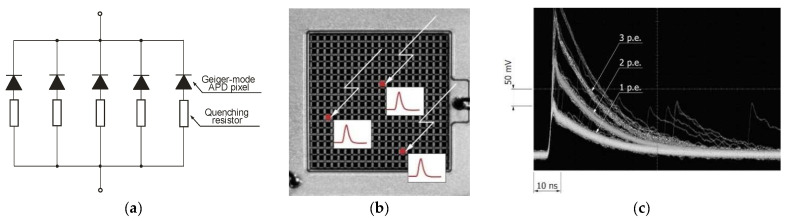
View of (**a**) structure, (**b**) image, and (**c**) pulse waveforms of MPPC photon counting. Reprinted with permission from Ref. [30].

In practice, there are many technologies to perform spectra matching on UV photoreceivers. The most popular of these uses optical filters (Figure 20a) or resonant cavity structures (Figure 20b). Optical filters with high transmittance in the UV range eliminate the influence of background radiation (visible radiation) and solar-irradiance leakage. Ideally, the responsivity of the selected photodetector should cover only the monitored wavelength. Figure 20a shows the responsivity of a silicon photodiode with and without a filter.

In typical surface-illuminated photodetectors, the small thickness of the absorption layer gives high bandwidth (limited by the transition time) but low quantum efficiency. Therefore, quantum efficiency and bandwidth are determined by compromise. On the other hand, resonant cavity enhanced (RCE) structures with a very thin absorption layer can surmount these difficulties. This layer is placed in an asymmetric Fabry-Pérot cavity of the RCE structure. Such a cavity is usually made of two parallel mirrors consisting of quarter-wavelength stacks (QWS) with periodic refractive index modulation [31,32].

The mirrors’ top and bottom are also frequently formed as distributed Bragg reflectors (DBRs). A thin absorption layer provides a large bandwidth, and simultaneously, multiple optical radiation transitions in the resonant cavity enhance the quantum efficiency. Consequently, both of the above-mentioned parameters can be optimized. Moreover, the resonant cavity acts as a narrow bandpass filter, and therefore the quantum efficiency of this photodetector is huge at the wavelength of interest and very small at others. A scheme of an RCE photodiode is portrayed in Figure 20b.

Another way of profiling the spectral characteristics of a silicon UV bulk sensor is to use a differentiated spectral response with high sensitivity in the ultraviolet wavelength band and low sensitivity in both the visible (VIS) and near-infrared (NIR) spectra. Figure 21 presents a UV silicon photoreceiver with high sensitivity without optical filters, but it is not effective during a UV radiation measurement under strong VIS and NIR backgrounds. The application of two photodiodes ensures overcoming that problem. High responsivity and high selectivity in the UV spectra required the extraction of the differential response of the ultrahigh-sensitive (PD1) and normal-sensitive UV (PD2) photodiodes [33].

In summary, Si-based photodiodes have a superior rank in UV detection technology regarding simplicity, high sensitivity, low costs, and integrated circuit compatibility. Excellent radiation hardness characterizes Schottky barrier photodiodes, but their dark current is relatively high and their breakdown voltage and responsivity are low. Decent responsivity of n-on-p junction diodes in the deep ultraviolet (DUV) spectral range relies on a positively charged oxide surface. Still, this surface affects stability and reduces responsivity in the EUV and VUV spectral ranges. However, recently developed technology of boron-based delta-doping materials allows designing a p–n junction photodiode characterized by higher responsivity in the DUV/VUV/EUV ranges than any commercially available UV silicon photodetector. However, the relatively high surface resistance of the sheet is a limiting factor for the high readout speed.

In their entirety, the above-described detectors typically operate as single-element devices. Charge transfer devices (CCDs and CMOS sensors) were discovered as compact image sensors for the consumer and industrial markets. Still, they are also leading visible and ultraviolet image sensors in many fields of scientific research, including space science and Earth science, as well as planetary remote sensing. Figure 22 compares the spectral characteristics of various imagers offered on the market.

Researchers have begun to develop thin, backside-illuminated devices to address the lack of silicon sensitivity to “blue” wavelengths. A more reliable and stable method involves introducing a thin layer of dopant (usually boron) into the back surface to form a p^+^ layer that creates a small degree of potential that repels photogenerated electrons towards areas of depletion.

The 4k × 4k CCD array operating in the near-UV and near-IR spectra that has been installed on the Hubble Space Telescope (HST) is shown in Figure 23a. Its read noise level is <3e^−^, and quantum efficiency is 30–60% in the spectral range from 200 nm to 300 nm [35]. An image registered using this array is shown in Figure 23b.

## 5. Wide Bandgap UV Photodetectors

The silicon photodetectors used for UV detection can face limitations caused by their material properties, i.e., in high-temperature operation, they must frequently be cooled to reduce leakage current and the detection system’s noise. What is more, an optical filter is often applied in order to block visible and infrared radiation. These treatments significantly elevate the complexity and costs. Therefore, investigations of the wide bandgap semiconductors for UV photodetectors (e.g., AlGaN, ZnMgO, SiC, diamond-based, and ZnS) have begun to overcome the mentioned disadvantages. Such materials provide many benefits compared to S-based devices, e.g., large breakthrough fields, insensitivity to solar or visible radiation (filters are unnecessary), high radiation resistance, and high thermal and chemical stability. The cut-off wavelength of these types of photodetectors is usually below 400 nm. In the case of photodetectors characterized by a cut-off wavelength of 280 nm or shorter, they can be called solar-blind because they respond only to radiation, the wavelength of which is shorter than the sun’s radiation penetrating the Earth’s atmosphere.

In space, the concept of solar blindness would have to be re-defined since there is no longer an atmosphere between the detector and the sun (Figure 24) [36]. In practice, however, wide-bandgap semiconductor detectors cannot offer such ideal properties. Real detectors are sensitive in the wavelength range below 200 nm but as little as possible in the wavelengths of 200–300 nm and longer ones. Figure 24 illustrates the mentioned spectral considerations for a diamond photoconductor with interdigitated ohmic contacts (metal-semiconductor-metal (MSM)). The dependence of the quantum efficiency on the spectral intensity of solar radiation as seen from space and the Earth at sea level is also illustrated.

### 5.1. AlGaN Photodetectors

III-nitrides currently represent one of the most interesting technical solutions for UV photodetection. Al_x_Ga_1–x_N (AlGaN) alloy is characterized by a direct band gap structure and relatively high carrier mobility. The saturation velocity of electrons is several times higher than that in silicon. This material absorbs UV radiation; however, it penetrates shallowly, depending on the wavelength. Compared to infrared detectors, in which the radiation fills the entire volume of the semiconductor, this imposes a specific way of construction they must follow/makes them limited to a particular structure.

By changing the Al composition in Al_x_Ga_1−x_N alloy from 0 to 1, the energy band gap of these materials varies from 3.4 eV to 6.2 eV, corresponding to a change in the long-term spectral responsivity from 200 nm to 365 nm. This spectral interval covers the ozone layer absorption “strategic window” (230 to 280 nm). The proximity of the long-term responsivity limit of the GaN detector (365 nm, *E_g_* = 3.4 eV) and the edge of the visible spectral range lead these detectors to detect UV radiation in a strong background of visible and infrared radiation. For these detectors, “blindness” to solar radiation, determining their quality, is based on the *rejection ratio*. Comparing different detectors (photomultipliers with photocathodes from SbKCs or CsTe), GaN detectors should be characterized by a rejection ratio above 10^6^. Heretofore, the highest published values of this parameter were close to 10^5^ for photodiodes. The obtained values are even lower in the case of Schottky-barrier and photoconductive detectors. Importantly, nitride semiconductors exhibit good electronic transport properties; they are physically robust, have high corrosion resistance, are chemically inert, and are nontoxic. These properties are also exploitable for use in hostile environments and at high temperatures.

Nowadays, Al_x_Ga_1−x_N alloys manufacture photoconductive, Schottky barrier photodiodes, p–n junction photodiodes, p-i-n photodiodes, avalanche photodiodes, metal-semiconductor-metal, metal-insulator-semiconductor (MIS) detectors, and phototransistors with the schematic designs listed in Table 1.

#### 5.1.1. Photoconductors

The simplest type of AlGaN detector is the photoconductive one. These detectors are made of a thin film with interdigitated metal contacts placed on their surface to maximize light transmission and minimize transit time. The photoconductive gain of these photoconductors reduces the requirement for low-noise preamplifiers. This gain is given by the ratio of the free carrier lifetime *τ* to the transit time *t_t_* between the detector electrodes:(19)$$g=\frac{\tau }{t_{t}}=\frac{\mu \tau V}{d^{2}}.$$

Here, *μ* is the electron mobility, *V* is the applied bias voltage, and *d* is the inter-electrode spacing. The carrier lifetime strongly depends on the semiconductor properties.

The gain in photoconductive detectors occurs because the recombination lifetime is much longer than the transit time. Its value depends on device dimensions and applied voltage. Additionally, the real figure of merit is defined by the *μτ* product of the material. The *μτ* product in AlGaN photodetectors is a strong function of the semiconductor resistivity and can vary by many orders of magnitude [37] (Figure 25). The surprising thing is that AlGaN photodetectors (with lower mobility than GaN due to alloy scattering effects and a shorter lifetime than GaN due to their more defective nature) have a higher *μτ* product than GaN. This fact is explained by the structure of AlGaN, which consists of atomically ordered domains. Thus, electron-hole pairs created by incident radiation are separated in the ordered domains, increasing the carrier lifetime.

The photoconductive gain and responsivity of AlGaN UV detectors may have different values [37,38]. However, a complicated issue is the relatively low response speed of AlGaN detectors (reaching milliseconds). It is determined by the high density of recombination centers in the energy band gap and the high density of dislocations of 10^10^ cm^−2^. Since the gain and the detector bandwidth product are constant, the response time is either long for a high photoconductive gain or short for a low photoconductive gain.

The normalized responsivities of Al_x_Ga_1−x_N photoconductors for x = 0, 0.21, 0.34, and 0.50 are presented in Figure 26a [39]. High sublinear photoresponse and excitation-reliant response times occur at relatively low excitation levels. It can be caused by increasing excitation levels, resulting in the redistribution of the charge carriers and a reduction of the photoconductive gain [40] (Figure 26b). Furthermore, the responsivity of this type of photodetectors depends on the persistent photoconductivity (PPC) phenomenon. The conductivity is elevated rapidly after the detector is exposed to radiation and can be sustained for an extended time after the light exposure stops. Consequently, the detectors’ operation is tremendously slow, and their responses are non-exponentially transient [41].

The frequency impact on a GaN photoconductor’s spectral response is depicted in Figure 26c. A slow detection mechanism outperforms the photoconductor’s poor and selective spectral response. Higher chopping frequencies improve the shape of its spectral characteristics. The spectral cut-off becomes sharper, and the spectrum below the bandgap approaches the photovoltaic response. The use of lock-in detection overcomes this limitation, but the detection procedure is more complex and expensive. The observed kinetics of photoresponse have important consequences for practical applications [37,42]. Photoconductors are not suitable for applications requiring high speed or a particular spectral selectivity.

To summarize, photoconductive detectors are not ideal devices in many real-world applications because of their slow response and poor UV-to-VIS rejection ratio. Photovoltaic detectors do not have these disadvantages.

#### 5.1.2. Schottky Barrier Photodiodes

The main parameters of photovoltaic detectors with different structures based on GaN and AlGaN are listed in Table 3.

Taking into account the device design, the simplest are Schottky barrier detectors, which have the advantage of a simple manufacturing process. They are characterized by high quantum efficiency, high UV/VIS contrast, high response speed, low dark current, and possible zero-bias operation.

The Schottky structure is made up of three parts: semiconductor (usually in the form of a thin film), ohmic contact, and Schottky contact (see Table 1). Metals with high work functions, such as Pt, Ni, Pd, or Au, are commonly used for Schottky contact processing.

Figure 27 illustrates the spectral response of AlGaN Schottky photodiodes (PDs) depending on the Al mole fraction. In contrast to photoconductors, a UV/VIS rejection ratio is typically between 10^3^ and 10^4^ [50]. The cut-off wavelength shifts with Al content from 295 nm (x = 0.35) to 360 nm (x = 0). As seen in the inset, the photocurrent of the device increases linearly with optical power for nearly five decades (over the entire measured range from 10 mW/m^2^ to 2 kW/m^2^). Current responsivities of 53 mA/W, 45 mA/W, and 29 mA/W were obtained for x = 0.19, 0.27, and 0.35, respectively. Detectivity *D** ∼1.2 × 10^9^ cmHz^1/2^W^−1^ was obtained in Al_0.22_Ga_0.78_N/Au detectors at a bias of −2 V, increasing to 3.5 × 10^9^ cmHz^1/2^W^−1^ for a bias of −1.35 V. Although the zero-biased Schottky barrier PDs do not exhibit photoconductive gain, the internal gain is present at reverse voltages, and a significant increase in responsivity is also observed.

The response time of AlGaN Schottky PDs is *RC*-limited, and time constants are in the nanosecond range [51]. These devices are especially suitable for broadband photodetection because their spectral response is flat (Figure 27).

MSM photodetectors comprise two Schottky contacts coupled back-to-back on an undoped semiconductor layer (Table 1), making them ideal for high-speed applications. With decreasing illumination wavelengths, the transparency of existing metals and metal oxides decreases dramatically [52]. As a result, the presence of opaque metal contact electrodes limits the quantum efficiency and detectivity of MSM photodetectors. Metal contacts with high transparency, a high Schottky barrier, and low screw dislocation are required for high-performance MSM devices. Moreover, increasing the annealing temperatures and introducing buffer layers can improve the device’s suitability for deeper UV detection [53,54]. Back-illumination can improve the sensitivity of MSM PDs when the substrate is transparent in the spectrum of interest. As a result, semi-transparent finger metallization is not required [55].

AlGaN-based UV MSM PDs with Schottky contact electrodes fabricated using MOCVD technology on a sapphire substrate were presented by J. Li et al. [49]. They stood out due to a dark current of about 20 pA at 1 V bias, a detectivity of 4.43 × 10^11^ cmHz^1/2^/W, and a responsivity of 0.19 A/W. The described UV-to-visible rejection ratio exceeded five orders of magnitude, with rise and fall times of 10 ns and 190 ns, respectively.

Pandit and Cho [56] described AlGaN/GaN UV MSM photodiodes with two interdigitated finger electrodes (back-to-back) made of reduced graphene oxide (rGO). Below the wavelength of 380 nm, the rGO demonstrated excellent transparency (>90%). When paired with AlGaN semiconductors, rGO as a transparent Schottky electrode has the potential to be used in visible-blind optical sensors.

Further improvements in detection performance can be achieved by using a resonant cavity or plasmonics. The first method was described by Kishino, with co-authors in Ref. [57], demonstrating an enhancement effect selectively at 352 nm and 363 nm in wavelength, with a factor of about two. In the paper of Li et al. [58], the application of Ag nanoparticles deposited on the GaN detector surface (nanoplasmonic enhancement) increased the responsivity of MSM UV detectors by over 30 times. Sun et al. [59] also demonstrated the reduction in the dark current using SiO_2_ nanoparticles (SNPs) to passivate the conductive dislocations. The reduction obtained was one order of magnitude lower, while the responsivity was measurably increased (Figure 28).

In summary, MSM photodetectors exhibit a sharp cut-off in the visible wavelength range, linear light response, low dark current, frequency response at the GHz level, and high sensitivity. For this reason, they tend to be desirable devices in military and civilian applications.

#### 5.1.3. p–n Photodiodes

UV detectors such as p-n, p-i-n, and avalanche photodiodes are crucial for many applications. Most UV photodetectors investigations focus on developing AlGaN-based p–i–n junction photodiodes, which demonstrate the capability of tailoring the cut-off wavelength by changing the alloy composition and, consequently, the bandgap energy of the active layer. A whole range of p-i-n Al_x_Ga_1−x_N photodiodes with a cut-off wavelength continuously controlled from 227 nm to 365 nm by an Al concentration in the range of 0–70% has been demonstrated. The current responsivity of these detectors at room temperature is shown in Figure 29.

The design and optimization of PD structures are critical to obtaining superior-performance devices. There is a trade-off between the p-i-n photodiodes’ responsivities and response time. The photodiode can achieve higher quantum efficiency with a thicker i-layer but a slower response time. As a result, for applications where speed is unimportant, a thicker depletion region is more efficient. When operating in photovoltaic mode (zero bias), their internal quantum efficiencies reached up to 86% and had a UV-to-visible rejection ratio of up to six orders of magnitude. With surface recombination and photoemission losses, the penetration depth is reduced, and the detection of charge carriers decreases at shorter wavelengths.

Although most AlGaN photodiodes are front-side illuminated, back-side illuminated devices have also been demonstrated. It was enabled as a consequence of using sapphire substrates that are UV-transparent. With silicon readouts, sapphire has only a modest thermal coefficient of expansion mismatch, although it is successfully applied in industrial production.

Nowadays, the most common UV detectors are based on photomultiplier tubes, which have high costs and a relatively short lifetime. Many commercial and military applications necessitate the detection of extremely weak optical signals. APDs are appealing for UV detection as they offer better durability under extensive usage, a lower cost, and better performance than a tube-based multiplier.

There are two types of photodiodes: those with an integrated region of absorption and multiplication and those with separate absorption charge and multiplication layers. In the first example, both absorption and avalanche multiplication of excited photo carriers occur in the “i” region. The schematic mechanism of impact ionization in a reverse-biased p–i–n junction is shown in Figure 30a. Impact ionization happens when one charge carrier (electron or hole) loses its energy after scattering with the crystal lattice, generating more energetic electrons and holes. Newly appearing electrons and holes continue their journey through the elevated electric field area, where they accelerate and finally lose energy when they scatter with the crystal lattice, creating more carriers and leading to an avalanche breakdown in the device.

The p-i-n structure necessitates a low bias voltage, which improves the breakdown voltage in APD-based imaging arrays. A comparatively thin absorption and multiplication layer reduces the bias voltage. The reduced thickness of this layer indicates insufficient UV absorption and low quantum efficiency. Solving this issue is possible thanks to APDs with separated absorption and multiplication (SAM) regions (Figure 30a,b). A thick absorption layer is used in this construction to absorb photons efficiently. A thin multiplication layer provides a reduced breakdown voltage, while a charge control layer profiles the electrical field within the device’s active region. A high electric field overreaches through the whole active layer of SAM-APDs in reach-through mode. This design increases the likelihood that more carriers created in the absorption layer will be swept into the multiplication layer, resulting in a higher output current. Another advantage of the SAM-APD device is that it injects only one type of carrier into the multiplication layer, which reduces multiplication noise, although the dark current elevates, causing a decline in SNR. As a result, the photodiode bias voltage should be carefully chosen to achieve the highest SNR value.

A silicon avalanche photodiode (Si APD) is the standard semiconductor device for detecting the visible part of the spectrum. Such detectors cannot achieve the SNR performance needed in UV applications. Unlike ordinary Si, III-V, and SiC diodes, in which the avalanche effect is well known, early GaN APD constructions were electron-initiated due to the absence of natural GaN substrates. A hole’s impact ionization coefficient is substantially higher than an electron’s, making hole-initiated APDs more appealing due to their low excess noise factor and high gain. Therefore, an APD with GaN-on-sapphire backside illumination (p-type on top) was designed and manufactured [61]. For bulk GaN substrates, however, the backside APD design (Figure 30a) is not possible. Such a structure of a SAM APD with the corresponding electric field distribution is shown in Figure 30b. It consists of a separated absorption and multiplication layer on top of a buried p-GaN layer. The epitaxial growth on a bulk GaN substrate is an advantage over back-illuminated structures.

**Figure 30 sensors-23-04452-f030:**
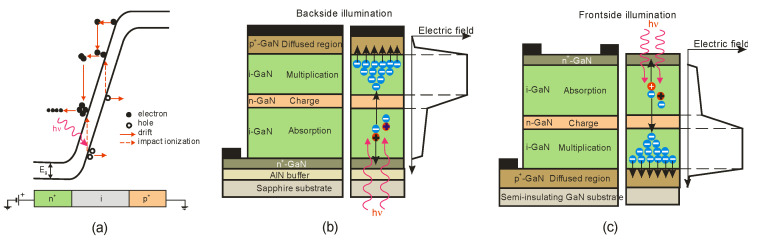
Avalanche photodiodes: (**a**) impact ionization process in a reverse-biased p–i–n junction; the separated absorption and multiplication (SAM) APD GaN structures: (**b**) based on GaN-on-sapphire and the corresponding electric field distribution along the vertical direction; (**c**) the proposed SAM structure based on GaN-on-GaN and the corresponding electric field along the vertical direction. Adopted with permission from Ref. [62].

Effective edge termination is an important requirement to ensure the robust avalanche capability of GaN-based devices. Edge termination provides additional negative charges to terminate the electric field lines at the device boundary to reduce any localized peak electric field at each surface or corner. Figure 31 shows several edge terminations proposed for GaN APDs, including an edge termination based on ion implantation and etching and a field-based edge termination in which the field plate is connected to the anode electrode [62]. They effectively uniformize the electric field distribution, help reduce the dark current, and prevent the early breakdown of a device.

Ji et al. [63] have developed a robust avalanche capability with a buried p-GaN design using the activation p-GaN technique and the Mg ion-implanted edge termination. The homogeneous p-i-n GaN structure was grown on free-standing GaN substrates with low dislocation densities of 10^5^ cm^−2^, which is 3–4 orders lower than those of GaN substrates from other manufacturers. It offers an excellent responsivity of 60 A/W (at a reverse bias of 280 V), an extremely low dark current of 1.5 × 10^−5^ A/cm^2^, and a high gain above 10^5^.

The main parameters of different avalanche photodiodes based on GaN and AlGaN are listed in Table 4. Their performance depends on the type of substrate. Due to the limited availability and high cost of commercially available UV-transparent free-standing AlN substrates, III-nitride films tend to be grown on foreign substrates, usually sapphire. There are serious problems with lattice constant mismatch and thermal expansion coefficient mismatch between GaN and sapphire substrates.

The significant lattice mismatch between GaN and alternative GaN substrates resulted in a high dislocation density of GaN epitaxial layers of 10^8^–10^9^ cm^−2^, severely inhibiting the impact ionization process. The high density of defects and the damage caused by etching contributed significantly to a high dark current density, which ranges from 10^−4^ to 1 A/cm^2^ [74,75]. With the introduction of epitaxial growth of the GaN active region on GaN substrates, a dark current density of less than 10^−4^ A/cm^2^ could be realized, with a gain of 10^4^. Significant progress in the visible-blind GaN APDs technology ensured the detection of a single photon [76].

Presently, GaN-based APDs can achieve gains from 10^5^ to 10^6^ and responsivities of about 1 A/W. What is more, they can detect weak, visible-blind signals. APDs must work at a moderate internal gain and need multiplication factor stabilization due to their unfavorable high-temperature sensitivity. Another disadvantage is that the APD operates at high voltage, and thus the noise is also amplified. Therefore, in the design of APD, both noise and gain parameters should be well balanced.

Kim et al. [68] compared the performance characteristics of Al_0.05_Ga_0.95_N UV APDs grown on different substrates. The research shows that UV-APDs grown on a free-standing GaN substrate show lower dark-current densities than similar UV-APDs grown on a GaN/sapphire template (Figure 32). Furthermore, the dark-current density of the UV-APD grown on the GaN/sapphire template increases dramatically even at a low reverse bias. Therefore, the high crystalline quality of Al_0.05_Ga_0.95_N grown on a free-standing GaN substrate with low dislocation density is responsible for the observed low leakage currents and high performance of the devices.

Ref. [46] reported high performances of a p-GaN-gated AlGaN/GaN heterostructure grown on cost-effective silicon substrates. The detector’s schematic structure and its spectral responsivity are shown in Figure 33. The detector is characterized by an ultra-high photon-to-dark current ratio of over 10^8^, a high responsivity of 2 × 10^4^ A/W, a cut-off wavelength of 395 nm, and a high UV-to-visible rejection ratio of over 10^7^. The high electrical gain results from the transistor-like operation of the device. The detector’s rise and fall times were 12.2 ms and 8.9 ms, respectively.

#### 5.1.4. Phototransistors

Phototransistors are light-sensitive transistors that typically have much larger base and collector areas compared to ordinary electronic transistors to enable them to absorb light more efficiently. As described in Table 1, phototransistors offer the advantage of high photoconductive gain (the so-called photogating effect), which results in high current responsivity. However, one of the main disadvantages of the phototransistors is their relatively long response time. Phototransistors are three-terminal devices that reduce dark current and noise by adjusting the gate voltage.

Figure 34a shows a recently reported high-performance Ni/Au-gated AlGaN/GaN UV phototransistor structure [77]. Due to the inherent polarization effects of III-nitride materials, two-dimensional electron gas (2DEG) with high concentration and high mobility exists on the AlGaN/GaN heterointerface, which increases photocurrent and photoresponsivity. This device exhibits a very high responsivity of 3.6 × 10^7^ A/W at 265 nm illumination (an absorption area of 20 µm^2^)—see Figure 34b. The ultrahigh detectivity value for this photodetector, equal to 6.5 × 10^18^ Jones, is overestimated—see discussion in Section 2. The phototransistor is simultaneously characterized by a very low dark current of ~20 pA and a high external quantum efficiency of 2.7 × 10^5^. The last value indicates the effective transport of photogenerated charge carriers into the GaN channel (strong photogating effect).

Armstrong et al. [78] have demonstrated visible- and solar-blind photodetectors using Al_0.45_Ga_0.55_N/Al_0.30_Ga_0.70_N and Al_0.85_Ga_0.25_N/Al_0.70_Ga_0.30_N high electron mobility transistors (HEMTs), respectively. For the visible-blind Al_0.45_Ga_0.55_N/Al_0.30_Ga_0.70_N HEMT, a peak current responsivity of 3.9 × 10^6^ A/W was observed, while for the solar-blind Al_0.85_Ga_0.15_N/Al_0.70_Ga_0.30_N HEMT, the peak responsivity was 4.9 × 10^4^ A/W. The analysis showed that the sub-bandgap absorption of incident UV radiation by defect states plays a major role in determining the photoresponse of the transistor. Defect states affect the slow photocurrent rise and fall times, while electrical pulsing improves the bandwidth at the cost of optical gain.

#### 5.1.5. Focal Plane Arrays

UV AlGaN solid-state imagers are mainly produced with a hybrid construction and standard 320 × 256 pixels (Table 5). Such devices employ the deposition of the array structure on a transparent substrate and back illumination through the substrate. However, illumination from the front is necessary for reaching shorter wavelengths than the cut-off wavelength of the substrate material, e.g., in the case of VUV applications. Backside illumination is conceivable in the EUV (i.e., below 30 nm) using a backside thinned device (e.g., removing the Si substrate [79]) and making it transparent to the EUV and X-ray radiation.

A cross-sectional view of an AlGaN p-i-n back-illuminated photodiode is shown in Figure 35 [80]. The first bottom layer of the n-type Si-doped AlGaN window serves as the array’s common n-side contact, and its alloy composition establishes the cut-off wavelength. The following absorption layer is an isotype n–N heterojunction, with the window layer formed by an unintentionally doped (*uid*) n-type AlGaN with a smaller band gap. The next p-type AlGaN layer doped with Mg creates a p–n homojunction and the n-type absorption layer. The final layer is a thin, strongly Mg-doped GaN cap layer for p-side contact.

Recently, the first 640 × 512 pixel hybrid AlGaN arrays with a 15-µm pitch have been demonstrated [91]. The wafers were grown by metalorganic chemical vapor deposition of AlGaN epilayers on sapphire substrates and, after processing, hybridized with CTIA ROIC (Figure 36).

### 5.2. SiC-Based Photodetectors

SiC detector technology is promising for UV applications as it provides a wide bandgap (2.4–3.3 eV) depending on the polytype and guarantees the performance of devices up to 1000 °C. Thus, selective UV detection in a sunny environment is feasible without needing any optical filter, and it also ensures frequency bandwidth up to 100 GHz. Moreover, SiC has a defect density many orders of magnitude lower than what is possible in GaN technology. Furthermore, because of the small concentration of intrinsic carriers, it is possible to reach a very low reverse dark current. Consequently, SiC photodetectors can record very low signals thanks to their high detectivity, up to 3.6 × 10^15^ cmHz^1/2^/W.

Generally, SiC-based photodetectors are able to achieve substantial gains and high SNR values. They also ensure solar-blind operation as well as excellent effective thermal conductivity and long-term stability even under severe UV radiation (up to 1000 W/m^2^) [92,93]. On the other hand, SiC is an indirect bandgap semiconductor that is characterized by a small optical absorption coefficient. Nevertheless, another advantage of its usage comes from the strong bonding in SiC, which improves device radiation hardness. What is more, SiC suffers less from device aging because of the radiation damage, resulting in much longer lifespans under intense, high-energetic radiation applications [94]. All aforementioned features make SiC-UV detectors excellent for selective UV detection.

Numerous research teams concentrated on creating effective and durable UV SiC photodetectors employing 4H and 6H crystal structures. Despite the fact that photodiodes fabricated in 6H-SiC have been studied in earlier research [95], 4H-SiC has received greater attention in many application domains for its improved electronic parameters. In comparison with other wide-bandgap materials, 4H-SiC has a great thermal conductivity of ∼4.9 Wcm^−1^K^−1^ (e.g., three times larger than Si) and an excellent electron saturation velocity of 2 × 10^−7^ cm/s (two times larger than that of Si). UV photodetectors based on 4H-SiC typically have a responsivity spectrum of 220–380 nm, with a maximum value at around 290 nm.

Over time, the SiC-based UV detector technology has been improved by adopting different constructions to increase responsivity and SNR. UV SiC photodetectors can be available in five topologies: Schottky barrier, MSM, p-n, p-i-n, and avalanche photodiodes. Ref. [96] shows UV-A Schottky PDs containing an optical filter made of SiN:H layers formed on the Ni_2_Si surface (Figure 37a). The photodiode with a 6 µm epilayer thickness has a wide UV responsivity with a maximum of 45 mA/W at 290 nm (Figure 37b). However, applying the thicker epilayer of 25 µm moves the responsivity spectrum towards longer wavelengths, and a maximum of 42 mA/W at 310 nm is reached. It is because of the photocarriers created at long wavelengths below the space charge area. On the other hand, SiN:H films cause a decrease in detector responsivity within the UV-C range and move the maximum responsivity to the UV-A range. At 330 nm, the detector has a very small leakage current density of roughly 0.2 pA/mm^2^ and a maximum responsivity value of 27 mA/W at 330 nm (at zero bias). It is important to mention that integrating SiN:H films with a UV Ni_2_Si/4H-SiC Schottky photodiode appears to be effective for constructing a monolithic UV-A/UV-B sensor designed for high irradiance conditions [97].

Wide-band gap semiconductors benefit from MSM SiC photodetectors based on two back-to-back Schottky contacts. MSM PDs have a high bandwidth, a low intrinsic capacitance, and a shorter response time than nanoseconds. They can be easily combined with field-effect transistors on a single chip without requiring any complex fabrication techniques. Many 4H-SiC-based Schottky and MSM detectors have been built employing metal and semi-transparent electrodes, such as Ni, Ti, and Ni/ITO [98,99,100].

Lien et al. [101] demonstrated 4H-SiC MSM PD operating at temperatures as high as 450 °C. The responsivity below 325 nm illumination was 30.5 mA/W at 20 V bias. The increase in temperature from 25 °C to 400 °C caused changes in the rise time and fall times of the PD from 594 to 684 μs and from 699 to 786 μs, respectively. The photocurrent-to-dark-current ratio of the PDs was as high as 1.3 *×* 10^5^ at 25 °C and 0.62 at 450 °C. This work demonstrates that 4H-SiC MSM PD can be used in extremely harsh environments. Figure 38 shows a schematic and optical image of the structure.

It is generally true that a standard 4H-SiC MSM detector has a meager responsivity and quantum efficiency due to reflection-caused optical losses and the effects of surface states. Zhang et al. [102] proposed Al_2_O_3_/SiO_2_ (A/S) films as antireflection coatings and passivation layers on the detectors to mitigate these losses (Figure 39a). With a bias voltage of 20 V, the photodetectors achieved the maximum responsivity of 0.12 A/W at 290 nm and the highest external quantum efficiency of 50% at 280 nm (Figure 39b). The MSM made of A/S/4H-SiC MSM has a greater UV-to-visible rejection ratio and a higher photocurrent than 4H-SiC MSM.

Further improvement of detector performance can be achieved by annealing. Ref. [103] showed that devices after annealing had lower dark current, higher light current, and higher responsivity compared to conditions in deposition. These parameter improvements are contingent on increasing the height of barriers and ideality factors after annealing at appropriate temperatures.

He et al. [104] have presented an outstanding performance of epitaxial graphene on silicon carbide (epigraphene) grown at high temperatures (*T* > 1850 °C). This paper shows that epigraphene on SiC MSM acts as a material for implementing solar-blind UV MSM detectors with high performance. It is suggested that the high-temperature process required to produce a high-crystalline epigraphene is the reason for the reduced concentration of defects at the graphene-SiC substrate interface, which could act as charge traps and/or recombination centers.

The MSM structures are made up of a reproducible junction between the high-purity semi-insulating 4H-SiC substrate and a graphene layer produced on top (Figure 40c). The MSM devices are built with two Schottky diodes (D_1_ and D_2_) coupled back-to-back through the SiC channel resistance *R* (Figure 40a,b). The Schottky barriers created at the left and right interfaces of SiC are shown in energy band diagrams of the device in contact with metal and graphene (acting as the metal). In the dark and with no applied voltage, the interfaces form a depletion type of contact in equilibrium. When UV photons are absorbed at the graphene interface, electron-hole pairs via interband transitions are generated. The carrier photogeneration causes a decrease in the Schottky barrier, and under forward bias, free electrons are gathered on the graphene contact (Figure 40b).

The authors of Ref. [104] have developed epigraphene MSM detectors with a peak external quantum efficiency of about 85% for wavelengths 250–280 nm, nearly 100% internal quantum efficiency, a current responsivity of 134 mA/W, low dark currents of 50 fA, and high detectivity *D** = 3.5 × 10^15^ Jones.

Generally, p-i-n SiC photodiodes deliver a low-noise, high-speed response, and high responsivity at low reverse bias due to a low terminal capacitance and a significant shunt resistance. Cai et al. [105] reported a p-i-n 4H-SiC photodiode with a sensitive area of 200 × 200 µm^2^, whose schematic structure of a chip and responsivity are shown in Figure 41a,b, respectively. At room temperature, the dark current is 1.8 pA at a reverse bias of 5 V and gradually increases to 9.7 pA at a reverse bias of 10 V, mostly due to the thermally ionized carrier concentration increase. The peak responsivity at room temperature is 0.13 A/W at 266 nm, slightly increasing to 0.15 A/W at 268 nm as the temperature increases to 450 K. As can be seen, the responsivity in the wavelength range above 260 nm increases, and the response spectra shift toward longer wavelengths as the temperature increases. It is due to the effect of a narrowing of the bandgap due to the higher temperature.

Table 6 compares the major parameters for Schottky and p-i-n detectors based on SiC reported in the literature.

Avalanche photodiodes are also made from silicon carbide. 4H-SiC has emerged as a candidate for low-level UV visible-blind detectors owing to its material maturity and stability. In addition, 4H-SiC APDs with a large ionization coefficient ratio of ~10 between holes and electrons have a low excess noise factor and a high internal multiplication gain, improving responsivity. The structures of 6H-SiC are used to extend operation to longer wavelengths. They offer a narrower bandgap of 3.03 eV compared to the 3.26 eV for 4H-SiC designs [112].

Bai et al. achieved 4H-SiC p-i-n APDs with extremely low dark current [113]. A schematic of the top-illuminated APD and its characteristics of reverse current voltage and gain voltage at room temperature are shown in Figure 42. At 280 nm, a photodiode is characterized by a photocurrent gain of 1000, an external quantum efficiency of 41%, a multiplied dark current of 5 fA (63 pA/cm^2^), a maximum responsivity at unity gain of 93 mA/W, and an excess noise factor *k* = 0.1. The minimum detectable power of continuous flux is approximately 20 fW. Because of the dead region around the active area, these detectors have a low fill factor, a fundamental limit for developing high-density APD arrays. Furthermore, many photodetectors have a high breakdown voltage, limiting their applicability in various applications.

The compromise between the maximum achievable responsivity, fast response time, and low operating voltage cannot be resolved by 4H-SiC APDs based on p-n or p-i-n structures [114,115]. The authors of Ref. [114] developed SAM-APD with a high outcome by using the n^+^-4H-SiC substrate with three epilayers made of n^−^, n, and p^+^-type layers. The schematic cross-section of the SAM-APD with a diameter of the optical window of 220 µm is shown in Figure 43a. The APDs’ problems mentioned above have surprisingly been solved by optimizing the thicknesses and concentrations of doping as well as absorption and multiplication layers. The spectral responsivities of the detector and its quantum efficiencies measured at different reverse bias voltages are presented in Figure 43b. Apparently, responsivity increased from 78 mA/W to 203 mA/W when the bias voltage was changed from 0 V to 42 V. A quantum efficiency of 93% proves the high performance, as does a dark current ~10 pA (at a reverse bias of 27.5 V), a UV/VIS rejection ratio higher than 10^3^, and a detectivity of 3.1 × 10^13^ Jones at 0 V bias.

The authors of Ref. [116] describe the fabrication of 4H-SiC p^+^-n APDs f in a fully planar method (Figure 44a). A thin TEOS (tetraethylorthosilicate) ring surrounds the device laterally. Low-energy implantation of Al+ ions onto an n-type 4H-SiC epitaxial layer defines a shallow p^+^-region in the active area. A planar edge ring formed by the deeper implanted p-region surrounds the detectors’ active region, preventing edge breakdown. This method ensures a significant increase in the geometrical fill factor. Such detectors have a low leakage current (<10 nA/cm^2^ at −30 V bias). At 0 V and −30 V, the highest responsivity of 60 mA/W and 100 mA/W were achieved at *λ* = 270 nm (Figure 44b). These responsivity values result from a quantum efficiency of about 29% and 45%, which is caused by an increase in the thickness of the depletion layer. At −94 V, the detector’s gain approaches 10^5^, while the breakdown voltage is around 88 V.

Table 7 compares significant parameters for APD detectors based on 4H-SiC reported in the literature.

In many applications, ultra-low-level UV detection is required, sometimes at a single photon level. SiC detectors can be used as photon-counting detectors. They are well-suited to situations where a weak signal is present in a relatively low background, e.g., bio-aerosol detection, lidar, communication, and low-light imaging [118,119]. The first 4H-SiC single photon counting avalanche photodiodes (SPAPD) were reported in 2005 [120,121]. The low dark current of SiC SAM-APDs allows operating in the linear “quasi-Geiger” and Geiger modes without any damage or degradation of the devices.

### 5.3. Diamond-Based Photodetectors

Developing diamond-based UV photodetectors enabled detection in the range between UV and visible, with a short cut-off wavelength of λ_c_ = 225 nm and high contrast (6 orders of magnitude) [122]. Diamond is characterized by numerous extraordinary properties among all semiconductors, such as the highest mechanical hardness. Ultra-wide bandgap energy (*E_g_* = 5.5 eV), a vast operating temperature, the highest thermal conductivity (22 W/mK), high mobilities of the charge carriers (4500 cm^2^/Vs for electrons and 3800 cm^2^/Vs for holes), a high breakdown electric field (above 10 MV/cm), a low dielectric constant (*ε* ≈ 5.7), high saturation velocity, and high chemical and thermal stability [123,124]. Therefore, it appears to be a distinctly potent material when it comes to producing electronic and optoelectronic components that have to withstand operating at large temperatures and suit high-power and high-frequency applications.

For materials described above, there are different kinds of diamond photodetectors, e.g., photoconductors, Schottky barrier photodiodes, MSM photodiodes, MIS structures, p-n, p-i-n photodiodes, and phototransistors [125] (Table 1). Generally, their detectives depend on the quality of the material used. An ultra-low dark current and solar blindness (up to 280 nm) can be achieved by high bandgap materials. Moreover, diamond is significantly stronger considering optical radiation hardness upon high-power DUV illumination because of its single-element nature and strong chemical carbon-carbon bonds [126].

Photoconductors, the simplest type of photodetectors, are made of a diamond layer including two ohmic contacts, generally with an interdigitated-finger structure. Because of the high barrier that is generated by most of the metal electrodes and the lack of strong doping levels for diamonds, the deposition of low-resistance ohmic contacts (typically obtained by Au, Ti/Au, Ti/TiN, or Ti/WC) is the main challenge of fabricating diamond-based photodetectors.

The MSM planar photoconductors presented by Liao and Koide [127] consist of unintentionally doped homoepitaxial diamond thin films [127]. The dark *I*-*V* characteristics of the photoconductor with a very low dark current, ~10^−12^ A at 20 V, are shown in Figure 45a. The *I*-*V* characteristics of the MSM photoconductor after annealing (at 600 °C for 1 h in argon ambient) are shown in Figure 45b compared to a standard as-fabricated device designated as MSM-PD. At 2 V, the photoconductor current is approximately four orders greater than the MSM-PD, and the photoconductor saturation results from the space-charge-limited photocurrent [128]. The spectral responsivity of the photoconductor for a bias of 20 V and the MSM-PD is shown in Figure 45c. At 210 nm, the maximum responsivity was achieved, while the response time measured using a 193 nm pulsed laser was approximately 10 ns. Between 210 nm and the visible range, the photoconductors have a UV/visible contrast (visible-blind ratio) of up to 10^8^. At a bias voltage of 3 V, the spectral responsivity reached about 6 A/W for 220 nm, resulting in a photoconductivity gain of 33.

Lin et al. [129] reported solar-blind photodetectors constructed from diamond employing interdigitated graphite. The graphite was manufactured via graphitization of diamond using a laser beam as electrodes. The responsivity and detectivity of the photodetector can reach up to 21.8 A/W and 1.39 × 10^12^ cmHz^1/2^/W when the bias voltage is 50 V. It should be noted that both are among the best values ever reported for diamond-based photodetectors. These photodetectors are used as the sensing pixels in an imaging system.

In general, the advantages of MSM-PDs include a low dark current, high speed, high signal-to-noise ratio, and simple preparation. They require only an undoped or a single dopant active layer, which is particularly suitable for WBG semiconductor materials that are challenging to dope. The main limitation of MSM photodiodes is their relatively lower external quantum efficiency caused by electrode shadowing. As active layers, the MSM diamond photodetectors were made using natural, synthetic, and CVD polycrystalline diamonds [130]. The natural diamond detectors show greater quantum efficiency and responsivity than the polycrystalline and synthetic ones (Figure 46). Poor performance of photodiodes based on synthetic diamonds is a consequence of low-quality polycrystalline diamond crystals and high concentrations of impurities in diamond.

Due to the development of chemical vapor deposition (CVD) technology, a high-quality diamond can be obtained [131]. Thanks to this, the performance of the diamond photodetectors could have been significantly improved. Balducci et al. presented a photodetector with a UV/visible rejection ratio of about 10^6^ [132]. Its time responses shorter than 5 ns and 0.2 s for a pulse and continuous mode were, respectively, observed. In another example of an MSM photodetector, the device’s finger spacing and active area based on the boron-doped homoepitaxial diamond were 10 µm and 52 × 10^−3^ mm^2^, respectively [133]. The structure is characterized by a UV/visible rejection ratio of about 10^7^ and a very low dark current (<1 pA).

The MSM photodetector construction based on microcrystalline diamond films with Al interdigitated finger electrodes has been described by Wang et al. [134]. It is characterized by the slow response, high responsivity, and sizable gain resulting from the shallow level trapping photogenerated electrons (minority carriers) during the irradiation.

Liao et al. present Schottky photodiodes (SPDs) made from Ti/tungsten carbide (WC). They contained the ohmic and semi-transparent WC contacts (Schottky contacts) for a boron-doped homoepitaxial p-diamond epilayer [135]. The presented SPDs are characterized by a responsivity of 0.99 A/W (at a reverse bias of 3 V), a deep UV/visible-blind ratio of 10^5^, and a dark current of 10^−12^ A (at a reverse bias of 1 V).

The advantages of the MSM structures and the SPD were implemented in a two-mode SPD containing inter-digitalized-finger (IDF) ohmic and Schottky contacts (Figure 47a) [135]. The IDF is 10 µm wide and 390 µm long, with Schottky and ohmic contact spacing of 20 µm. The SPD-IDF can operate in either forward or reverse mode. It provides high responsivity in the forward mode because of its ohmic contact nature. As for the reverse mode, a fast response can be obtained due to the inherent properties of the Schottky contact. The dark current of this type of SPD is lower than 10^−14^ A at reverse biases and is about 10^−12^ A at forward biases up to a value of “±” +/−30 V.

The photocurrent and responsivity vs. biasing voltage characteristics of the IDF-SPD, assuming illumination with 220 nm, radiation are presented in Figure 47b. A responsivity of above 18 A/W is reached at a forward bias of −23 V. The response time was longer than 10 ms at a forward bias of −32 V; however, it was shorter than 10 ms at 32 V. In the region from 210 nm to visible light, rejection ratios of about 10^7^ and 10^8^ are observed using an inverse bias of 20 V and a forward bias of −10 V, respectively.

The p-i-n photodiodes consisting of a CVD diamond ensure deep UV and extreme UV detection, as reported by BenMoussa et al. [136]. In Figure 48, a construction of the photodiode providing a high responsivity of around 27.2 mA/W at a narrow spectral range of approx. 200 nm with an active area of 13.85 mm^2^, is shown. Moreover, the values of the dark current and the current rejection ratio between 210 nm and 310 nm in the case of zero bias are about 4 × 10^−13^ A and 10^6^, respectively.

Table 8 compares the photoelectrical properties of diamond photodetectors based on different types of devices.

With the advancement of diamond device technology, prototype detectors such as single-pixel detectors, position-sensitive detectors, and sensor array detectors have been made. For example, Girolami et al. [140] fabricated a beam profiling detection system (Figure 49a). This system contained a 36-pixel array detector (750 × 750 μm^2^, 150 μm spacing) on a 10 × 10 mm^2^, 270 μm thick polycrystalline diamond. In turn, Figure 49b shows a 12-pixel single-crystal diamond-based mosaic detector [141].

### 5.4. Ga_2_O_3_-Based Photodetectors

Discovered in the 1950s, Ga_2_O_3_ initially emerged as a potential substitute for potent and diverse applications, including but not limited to optoelectronics, semiconductor devices, and FET transistors (Table 9) [142]. Regarding its various atomic compositions, seven separate crystal structures of Ga_2_O_3_ can be mentioned: β-monoclinic (4.9 eV), α-corundum (5.2 eV), amorphous, γ-defect spinel (4.54 eV), ε-orthorhombic (4.27 eV), polycrystalline, and δ-orthorhombic (4.62 eV) phases. Among the different phases, β-Ga_2_O_3_ is the most widely used in solar-blind photodetectors as it is the most stable [143].

Due to the hole’s huge effective mass and limited mobility, most doped holes tend to become trapped by local lattice structures, resulting in a very low p-type probability. Unlike other wide band gap materials, Ga_2_O_3_ has an adequate band gap and thus does not require modifying doping, preventing fluctuations in alloy composition and phase separation. More importantly, 6-inch Ga_2_O_3_ single crystals and epitaxial films are now available [144].

Ga_2_O_3_ UV photodetectors are manufactured from materials in a variety of forms, from single crystals and epitaxial films to nanostructures of both binary materials and their ternary alloys [143]. In comparison with common WBG semiconductors, Ga_2_O_3_ has unique advantages, including a suitable bandgap, the availability of a single-crystal substrate, and the cost-saving growth of heterostructures. However, the basic parameters of photodetectors, such as current responsivity, response speed, and detectivity, are scattered over a rather wide range due to different detector architectures, different material phases, and their crystal properties. Due to the physical mechanisms involved in carrier trapping and persistent photoconductivity, a large scatter in photoelectric gain is observed.

Figure 50 presents different structures of Ga_2_O_3_ ultraviolet photodetectors grown by various techniques [145], including plasma-assisted MBE, sol-gel technique, pulsed spray pyrolysis, MOCVD, CVD, atomic layer deposition, radio frequency sputtering, and microwave irradiation. Ga_2_O_3_ is also mechanically exfoliated and quickly transformed to a substrate (e.g., in the fabrication of FET phototransistor—Figure 50d). Performance improvement has been achieved through various approaches, such as the use of a buffer layer, temperature optimization during material synthesis, surface passivation, rational impurity doping, carrier multiplication processes, and contact electrode optimization.

Table 10 compares the performance of Ga_2_O_3_-based UV photodetectors with different values of responsivity, detectivity, UV rejection ratio rise time, decay time, and dark current [145]. Over the past few years [146,147,148,149,150,151,152,153,154,155,156,157,158,159,160,161,162,163], the available literature has shown great progress in improving the performance of Ga_2_O_3_ ultraviolet detectors (Figure 51), even exceeding the physical limit of detection determined by signal fluctuations (signal fluctuation limit—SFL) for FET phototransistors.

Figure 52 compares the current responsivity of the β-Ga_2_O_3_ photodiode with commercial devices, including GaN, SiC, and AlGaN wide bandgap semiconductor materials (Boston Electronics Corporation: data for GaN, AlGaN, and SiC are for devices with part numbers AG38S-TO, AG32S, and SG01S, respectively) [164]. As can be seen in the figure, the β-Ga_2_O_3_ photodiode demonstrates performance superior to commercial devices in the UVC range below 225 nm.

Figure 53 shows the detectivity-gain relationships collected from the literature for different UV photodetector structures with peak detections in the range from 250 to 360 nm [165]. The largest values of *D**, including unrealistic values (exceeding SFL limits, see Section 2), are marked for Ga_2_O_3_ FET phototransistors and a new generation of photodetectors with active areas containing low-dimensional solids. As noted in Section 2, the main reason for the inflated detectivity values is believed to be the incorrect (or lack of) inclusion of internal gain in the shot noise value. The error in shot noise estimation increases with higher gain and is particularly significant for photodetectors with very high internal gain (above 10^8^, as marked in Figure 53).

#### Other Wide Bandgap Photodetectors

Among the various ultra-wide bandgap semiconductors, materials such as II–VI semiconductors (e.g., ZnMgS, ZnMgSe, and ZnMgO) [94] have emerged as suitable materials for use in solar detection devices due to their high radiation resistance and high chemical and thermal stability.

Compared with AlGaN alloys, ZnO-based materials have many impressive advantages, such as a lower density of defects and stronger radiation hardness. In turn, alloying ZnO with MgO increases the bandgap of ZnO from 3.37 eV to 7.8 eV, which indicates a great potential application in UV detection. In the latter case, the problems of obtaining p-type material are revealed, which makes it difficult to master p–n junction photodetector technology. However, MSM structures and Schottky junctions are successfully fabricated with excellent performance.

It is well known that ZnO is stabilized in the hexagonal wurtzite structure, while MgO is stabilized in the cubic rock salt structure. In consequence, with increasing Mg content, the crystal structure of ZnMgO alloy changes from wurtzite structure (w-ZnMgO) to cubic structure (c-ZnMgO) [166] (Figure 54). For this reason, the coexistence of two structures in ZnMgO seems to be unavoidable in the structural transformation process, and this ZnMgO with a hexagonal and cubic phase is usually called mixed-phase ZnMgO (m-ZnMgO). Thus, ZnMgO photodetectors are divided into three device structures: w-ZnMgO, c-ZnMgO, and m-ZnMgO.

The performance of ZnMgO photodetectors strongly depends on the structure of the semiconductors [167]. Since the wurtzite structure is the most stable crystalline structure, w-ZnMgO is the most commonly used material for UV photodetectors. Table 11 collects selected parameters of MgZnO photodetectors [168]. The w-ZnMgO photodetector usually has a high current responsivity, but its dark current is also high. In turn, the dark current of the c-ZnMgO device is very small; however, its responsivity is low. In the case of the m-ZnMgO detector, its performance is better compared to a single-phase device (lower dark current and higher responsivity). An example of the spectral characteristics of the MSM Mg_0.46_Zn_0.54_O photodetector is shown in Figure 52. In general, however, the performance of ZnMgO photodetectors is still far from the level of practical applications and needs further improvement, especially control of p-type doping, reduction of dark current, and improvement of repeatability.

## 6. New Concepts of UV Photodetectors

Nowadays, nanotechnology and its methods of fabricating versatile materials have initiated new ideas for UV photodetector design geared towards lower energy consumption and greater miniaturization. In this section, ways of realizing devices by exploring new materials and novel architectures for UV photodetectors inspired by developments in the physics of low-dimensional devices are mainly discussed in a brief way. An attempt is also made to further define their development towards next-generation photodetectors. Atomically thin two-dimensional (2D) materials show great potential and can be considered as alternative materials for well-known and technologically advanced and developed ultraviolet photodetector fabricated from A^III^B^V^ and A^II^B^VI^ compounds. This is due to the fact that the performance of conventional photodetectors made from bulk materials is limited by rather large dark currents. One solution to this problem seems to be the use of nanomaterials and nanostructures as active layers in UV photodetectors.

In this section, we first review a selection of new-concept UV photodetectors that have recently emerged from traditional devices and exhibit various special properties and novel potential applications. Then, the challenges and opportunities of novel detectors in practical applications will be discussed.

New concepts in the design of a new generation of UV devices are being conditioned by:Introduction of various methodologies developed for fabricating semiconductor nanostructures, which can be divided into two categories: top-down (where the dimensional reduction is realized by lithography or patterning techniques) and bottom-up (where the desired nanostructures are built up by individual atoms and molecules) [169];Plasmonic enhancement of photodetector performance [170];Introduction of low-dimensional solid materials such as 2D materials and various nanostructures [171,172];Development of self-powered photodetectors [142,173,174];Portable photodetectors based on flexible electronics [175].

Table 12 collects the parameters of selected ultraviolet photodetectors in which the above-highlighted new concepts are implemented.

### 6.1. Low-Dimensional Solid Photodetectors

Among the various nanoparticles (NPs) enhanced with surface plasmons, silver (Ag) nanoparticles are one of the best for UV photodetectors due to their low parasitic absorption, high scattering efficiency, and ease of preparation. For example, Figure 55 shows the spectral responses of GaN MSM photodetectors without Ag, with as-deposited Ag, and with annealed Ag. The schematic structure of the surface plasmon-enhanced photodetector is also shown in this figure. The Ag nanoparticles were fabricated by electron beam evaporation and then annealed at 800 °C for 5 min. The high responsivity (enhancement more than 30 times) is attributed to the localized field of Ag NPs and the plasmonic scattering effect.

Ag NPs also significantly affect the performance enhancement of the ZnO-based UV detector [176]. To match the resonance of surface plasmons, optimizing the type of metal, its size, and its distribution are necessary. While Ag nanoparticles are used to enhance detector performance in the near-UV band, in the deep UV, Al nanoparticles have an advantage [177].

Up until now, the investigation of low-dimensional solid UV photodetectors is still in the prototype stage. However, it is predicted that nanotechnology could pave the way for exploring nanoscale smart, intelligent, and multifunctional photodetectors with novel optical and electronic properties. An example of such structures is shown in Figure 56—it is a visible-blind photodetector based on p–i–n junction GaN nanowire ensembles [178]. The nanowires were grown by plasma-assisted molecular beam epitaxy on an n-doped Si(111) substrate. Their density was ∼10^10^/cm^2^, giving an average substrate coverage of ∼75%, an average height is (0.9 ± 0.1) μm, and a diameter of (50 ± 20) nm. The spin coating technique filled the space between nanowires with spin-on-glass (hydrogen silsesquioxane—HSQ). Oxygen plasma and thermal annealing were used to transfer the HSQ into SiO_x_. Approximately 10^7^ nanowires create one mesa. An indium tin oxide (ITO) layer is applied to design Ti/Au contacts. The detector presents a peak responsivity of 0.47 A/W at a biased voltage of −1 V. The reverse leakage current at −1 V is 8 × 10^−7^ A. The UV-to-VIS rejection ratio was 2 × 10^2^, but the operation speed of this photodiode was slow (−3 dB cut-off frequency, ∼100 Hz).

For most III-V film UV photodetectors, the trade-off between responsivity and response speed is observed. To overcome this phenomenon, Zhao et al. reported the ZnO-Ga_2_O_3_ core-shell microwire photodetector shown in Figure 57a [179]. This photodetector has been synthesized following a one-step approach, in which the ZnO core and the Ga_2_O_3_ shell were both single-crystalline, with hexagonal and monoclinic crystals, respectively. In and Ti/Au are used to form ohmic contact with ZnO and Ga_2_O_3_, respectively. The current-voltage characteristics suggest that avalanche multiplication dominates the internal gain in the photodetector. The high performance of these microwire UV photodetectors in the spectral range from 200 to 280 nm is supported by the parameters: the current responsivity up to 1.3 × 10^3^ A/W under a −6 V bias (Figure 57b), detectivity as high as 9.91 × 10^14^ Jones, fast response with a rise time shorter than 20 μs, and a decay time of 42 μs.

A multifunctional photodetector with novel optical and electronic properties is shown in Figure 58. This figure presents the design of a three-band photodetector integrated by two different semiconductor nanostructures (a CdS nanowire (2.5 eV) and a ZnO nanowire (3.37 eV) tetrapod-granular) on a wide-bandgap diamond [179].

As mentioned previously, concepts of photodetectors made of 2D materials are also used in the ultraviolet range. An example of such a detector is the hybrid detector shown in Figure 59 [180]—its idea is described in Table 1 as a photo-FET. The vertical barrier heterostructure photodetectors (VBHPs) consist of a graphene bottom electrode, a MoS_2_ active (absorber) layer, and an h-BN energy barrier. In this hybrid detector (between the photodiode and photoconductor), photosensitization and carrier transport take place in separately optimized regions: MoS_2_ for efficient light absorption and the second (graphene) to provide fast charge reticulation (photogating effect).

The asymmetric barrier distribution at the MoS_2_/h-BN interface results in high-performance parameters with a responsivity of 416.2 mA/W at 360 nm. Figure 60 compares VBHP’s performance with commercial products. It can be seen that its responsivity is higher than that of other photodetectors.

### 6.2. Self-Powered Photodetectors

Most conventional ultraviolet UV photodetectors are biased by an external voltage, which requires an external power source. This forces a complex electronic circuit to power the detector, which increases the cost of the instrument and limits its use in special conditions. For these reasons, the design of self-powered UV photodetectors (without an external power source) has become an attractive research direction in photoelectric detection.

Self-powered UV photodetectors can be divided into three groups of power sources: semiconductor-based diodes (Schottky junctions, p–n junctions, and heterojunctions based on the photovoltaic effect), photochemical batteries, and devices with triboelectric nanogenerators [181]. Most popular self-powered photodetectors are fabricated using various junction-based materials such as ZnO [173,182], TiO_2_ [183], Ga_2_O_3_ [142], and perovskite [184]. A built-in electric field at the junctions separates the photogenerated electrons and holes, creating a photovoltaic voltage. Figure 61 shows examples of self-powered UV photovoltaic photodetectors. The first one shows a homojunction p-ZnO:(Li,N)/n-ZnO UV photodetector [185]. The next one is a self-powered core-shell nanowire heterojunction based on n-ZnO/p-NiO characterized by high spectral-selective characteristics and high response speed [186]. In this case, a NiO shell layer with a thickness of about 45 nm is deposited (using the chemical vapor deposition technique and magnetron sputtering technique) onto the highly oriented vertical ZnO nanowires with a diameter of about 30 nm. Figure 61c presents Au/β-Ga_2_O_3_ nanowire array Schottky junction [142], where the vertical Au/Ga_2_O_3_ nanowire Schottky junction was synthesized by thermal oxidation, and 20-nm Au was subsequently deposited to form a Au/Ga_2_O_3_ nanowire.

Figure 62 shows the characteristics of a self-powered UV photodetector based on n-ZnO/p-NiO core-shell heterojunction nanowire arrays. The rectification current-voltage behavior arises from the n-ZnO/p-NiO interface. A turn-on voltage of about 1.7 V is observed. The selective response spectrum of the photodetector under front illumination conditions at zero bias, with a peak centered at around 372 nm, is caused by the “filter” effect of the outer layer of the p-NiO shell layer.

At the present stage of technology, the performance of self-powered photodetectors is limited by the low quality of junctions, and photodetectors with higher performance are highly desired.

#### Flexible Photodetectors

Low-dimensional solids are also considered promising materials in the design of flexible photodetectors due to their large surface-to-volume ratio, excellent mechanical strength, superior structural stability, low power dissipation, and synergistic heterojunction effect. Flexible UV photodetectors are finding increasing applications in portable and wearable optoelectronic systems, such as foldable displays, wearable electronics, and smart medical monitoring systems. Popular substrates that have found application in this field are polyethylene terephthalate (PET), polyimide (PI), PMMA, polyethylene naphthalate (PEN), poly(dimethylsiloxane) (PDMS), polyvinylidene fluoride (PVDF), etc. [187]

Due to its wide energy gap, high charge carrier mobility, and various deposition techniques under relatively low temperature conditions, among other miscellaneous materials, zinc oxide (ZnO) has a vast potential for constructing UV optoelectronic devices using flexible plastic substrates (3.37 eV). For instance, the sequence of ITO/ZnO UV photodetector fabrication on a flexible polyimide (PI) base surface is presented in Figure 63. This process is based on nanocrystalline ZnO films deposited by shadow mask deposition and RF magnetron sputtering.

Due to the piezo-photronic effect observed in ZnO, the displacement of ions induces piezoelectric polarization, which modulates the concentration of carriers, the barrier height, and the built-in electric field at the interface [189,190]. These phenomena are the basis of the operation of photodetectors made of ZnO nanometer structures; an example of such a structure and its principle of operation is shown in Figure 64. ZnO nanorods elongated along the c-axis are mechanically stable and have high conductivity. Their better performance as photodetectors, compared to conventional ZnO films, is due to the fact that the structure of the nanorods enhances light capture through reflection and absorption.

In recent years, lightweight and flexible UV photodetectors have become increasingly important in various applications. For this purpose, organic and inorganic nanocomposites are used as active layers with photodiode configurations. By controlling the microstructure and thickness of the active layers, the properties of the photodetectors can be controlled. Such a detector structure is shown in Figure 65. This flexible photodetector structure was fabricated on indium tin oxide-coated poly(ethylene terephthalate) substrates with a nanocomposite active layer composed of ZnO nanoparticles blended with a wide band gap conjugated polymer, poly[(9,9-dioctylfluorenyl-2,7-diyl)-alt-co-(bithiophene)] (F8T2) [191]. As is shown in Figure 65c, the device shows two narrow response peaks at 360 and 510 nm under reverse biases and exhibits greater than 100% external quantum efficiency, indicating a photomultiplication effect. The highest detectivity value was 8.8 × 10^11^ Jones. Moreover, the photodetectors showed good photoresponse stability after tensile and compressive bending.

## 7. Conclusions

Ultraviolet detectors have come a long way in their evolutionary development. The older detection techniques, elaborated in the interwar period between the first and second world wars, used photographic films, detectors based on gas photoionization, and photoemissive detectors. The complexity of their mechanics and biasing circuits makes fabrication difficult, and they do not allow for compact and robust structures. Photocathodes fabricated on alkali telluride and alkali antimony have long been used in photomultipliers (PMTs) and image intensifiers and still represent a niche market due to their high sensitivity in the ultraviolet range of the spectrum while strongly discarding sensitivity in other parts of the spectrum.

Today, older types of ultraviolet detectors are being replaced by solid-state photon detectors with higher sensitivity and greater accuracy of photometric information, with immediate availability. In addition, solid-state detectors, with their small size, ruggedness, and ease of use, are supporting the further development of low-light imaging systems for military and civilian applications, ensuring their dominant position in the near future.

Semiconductors with wide bandgaps have been used in photoemission detectors (e.g., as photocathode coatings in microchannel wafer detectors), but today they are mainly used in photodetectors, where charge formation and accumulation take place inside the material. Currently, among the various types of UV detectors, photovoltaic detectors are dominant. Table 13 compares their advantages and disadvantages. Among the various materials used in ultraviolet detectors, Si-based photodiodes have a superior rank regarding simplicity, high sensitivity, low costs, and integrated circuit compatibility. However, when operating in high-temperature environments, such photodetectors must often be cooled to reduce their leakage current and, consequently, the noise of the detection system. Additionally, an optical filter is usually needed to filter out visible and infrared light, which significantly increases the complexity and cost of photodetectors. To overcome these problems, wide bandgap semiconductors such as AlGaN, SiC, diamond-based, and Ga_2_O_3_ have been investigated for UV light detection. Compared to Si-based devices, the application of these materials has many advantages, such as a high breakthrough field, filter-less UV light detection, high radiation resistance, and high chemical and thermal stability.

Two imaging array fabrication technologies are being developed: monolithic with deposition of active material to create charge directly on the signal readout IC (ROIC) and hybrid, where the semiconductor array and silicon-based readout chip are developed and fabricated independently and then bonded together using indium flip-chip integration.

Further optimization of UV photodetector performance is possible, allowing the design and fabrication of devices with better performance than conventional bulk devices. Currently, new UV photodetector concepts are inspired by new device architectures based on low-dimensional solid materials, self-powered photodetectors, and plasmonic amplification.

In comparison with standard semiconductors, the random stacking and fabrication of 2D material devices is a tremendous advantage due to the absence of lattice matching. A weak stack of atomic planes stacked on top of each other, held together by van der Waals (vdW) forces, leaves no dangling bonds, facilitating the fabrication of vertical heterostructures and the integration of 2D materials into silicon chips. In addition, vdW interactions allow large-scale growth of 2D materials regardless of substrates.

Further research will target three priority areas: materials, detector architectures, and applications. It is believed that future work in these directions will be concerned with creating semiconductor nanostructures in a more controlled, predictable, and simpler manner. In addition, it is assumed that smart, portable, and multifunctional concepts will derive more types of UV photodetectors and fulfill multifunctional tasks to meet complex expectations.

## Figures and Tables

**Figure 1 sensors-23-04452-f001:**
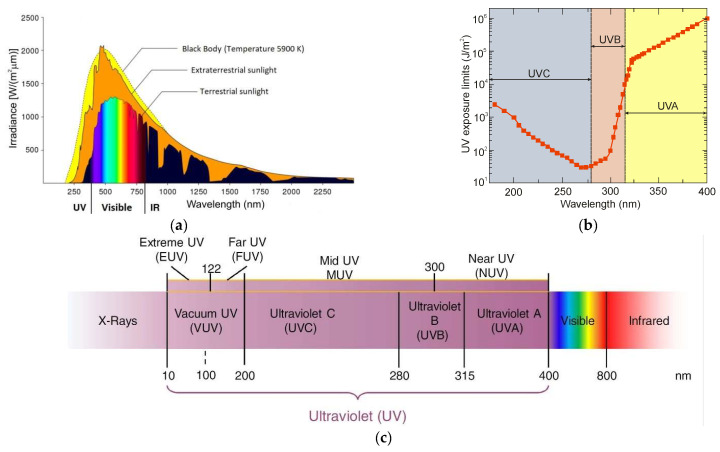
Solar irradiation spectrum: (**a**) threes spectra [the spectrum of a black body with a temperature of 5900 K similar to the solar spectrum, the actual spectrum of the sun at the outer edge of the earth’s atmosphere (extra-terrestrial solar radiation), and the spectrum at sea level (terrestrial solar radiation), (**b**) human exposure limits as a function of wavelength in the UV spectrum, and (**c**) the UV spectral region and its subdivisions.

**Figure 2 sensors-23-04452-f002:**
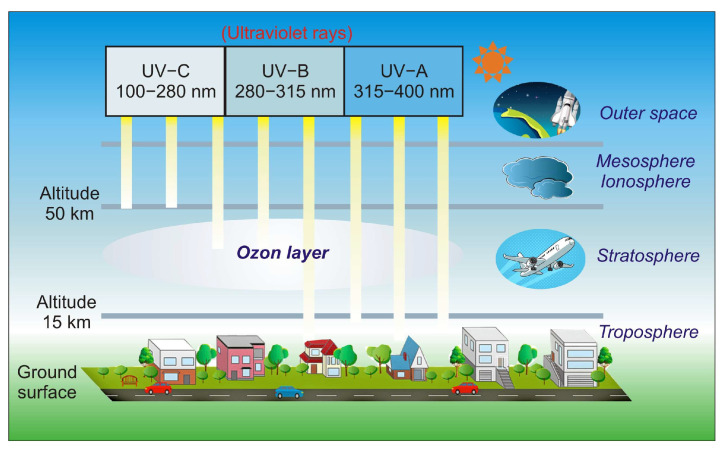
Blocking different bands of UV radiation.

**Figure 3 sensors-23-04452-f003:**
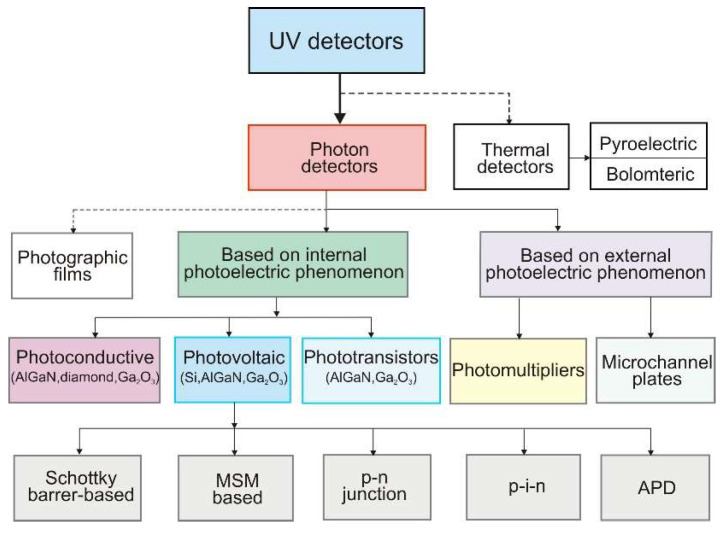
Classification of UV detectors.

**Figure 4 sensors-23-04452-f004:**
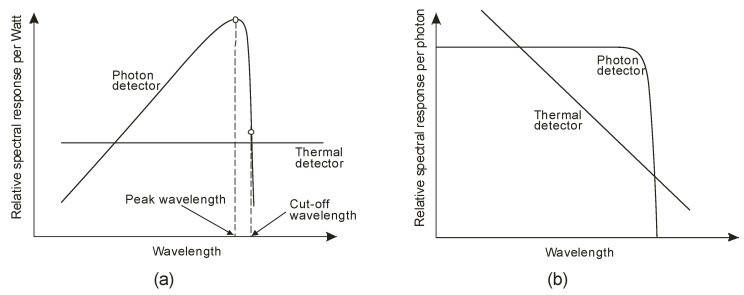
Relative spectral response for photon and thermal detectors analyzing (**a**) radiation power and (**b**) photon flux.

**Figure 5 sensors-23-04452-f005:**
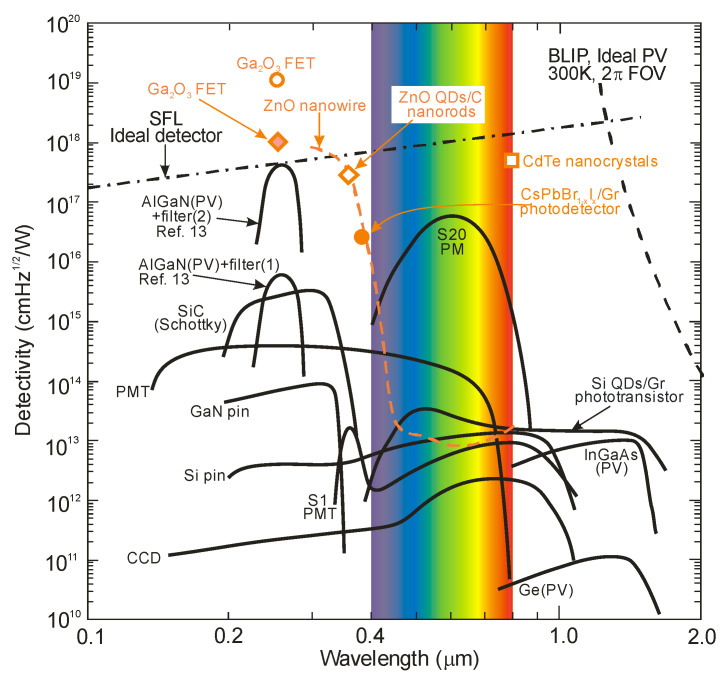
Spectral dependence of detectivity for photodetectors operating at room temperature in the wavelength range of 0.1–2 µm. Designations adopted: PC—photoconductive detector; PV—photovoltaic detector; and PM—photomultiplier. Additional data for AlGaN photodiodes and other various types of photodetectors (pink color) are from the literature.

**Figure 6 sensors-23-04452-f006:**
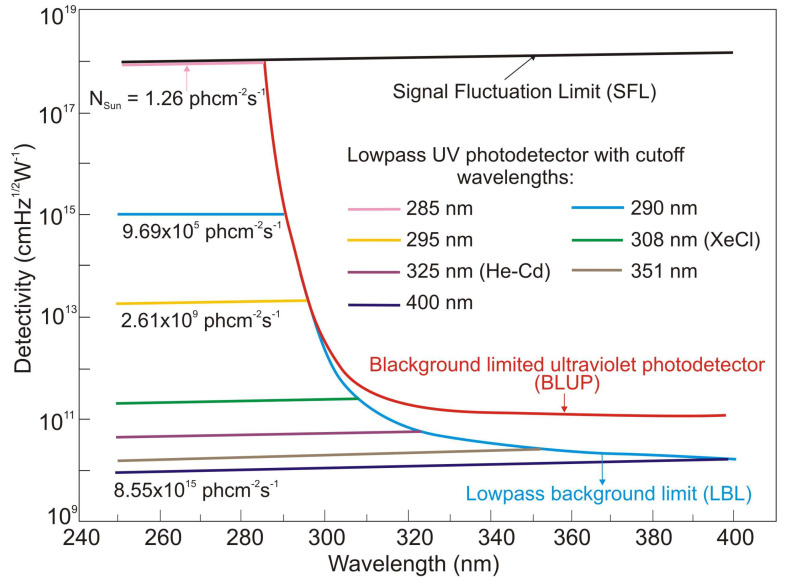
The spectral performance of the UV photodetectors. Two cut-off wavelengths correspond to the commercial lasers, as indicated in brackets (He-Cd and XeCl). Adapted with permission from Ref. [13].

**Figure 7 sensors-23-04452-f007:**
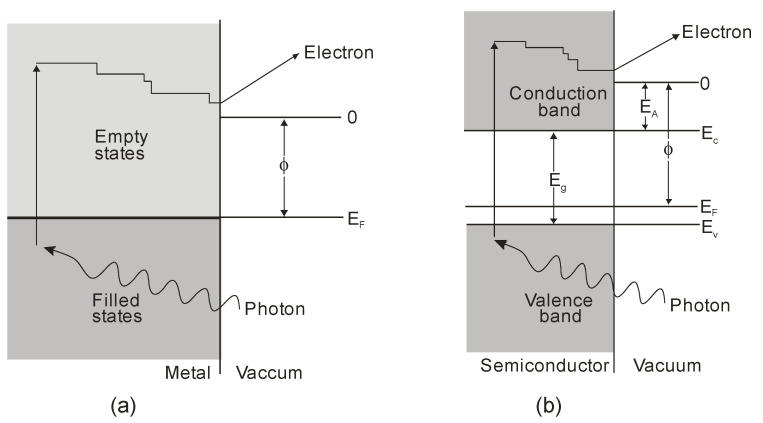
Diagrams of the energy band illustrating the photoemissive effect: (**a**) in a metal, and (**b**) in a semiconductor.

**Figure 8 sensors-23-04452-f008:**
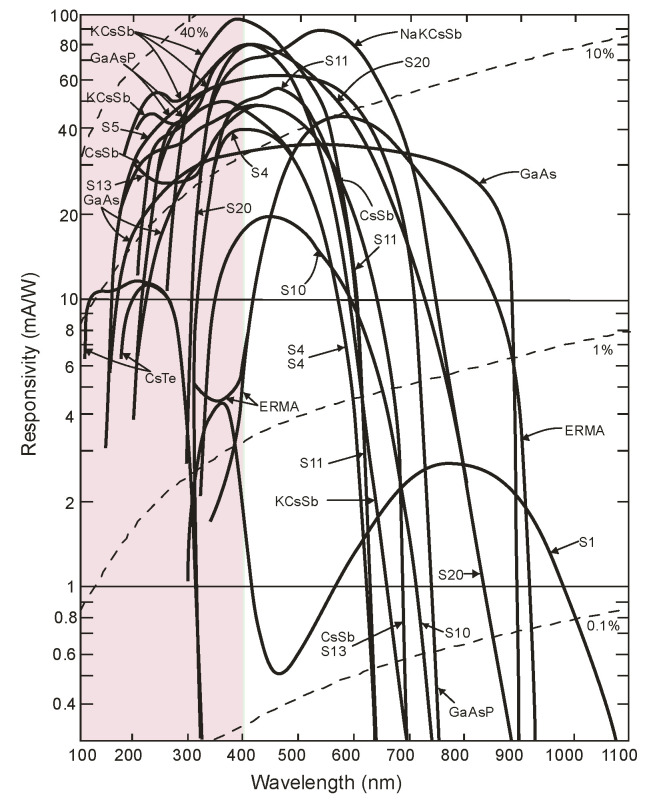
Typical spectral characteristics of photocathodes with a quantum efficiency indicated by dotted lines. Chemical formulas are abbreviated to save space. S1 = AgOCs with lime or borosilicate crown-glass window; S4 = Cs_3_Sb with lime or borosilicate crown-glass window (opaque photocathode); S5 = Cs_3_Sb with ultraviolet-transmitting glass window; S8 = Cs_3_Bi with lime or borosilicate crown-glass window; S10 = AgBiOCs with lime or borosilicate crown-glass window; S11 = Cs_3_Sb with fused-silica window (semi-transparent photocathode); S13 = Cs_3_Sb with fused-silica window (semi-transparent photocathode); S19 = Cs_3_Sb with fused-silica window (opaque semi-cathode); S20 = Na_2_KCsSb with lime or borosilicate glass window. ERMA = extended red multialkali.

**Figure 9 sensors-23-04452-f009:**
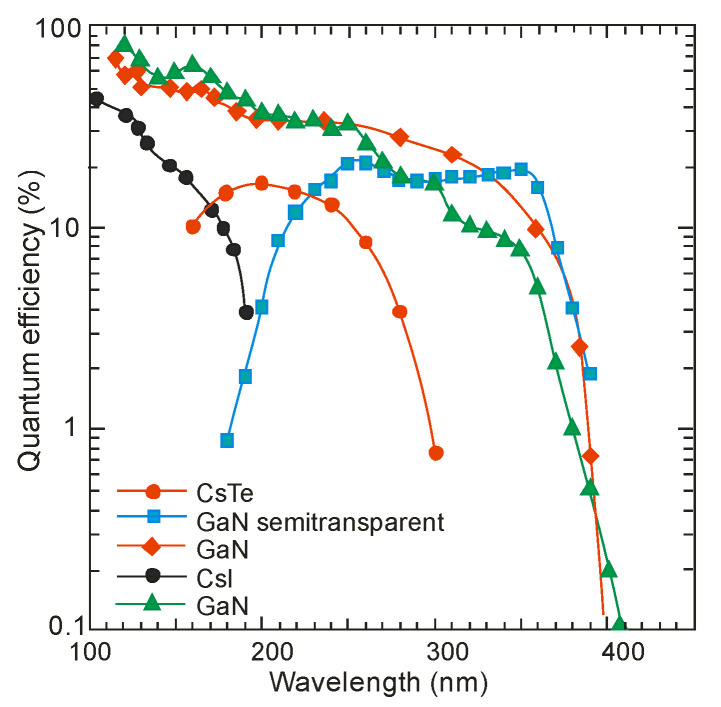
Quantum efficiencies as a function of wavelength for several photocathodes. Reprinted with permission from Ref. [17].

**Figure 10 sensors-23-04452-f010:**
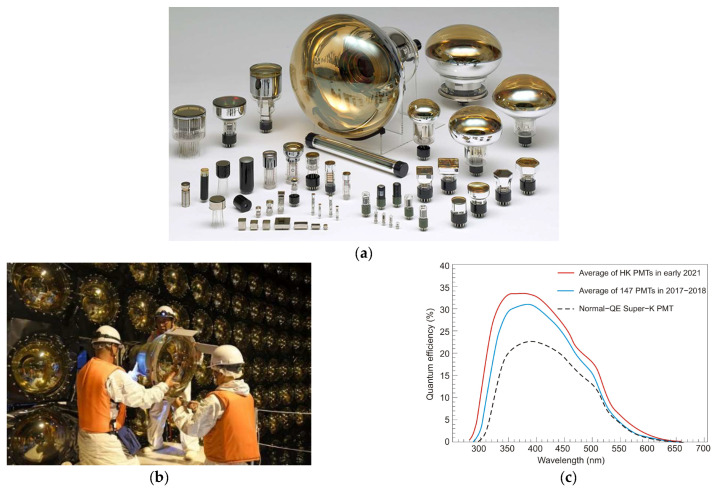
Photomultiplier tubes: a representative selection of different PMT types fabricated by Hamamatsu (**a**); the custom-designed large 20-in-diameter PMTs (the largest in this figure) are fabricated for use in the neutrino SuperKamiokande observatory near Hida, Japan (during their installation—see (**b**)); the SuperKamiokande facility employs 11,200 PMTs paving the walls of a 40-m diameter tank of water that probes the neutrinos. The spectral quantum efficiency characteristics of the PMTs installed in SuperKamiokande are shown in (**c**). Reprinted with permission from Ref. [20].

**Figure 11 sensors-23-04452-f011:**
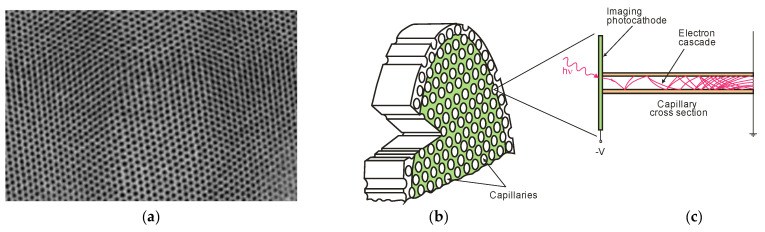
Schematic presentation of a microchannel plate: (**a**) SEM view of a microtube set, reprinted with permission from Ref. [21], (**b**) cutaway view, and (**c**) a single capillary.

**Figure 12 sensors-23-04452-f012:**
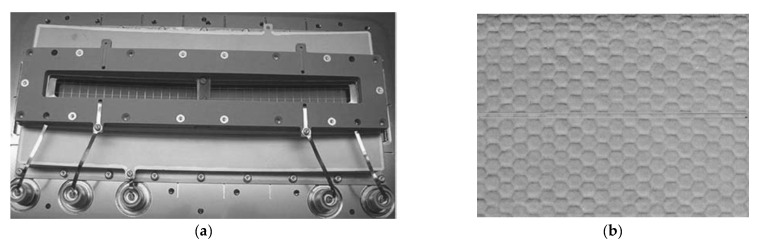
UV detector installed in the Cosmic Origins Spectrograph, reprinted with permission from Ref. [21]: (**a**) two MCP detectors consisting of two 85 × 10 mm segments; (**b**) an image of an active area of the MCP (10 × 13 mm).

**Figure 13 sensors-23-04452-f013:**
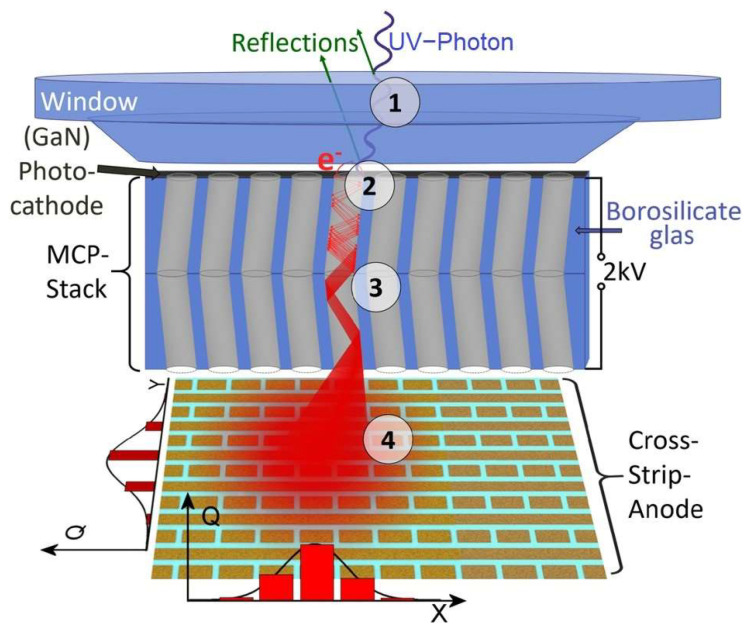
Principle of operation of the UV-MCP detector. Reprinted with permission from Ref. [22]. Where 1 indicates the window; 2—the photocathode; 3—the MCP–stack; 4—the cross-ctrip anode.

**Figure 14 sensors-23-04452-f014:**
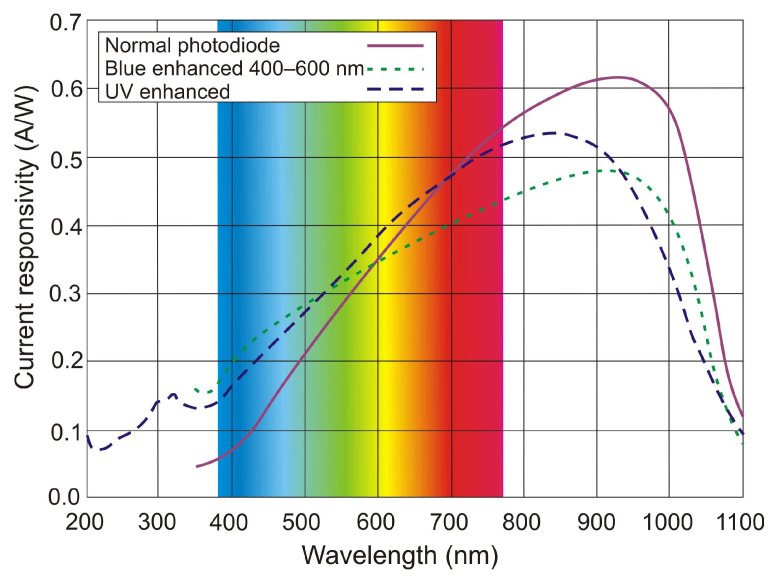
Current responsivity of some planar diffused silicon photodiodes.

**Figure 15 sensors-23-04452-f015:**
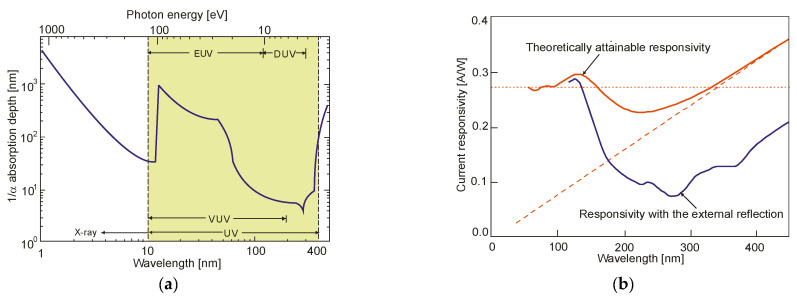
Penetration depth in Si vs. incident radiation wavelength/photon energy (**a**) and spectral responsivity of silicon-based photodiodes (**b**). Reprinted with permission from Ref. [23].

**Figure 16 sensors-23-04452-f016:**
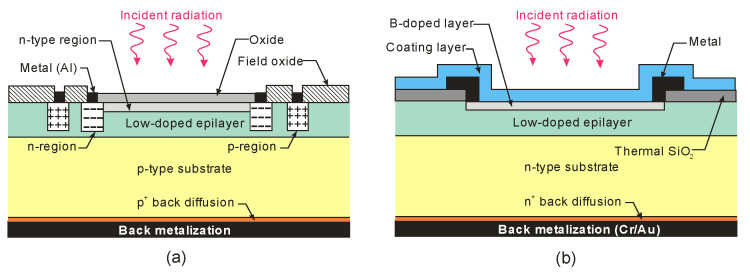
Vertical cross sections of n-on-p (**a**) and p-on-n (**b**) Si-based photodiodes.

**Figure 17 sensors-23-04452-f017:**
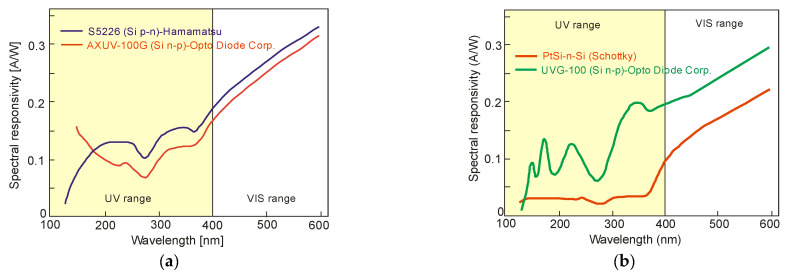
Spectral responsivity: (**a**) of S5226 and AXUV silicon photodiodes and (**b**) other commercially available UV photodiodes. Reprinted with permission from Ref. [24].

**Figure 18 sensors-23-04452-f018:**
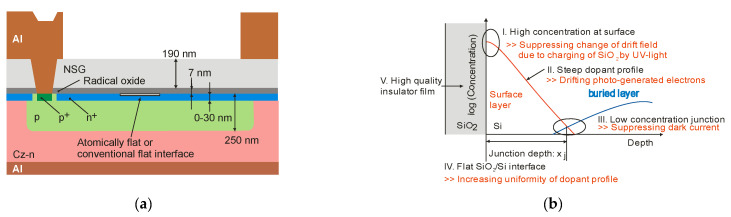
The developed Si PD technology for UV/VIS/NIR imaging: (**a**) the cross-sectional view of the n^+^-p-n photodiode; (**b**) key features to achieve high sensitivity and high stability in the UV photodiode. Reprinted with permission from Ref. [26].

**Figure 20 sensors-23-04452-f020:**
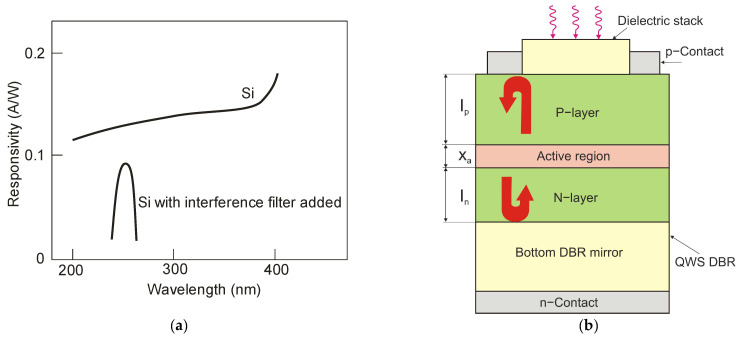
Spectra matching of UV photoreceivers: (**a**) responsivity plot of a silicon photodiode with and without filter, and (**b**) resonant cavity photodiode.

**Figure 21 sensors-23-04452-f021:**
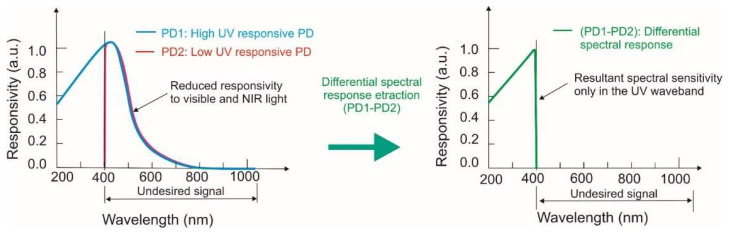
Conceptual diagram of a UV sensor with a differential spectral response. Reprinted with permission from Ref. [34].

**Figure 22 sensors-23-04452-f022:**
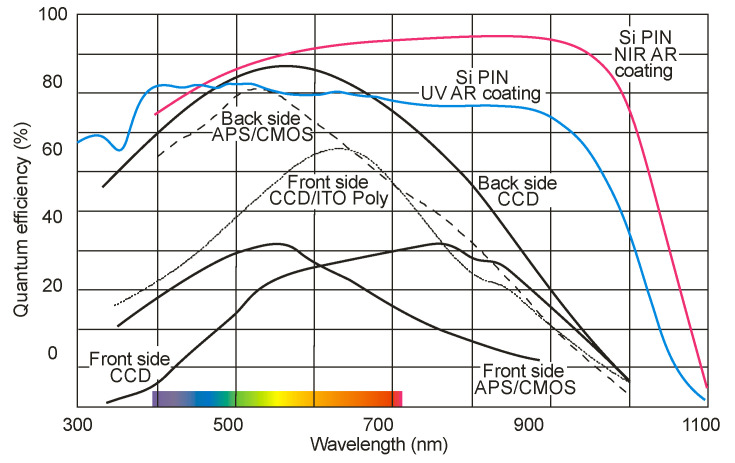
Comparison of the quantum efficiency of different silicon photodetector technologies.

**Figure 23 sensors-23-04452-f023:**
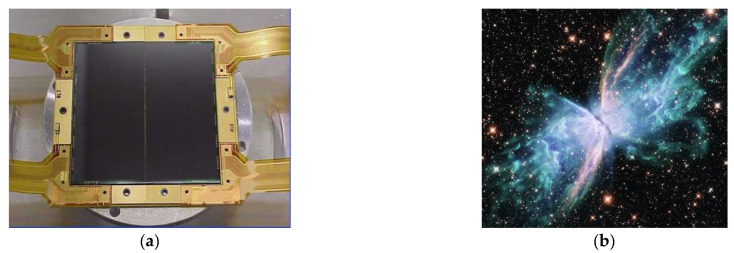
The CCD array developed by e2v (formerly Marconi), reprinted with permission from Ref. [35]: (**a**) mounted on its TEC cooler; (**b**) an emerging WFC3 butterfly snapped from a stellar demise in the planetary Nebula NG C 6302, more commonly known as the Bug Nebula or the Butterfly Nebula.

**Figure 24 sensors-23-04452-f024:**
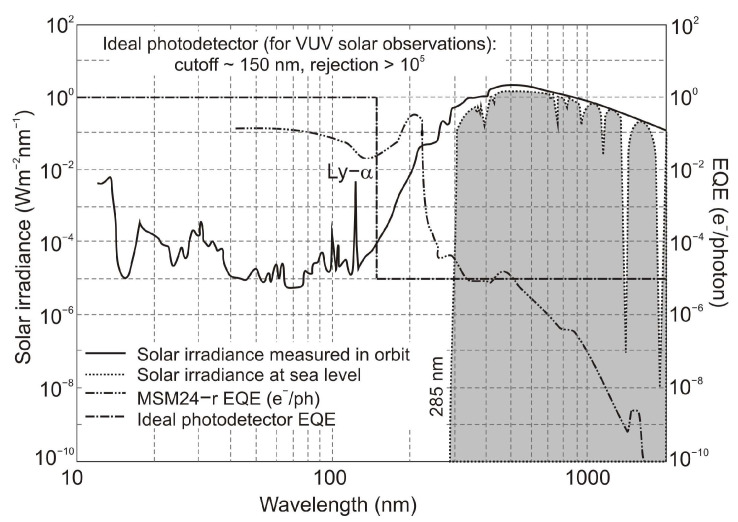
The solar spectral irradiance measured in space and from the Earth (left ordinate axis) compared with the quantum efficiency (right ordinate axis) of an ideal wide bandgap detector for solar observations and of a real diamond MSM detector. Reprinted with permission from Ref. [36].

**Figure 25 sensors-23-04452-f025:**
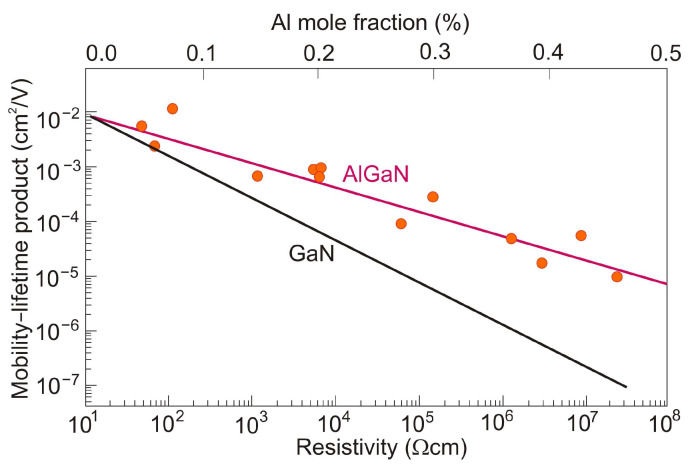
Mobility-lifetime product of Al_x_Ga_1−x_N photodetectors as a function of the film resistivity. The top axis shows the corresponding AlN mole fraction for the Al_x_Ga_1−x_N alloys. Reprinted with permission from Ref. [37].

**Figure 26 sensors-23-04452-f026:**
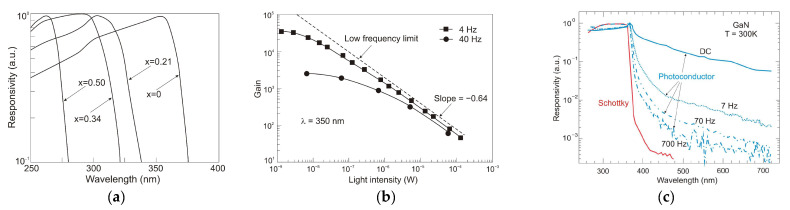
Al_x_Ga_1−x_N photodetectors: (**a**) normalized photoresponse of photoconductors at x = 0, 0.21, 0.34, and 0.50; (**b**) change in GaN photoconductor gain with a light intensity at 350 nm and (**c**) normalized spectral response of the GaN photoconductor measured at different chopping frequencies (the spectrum of a Schottky diode on the same sample is also shown for comparison).

**Figure 27 sensors-23-04452-f027:**
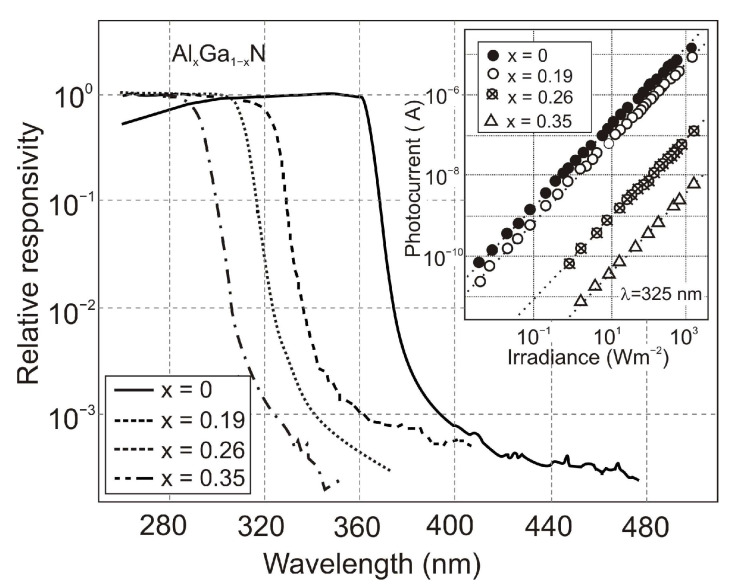
Spectral response of n-Al_x_Ga_1−x_N/Au Schottky photodiodes as a function of Al mole fraction (room temperature). In the inset, photocurrent vs. irradiance at 325 nm shows their linear behavior. Reprinted with permission from Ref. [50].

**Figure 28 sensors-23-04452-f028:**
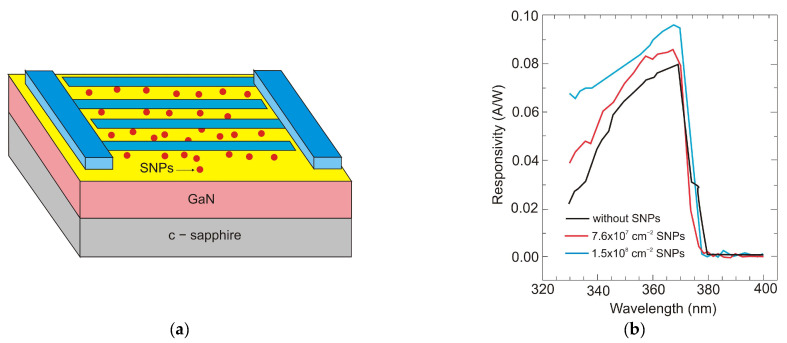
GaN MSM photodetector with deposited SiO_2_ nanoparticles: (**a**) schematic device structure and (**b**) spectral responsivities with different SiO_2_ nanoparticle densities. Reprinted with permission from Ref. [59].

**Figure 29 sensors-23-04452-f029:**
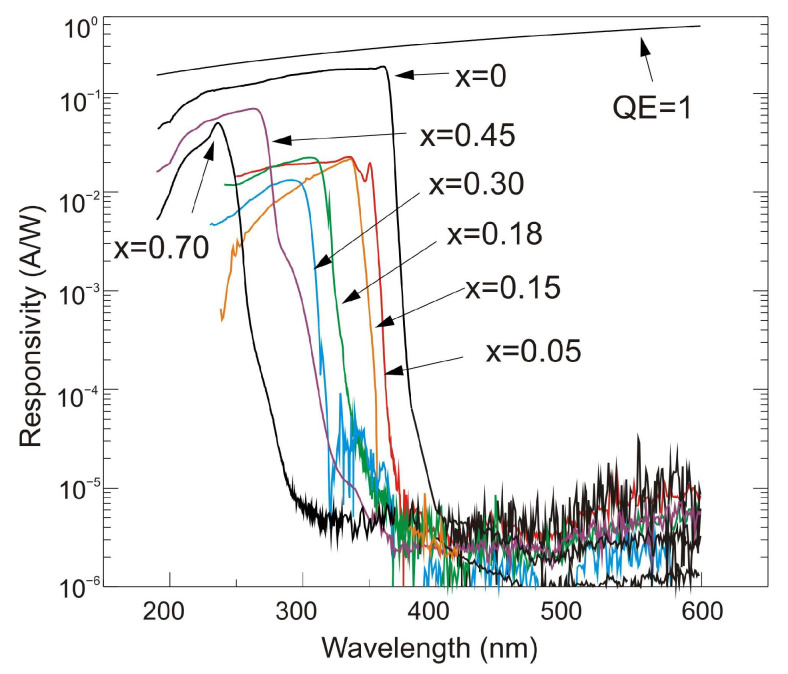
Responsivity of p-i-n Al_x_Ga_1−x_N photodiodes showing a cut-off wavelength continuously tuneable from 227 nm to 365 nm, corresponding to an Al concentration in the range of 0–70%. Reprinted with permission from Ref. [60].

**Figure 31 sensors-23-04452-f031:**
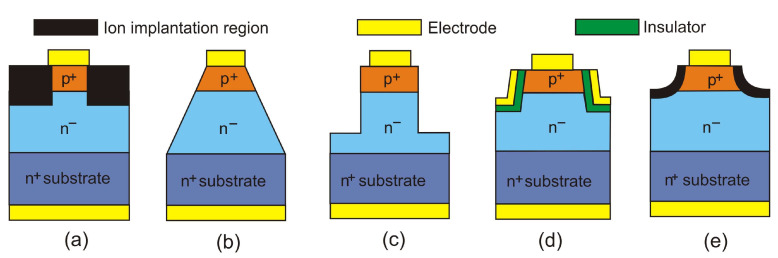
Edge terminations that allow GaN diodes to operate with avalanche capability: (**a**) ion-implanted edge termination; (**b**) bevel-etched edge termination); (**c**) deep-etched edge termination; (**d**) field-plate edge termination; and (**e**) moat-etched edge termination with ion-implantation compensation. Reprinted with permission from Ref. [62].

**Figure 32 sensors-23-04452-f032:**
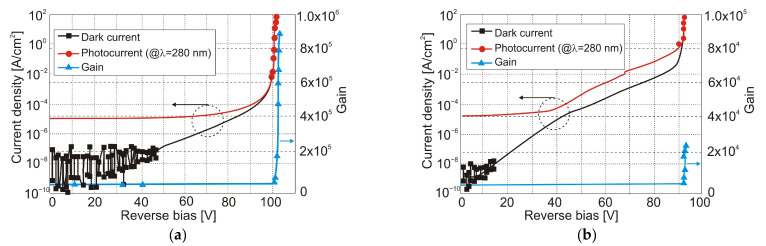
Reverse-biased *I-V* characteristics for 40-µm-diameter Al_0.05_Ga_0.95_N p-i-n UV-APD grown on (**a**) a free-standing GaN substrate and on (**b**) a GaN/sapphire template. Reprinted with permission from Ref. [68].

**Figure 33 sensors-23-04452-f033:**
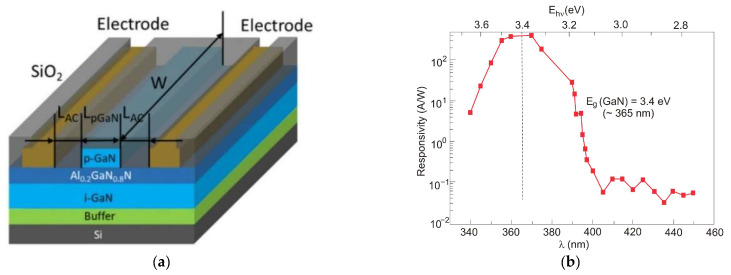
The p-GaN/AlGaN/GaN photodetector: (**a**) the schematic structure and (**b**) the wavelength-dependent responsivity of the photodetector, where the gate length *L_p_*_-GaN_ is 5 µm, the distance between the electrode edge and the p-GaN edge *L_AC_* is 1.5 µm, and the width *W* is 100 µm. Reprinted with permission from Ref. [46].

**Figure 34 sensors-23-04452-f034:**
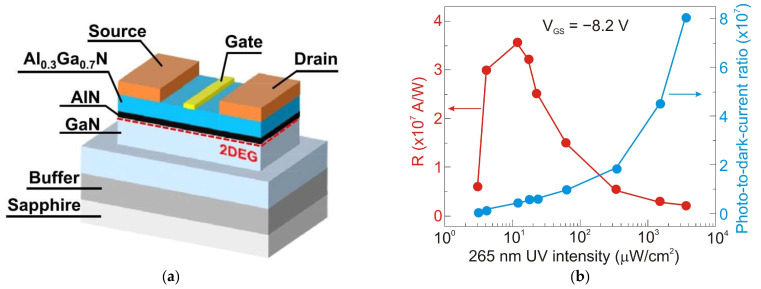
AlGaN/GaN phototransistor: (**a**) schematic structure; (**b**) current responsivity and photo-to-dark-current ratio versus power intensity from incident 265 nm illumination. Reprinted with permission from Ref. [77].

**Figure 35 sensors-23-04452-f035:**
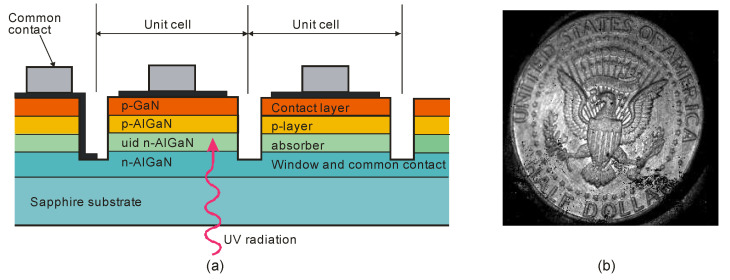
Hybrid 256 × 256 AlGaN p-i-n photodiode FPA with 30 × 30 µm unit pixels: (**a**) cross section of the back-illuminated photodiode array, and (**b**) UV reflection image of a U.S. half-dollar coin taken with this array. Reprinted with permission from Ref. [80].

**Figure 36 sensors-23-04452-f036:**
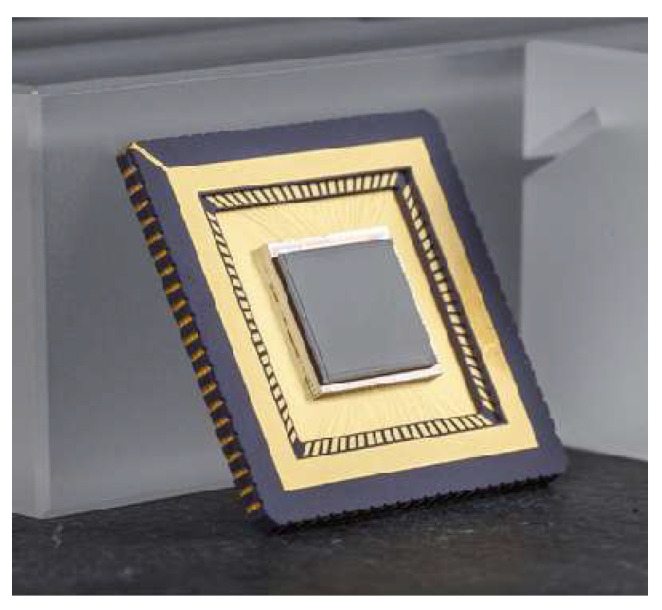
Solar-blind 640 × 512-µm pitch AlGaN FPAs mounted in an 84-pin chip carrier. Reprinted with permission from Ref. [91].

**Figure 37 sensors-23-04452-f037:**
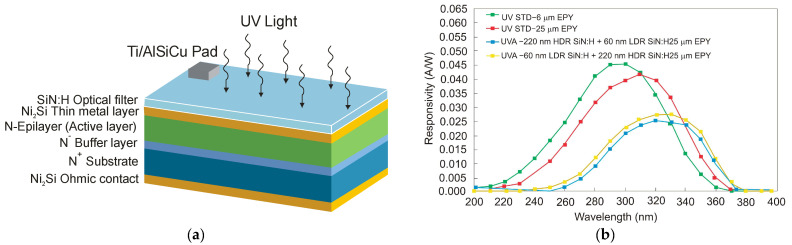
(**a**) View of the UV-A 4H-SiC detector schematic section; and (**b**) responsivity curves obtained at 25 °C and 0 V bias, where STD is the standard UV broadband detector, HDR and LDR are the high deposition rate and low deposition rate of SiN:H films, respectively. Reprinted with permission from Ref. [96].

**Figure 38 sensors-23-04452-f038:**
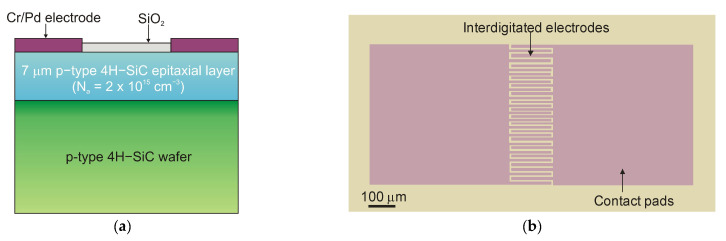
View of (**a**) schematic and (**b**) optical images of 4H-SiC MSM PDs. Reprinted with permission from Ref. [101].

**Figure 39 sensors-23-04452-f039:**
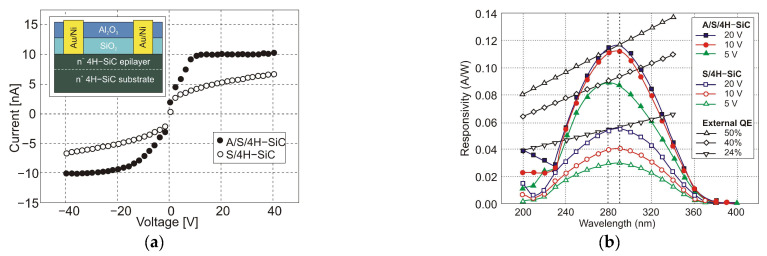
View of (**a**) *I*-*V* characteristics of Al_2_O_3_/SiO_2_/4H-SiC and SiO_2_/4H-SiC MSM photodetectors and (**b**) spectral response and external quantum efficiency (QE) of these photodetectors under bias voltages from 5 to 20 V. Reprinted with permission from Ref. [102].

**Figure 40 sensors-23-04452-f040:**
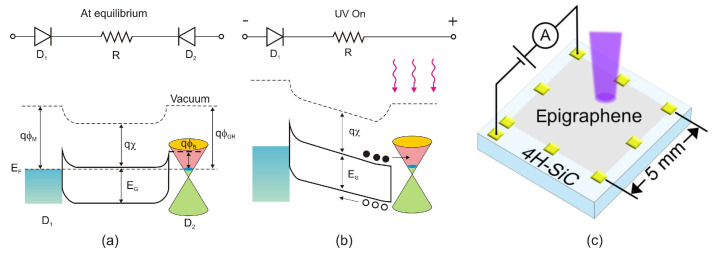
The energy band diagrams of semi-insulating 4H-SiC in contact with metal and graphene: (**a**) at equilibrium, (**b**) upon illumination, and (**c**) schematic of the device, where *ϕ_B_* is the high graphene barrier and *ϕ_GR_* is the work function of graphene. Reprinted with permission from Ref. [104].

**Figure 41 sensors-23-04452-f041:**
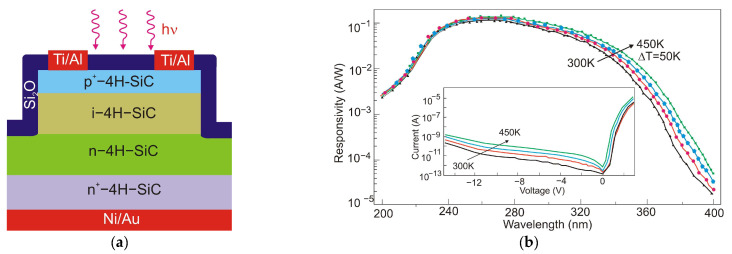
View of (**a**) the chip cross-section and (**b**) the photoresponsivity of the 4H-SiC p-i-n photodiode. The corresponding dark *I-V* characteristic is shown in the inset of (**b**). Reprinted with permission from Ref. [105].

**Figure 42 sensors-23-04452-f042:**
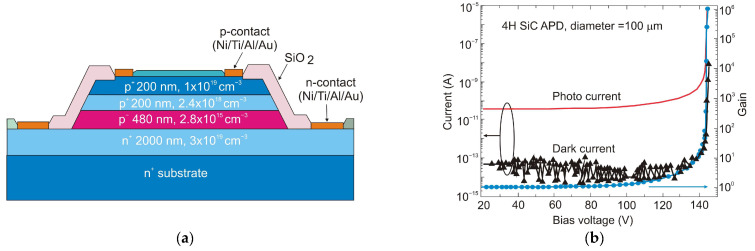
4H-SiC APDs: (**a**) schematic cross-section and (**b**) reverse current-voltage and gain-voltage characteristics of a 100-µm diameter APD at room temperature. Reprinted with permission from Ref. [113].

**Figure 43 sensors-23-04452-f043:**
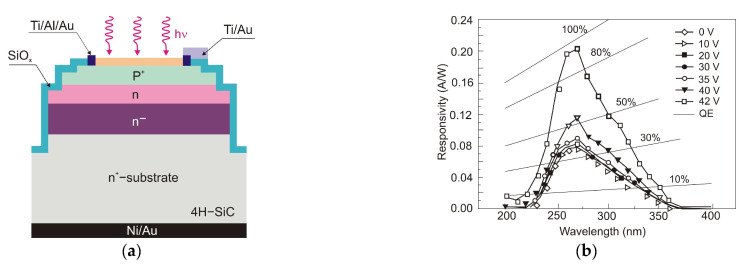
SAM 4H-SiC ultraviolet APD: (**a**) schematic cross-section and (**b**) spectral responsivities measured under various biases. Reprinted with permission from Ref. [115].

**Figure 44 sensors-23-04452-f044:**
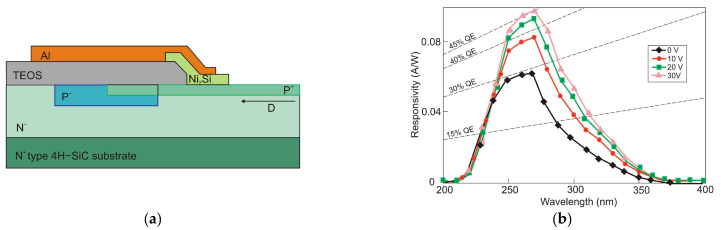
Planar 4H-SiC APD: (**a**) cross-section scheme and (**b**) responsivity spectra. Reprinted with permission from Ref. [116].

**Figure 45 sensors-23-04452-f045:**
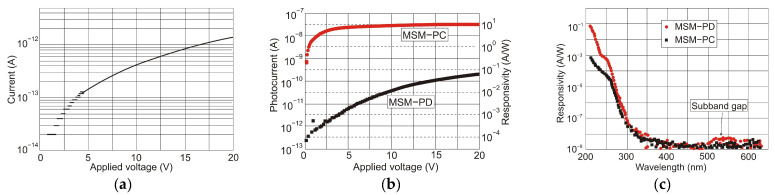
MSM diamond photoconductor characteristics: (**a**) dark current-voltage, (**b**) *I-V* characteristics of the photoconductor in comparison with a typical MSM-PD during a 220 nm light illumination, and (**c**) a spectral responsivity. Reprinted with permission from Ref. [127].

**Figure 46 sensors-23-04452-f046:**
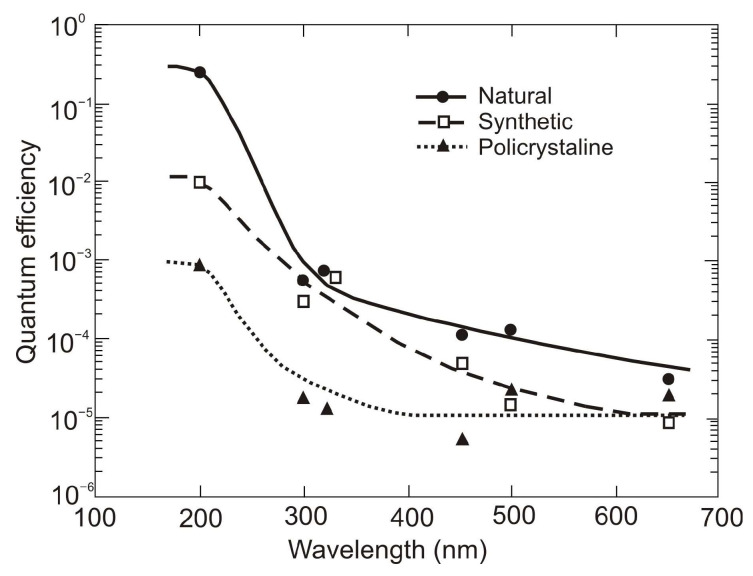
Quantum efficiencies of MSM detectors based on different types of diamond with a bias of 100 V. Reprinted with permission from Ref. [130].

**Figure 47 sensors-23-04452-f047:**
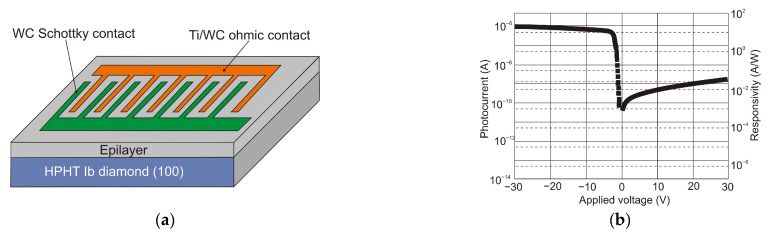
IDF-SPD based on a p-diamond epilayer: (**a**) schematic drawing, and (**b**) dependence of photocurrent and responsivity on the applied bias upon the 220 nm light illumination. Reprinted with permission from Ref. [135].

**Figure 48 sensors-23-04452-f048:**
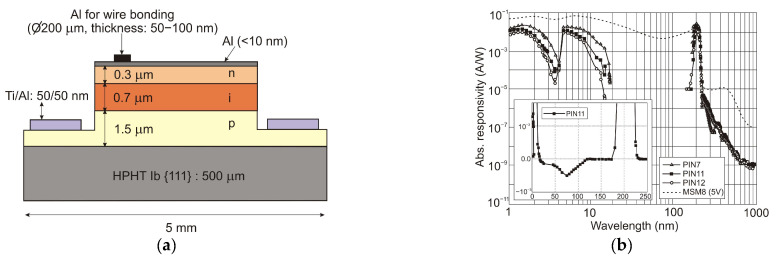
Diamond p-i-n photodiode: (**a**) schematic design and (**b**) absolute spectral responsivity at room temperature between 1 nm and 1000 nm. Reprinted with permission from Ref. [136].

**Figure 49 sensors-23-04452-f049:**
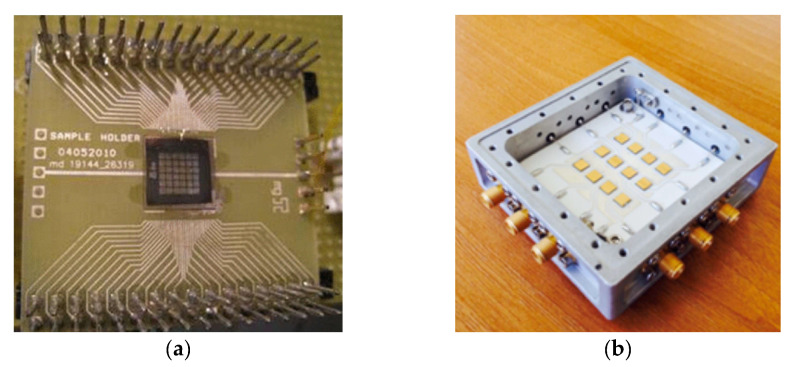
Mosaic diamond detectors demonstrated by Girolami et al.: (**a**) printed circuit board (reprinted with permission from Ref. [140]) and (**b**) integrated device (reprinted with permission from Ref. [141]).

**Figure 50 sensors-23-04452-f050:**
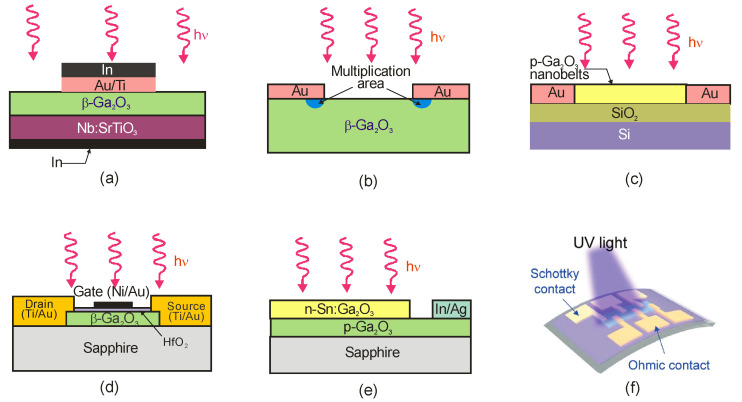
Schematic illustrations of different types of Ga_2_O_3_ ultraviolet photodetectors: (**a**) β-Ga_2_O_3_/Nb:SrTiO_3_ heterojunction, (**b**) Schottky barrier β-Ga_2_O_3_ photodiode and avalanche multiplication (effect of strong electric field beneath Au electrodes), (**c**) β-Ga_2_O_3_ nanobelts, (**d**) β-Ga_2_O_3_ phototransistor, (**e**) self-powered p–n junction, and (**f**) β-Ga_2_O_3_ nanomembrane-based flexible Schottky barrier photodiode.

**Figure 51 sensors-23-04452-f051:**
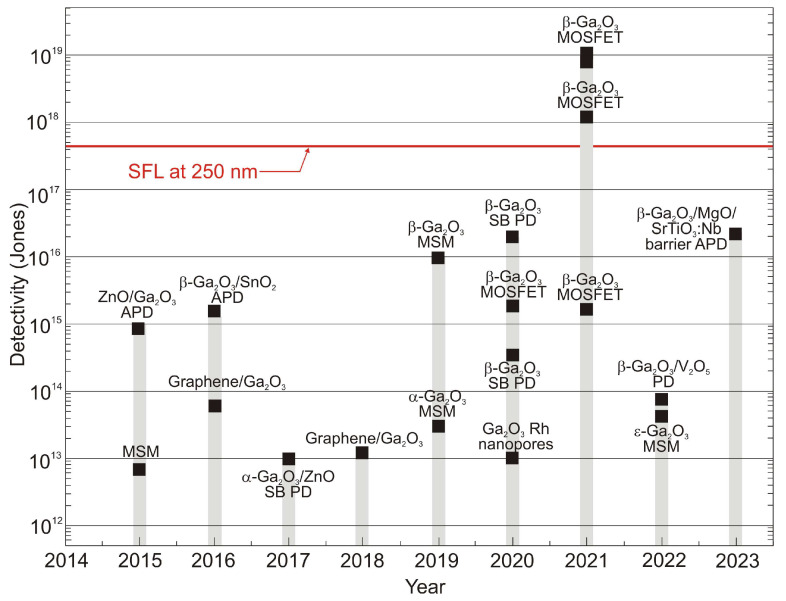
State-of-the-art detectivity in the last decade of Ga_2_O_3_-based photodetectors with various structures.

**Figure 52 sensors-23-04452-f052:**
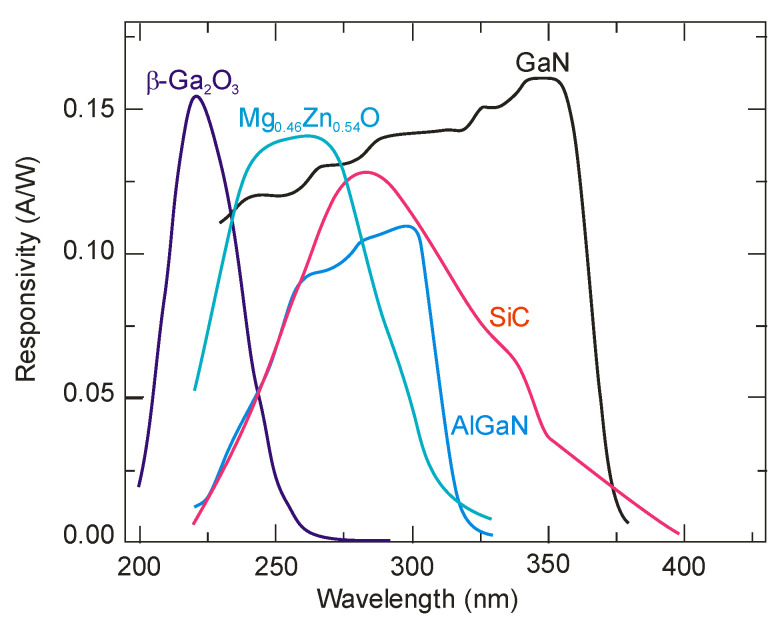
Comparison of the current responsivity of the β-Ga_2_O_3_ Schottky photodiode with commercial devices based on GaN, SiC, and AlGaN wide bandgap semiconductors. The data for GaN, AlGaN, and SiC photodetectors are taken from Boston Electronics Corporation spec sheets.

**Figure 53 sensors-23-04452-f053:**
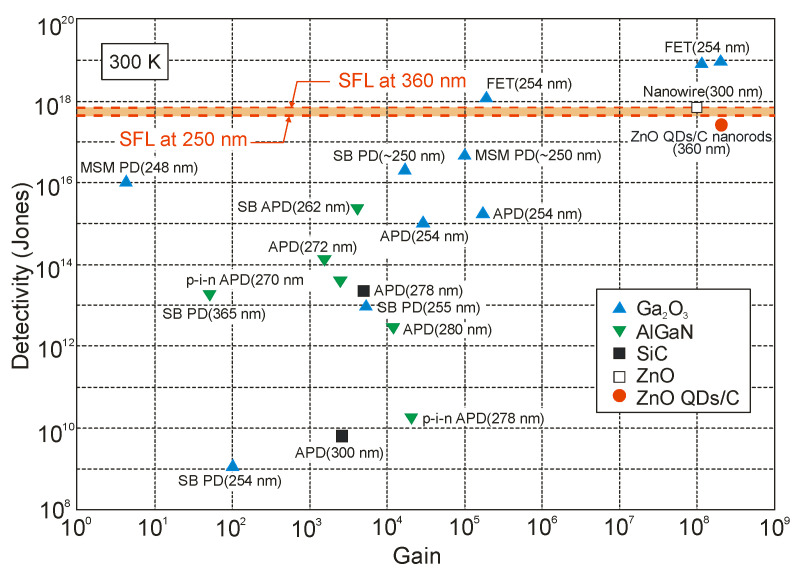
Plot of the dependence of detectivity on gain for different types of ultraviolet photodetectors at room temperature. The data were collected from the literature. Theoretical predictions for the SFL limit for the wavelength range of 250 nm to 360 nm are also highlighted. Designations: FET—field effect transistor; PD—photodiode; SB—Schottky barrier; MSM—metal-semiconductor-metal; APD—avalanche photodiode; QDs—quantum dots.

**Figure 54 sensors-23-04452-f054:**
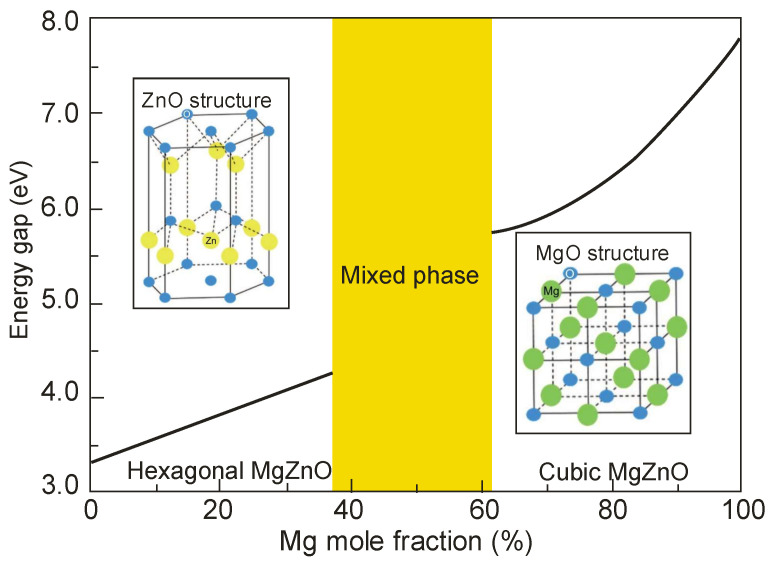
Energy band gap as a function of Mg content in the Mg_x_Zn_1−x_O ternary alloy.

**Figure 55 sensors-23-04452-f055:**
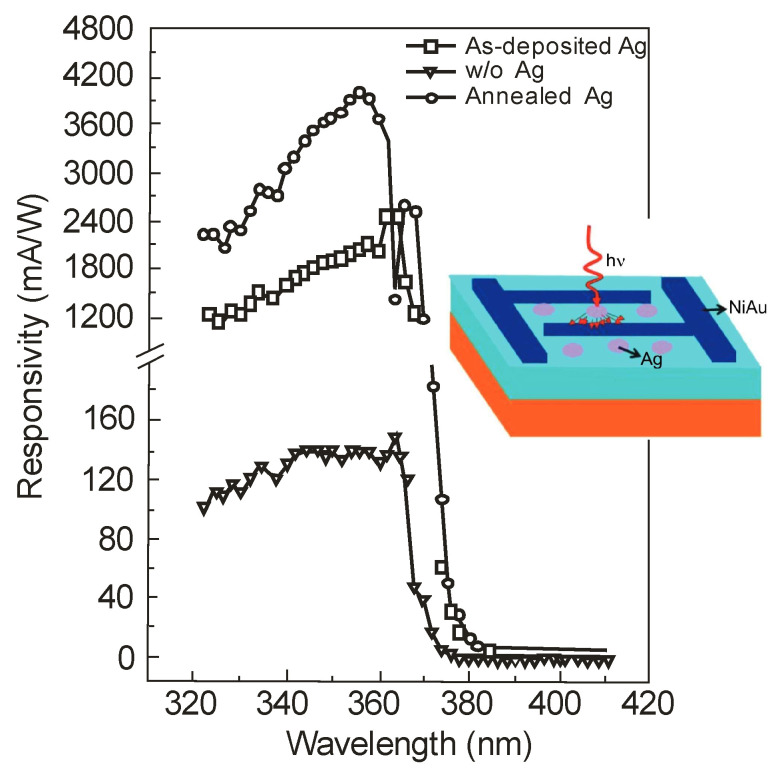
Spectral current responsivity of a surface plasmon-enhanced GaN MSM photodetector with and without Ag nanoparticles. The schematic structure of the detector is shown inside the figure. Reprinted with permission from Ref. [58].

**Figure 56 sensors-23-04452-f056:**
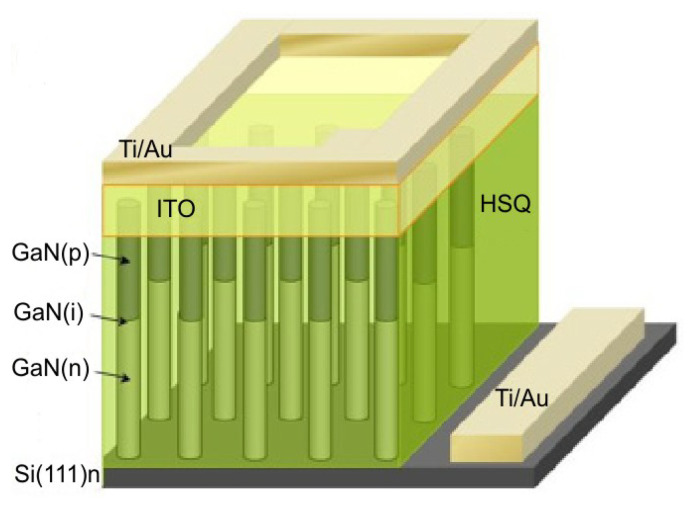
Schematic of a mesa p–i–n junction GaN nanowire ensemble. Reprinted with permission from Ref. [178].

**Figure 57 sensors-23-04452-f057:**
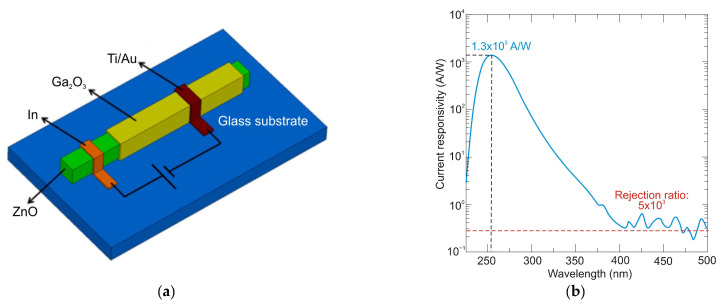
ZnO–Ga_2_O_3_ core-shell microwire avalanche photodetector: (**a**) schematic diagram and (**b**) spectral response of the device at −6 V bias. Reprinted with permission from Ref. [147].

**Figure 58 sensors-23-04452-f058:**
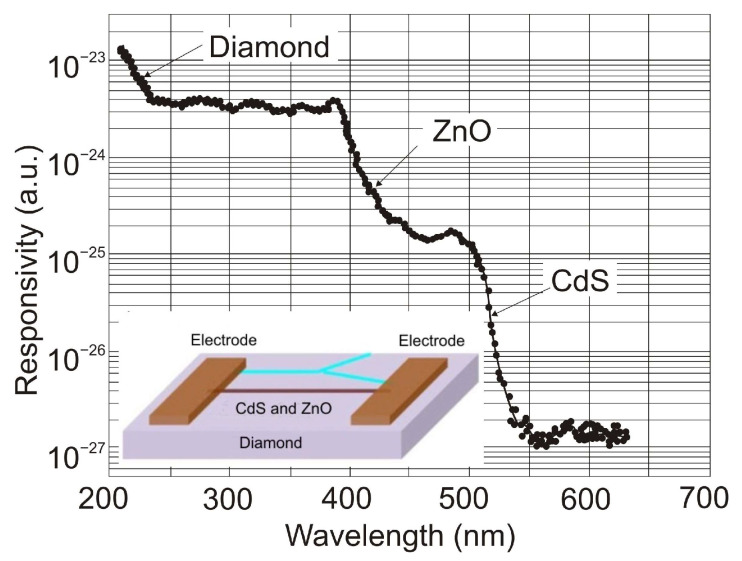
Spectral response of a three-band UV photodetector (operated in visible light, UVA light) fabricated by integrating ZnO sub-microrods and CdS nanowires on a diamond layer. The photodetector’s schematic design is shown in the figure. Reprinted with permission from Ref. [179].

**Figure 59 sensors-23-04452-f059:**
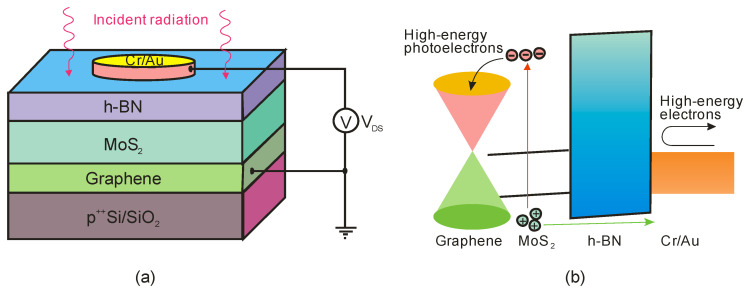
Hybrid UV photodetector: (**a**) schematic device structure and (**b**) energy band.

**Figure 60 sensors-23-04452-f060:**
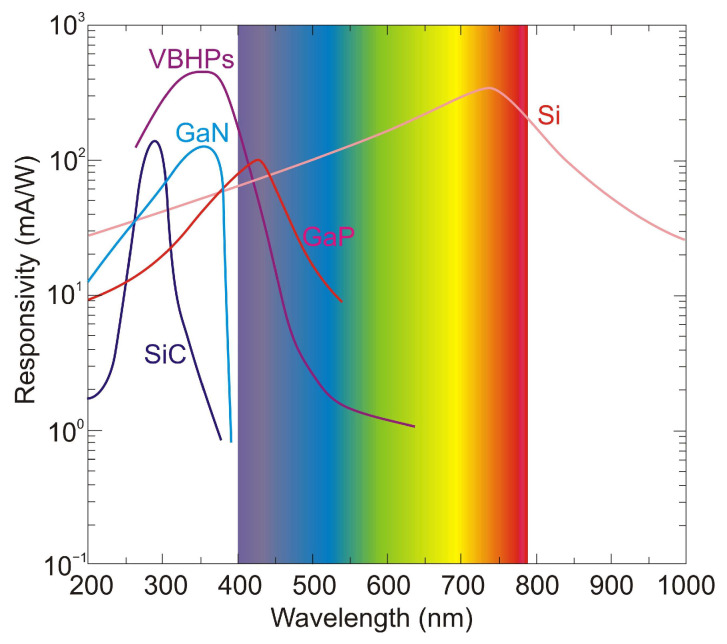
Spectral responsivities of VBHP hybrid photodetector and commercial photodetectors [Si-PD (PDA10A2, Thorlabs), GaP-PD (PDA25K2, Thorlabs), SiC-PD (18ISO90, Boston Electronics), and GaN-PD (G365S01S, Hefei Photosensitive Semiconductor)]. Reprinted with permission from Ref. [180].

**Figure 61 sensors-23-04452-f061:**
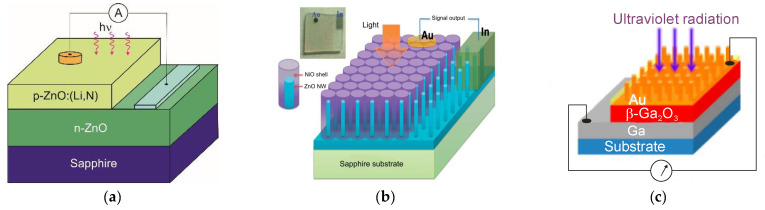
Self-powered photovoltaic ultraviolet photodetectors with different working mechanisms: (**a**) homojunction p-ZnO: (Li,N)/n-ZnO UV photodetector (reprinted with permission from Ref. [185]), (**b**) n-ZnO/p-NiO core-shell nanowire arrays (reprinted with permission from Ref. [186]), (**c**) vertical Au/β-Ga_2_O_3_ nanowire array Schottky junction photodetector (reprinted with permission from Ref. [142]).

**Figure 62 sensors-23-04452-f062:**
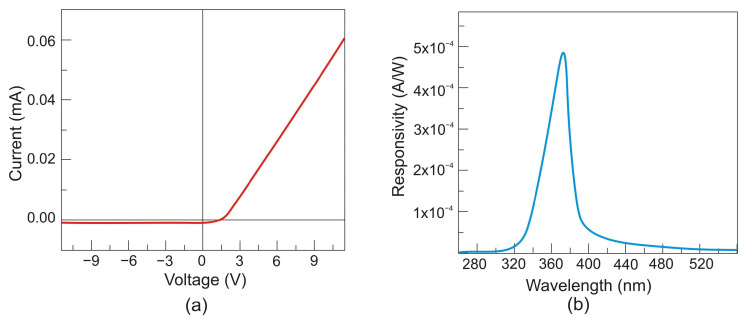
n-ZnO/p-NiO core-shell nanowire arrays (reprinted with permission from Ref. [186]): (**a**) I–V characteristics under dark conditions, and (**b**) current responsivity under front illumination conditions at zero bias.

**Figure 63 sensors-23-04452-f063:**
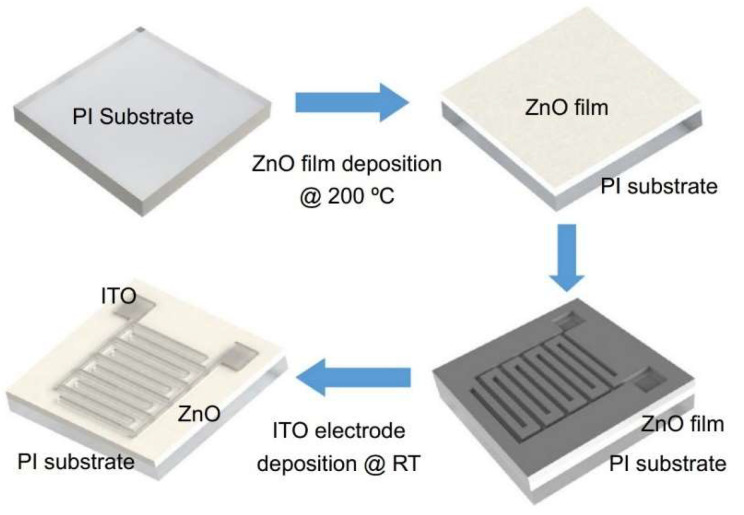
Fabrication process of an ITO/ZnO UV photodetector on a flexible PI substrate. Reprinted with permission from Ref. [188].

**Figure 64 sensors-23-04452-f064:**
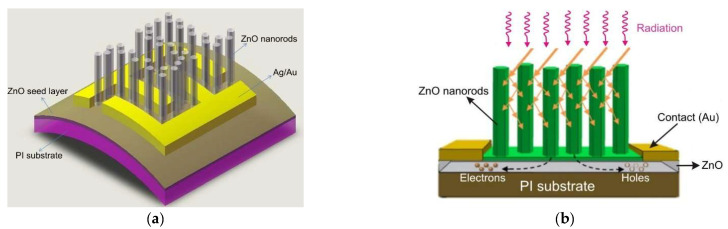
Nanostructure-based flexible ZnO photodetector: (**a**) MSM nanorod photodetector, and (**b**) enhanced light trapping by nanorods.

**Figure 65 sensors-23-04452-f065:**
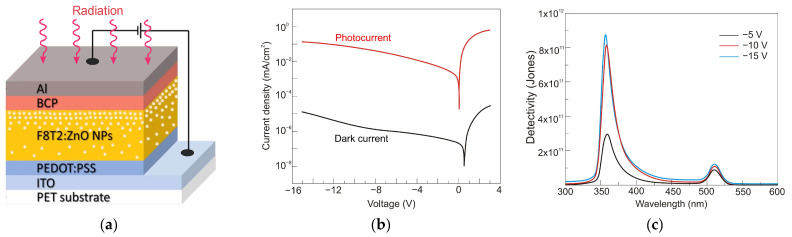
Flexible narrowband UV photodetector: (**a**) the device structure; (**b**) *J-V* curves in the dark and under 360 nm light illumination at bias voltages between −15 and 3 V; (**c**) detectivities under reverse biases of −5, −10, and −15 V. Reprinted with permission from Ref. [191].

**Table 1 sensors-23-04452-t001:** Photon detectors.

Type of Detector	Detector Structure	Diagram of Energy Band
Photoemissive detector The photoelectric effect involves the emission of electrons when optical radiation hits a photocathode with sufficiently high kinetic anergy, greater than the vacuum level barrier, to leave the photocathode and be emitted as a free electron. Suppose a large electric field is placed between the cathode and the anode. In that case, the emitted electrons are accelerated in the space between the detector electrodes, and the collecting anode produces a photocurrent proportional to the intensity of the incoming photon. In the photomultiplier tube (PMT), photoemitted electrons impact other dynodes specially placed in the tube, creating a cascade of emitted electrons by a secondary emission process.	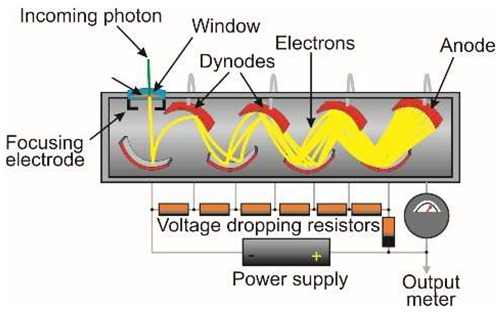	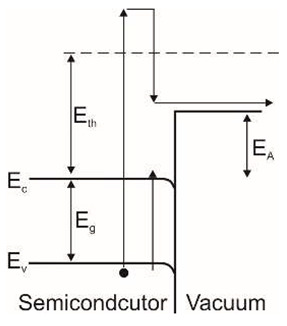
Photoconductor It is an optical radiation-sensitive photoresistor in which incident radiation creates electron-hole pairs in a homogeneous semiconductor material directly across the band gap. This band gap determines the spectral response. During the same phenomenon, a quantum-well photoconductor photoexcites electrons or holes from the potential well in the band-gap regions of the semiconductor.	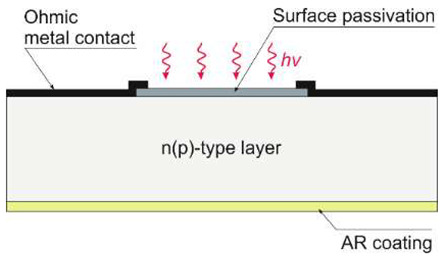	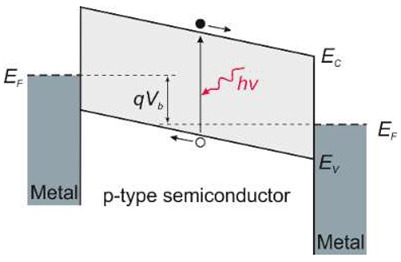
P–N junction photodiode This is a widely used photodetector in a typical p-on-n configuration with a shallowly diffused p-region on the n-type active layer. An n-on-p structure is also available. An electric field separates photo-created electron-hole pairs on either side of the junction in the space charge region. The generated photocurrent changes the open-circuit junction voltage or the short-circuit junction current.	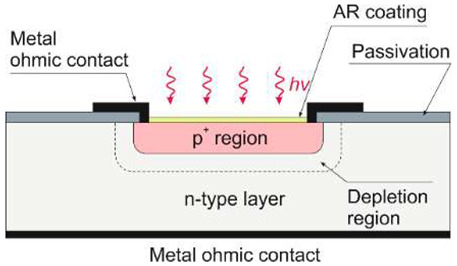	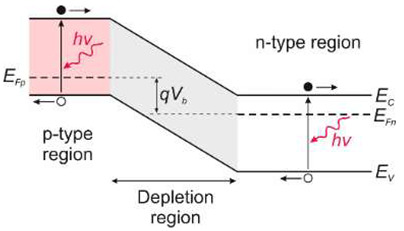
P-i-N photodiode P-i-N is merely a standard photodiode in which an intrinsic i-region is incorporated between the P and N sides of the junction (a capital letter stands for a wider energy gap). The depletion region occupies the entire intrinsic volume of the reverse bias voltage. Only in this region do incident photons generate electron-hole pairs. If there is no electrically neutral volume, zero diffusion current in the device is observed, and minority carriers generated through defect centers in the diode depletion region compose the dark current.	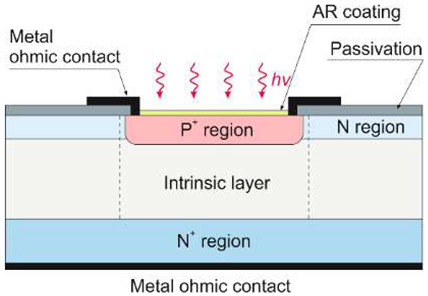	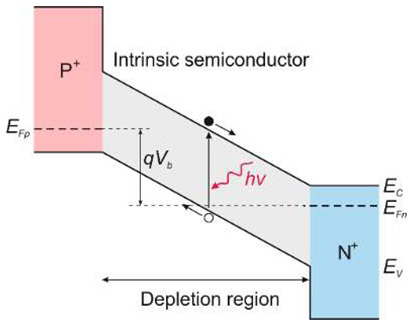
Avalanche photodiode (APD) The avalanche photodiode detects extremely low-intensity radiation. The n^+^-n^−^-p architecture, with controlled doping profiles and a geometry that ensures a uniform electric field, is easy to produce. The high reverse bias generates the depletion layer across the n^−^region, and photons are absorbed in the p-region. The applied electric field accelerates photogenerated minority carrier electrons at the edge of the depletion region. Their collisions with the crystal lattice cause an avalanche of ionization. This process provides new carriers and causes a rapid increase in current flowing in the reverse direction. The APD offers a combination of high speed, high sensitivity, and high quantum efficiency.	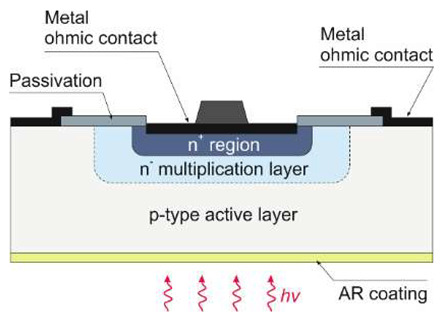	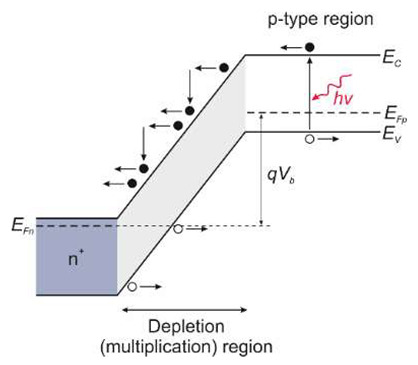
Schottky barrier photodiode It is a majority carrier device formed at the metal-semiconductor junction. As with the p–n junction, this junction provides a potential barrier to separate photoexcited electron-hole pairs within the semiconductor or at the metal-semiconductor interface. Compared to a p-n photodiode, a Schottky barrier photodiode has some advantages, e.g., fabrication simplicity (deposition of the metal barrier on the n(p)-semiconductor), lower-temperature diffusion processes, and a higher reaction rate. However, a Schottky junction is generally characterized by a more significant dark current.	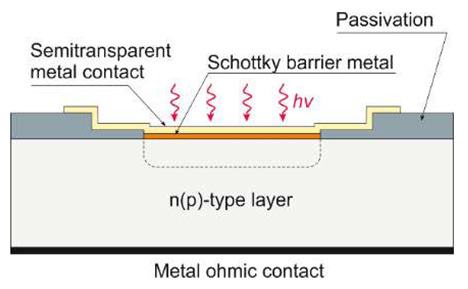	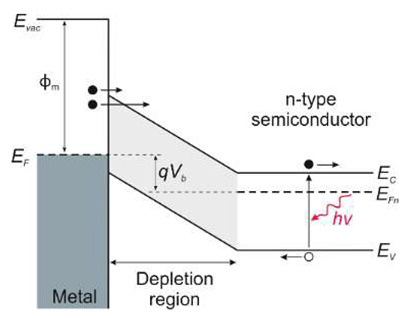
Metal-semiconductor-metal (MSM) photodiode This structure is similar to the interdigitated photoconductor, in which Schottky barriers fabricate the metal-semiconductor and semiconductor-metal junctions instead of ohmic contacts. Processing steps nearly identical to those required for making field-effect transistors are used to produce a planar structure with monolithic integration of the MSM photodiode. A lower dark current characterizes this type of design compared to a single Schottky diode and has a faster response speed than a p-i-n photodiode.	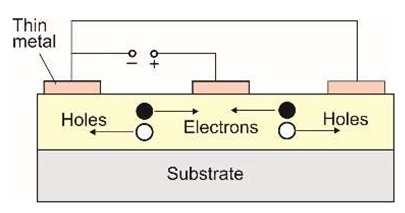	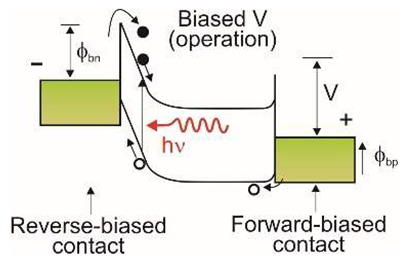
Metal-insulator-semiconductor (MIS) photodiode This device is built with an insulator separating the metal gate from the semiconductor surface. The thickness of the insulator must not fall below a minimum of about 10 nm. The use of thinner insulation than this leads to the occurrence of tunneling through the insulating layer. A negative voltage relative to the metal electrode creates a depletion region, repelling electrons from the insulator-semiconductor interface. Its “well capacitance” defines the total charge accumulated on a photogate. This capacitance depends on the gate bias, insulator thickness, electrode surface area, and semiconductor background doping.	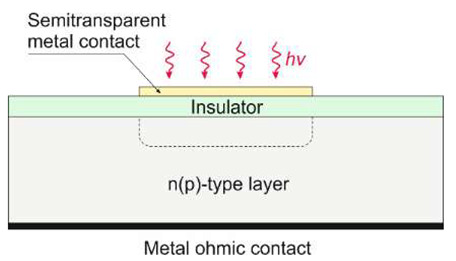	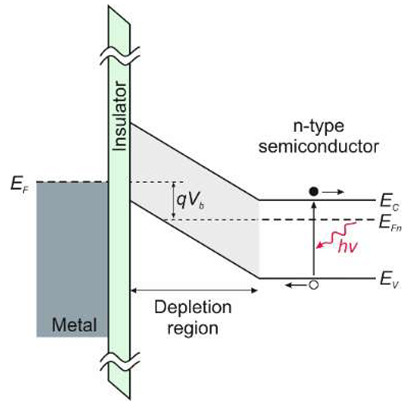
Photo-FET The device’s configuration is similar to lateral photoconductors, with the MSM architecture forming the source and drain electrodes. The modulation of the channel conductivity is performed using a third gate electrode that is electrically isolated from the semiconductor channel by a thin dielectric layer. The *V_G_* gate voltage electronically controls the density of the carrier by modulating the field effect and advantageously switches off the dark current by operating the device in the depletion regime. The incident light activates the conductivity of the device channel by photo-generating the carriers, which then ideally benefit from the photoconductivity enhancement mechanism (photoelectric gain), as in photoconductors.	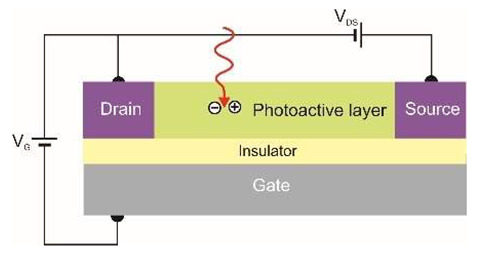	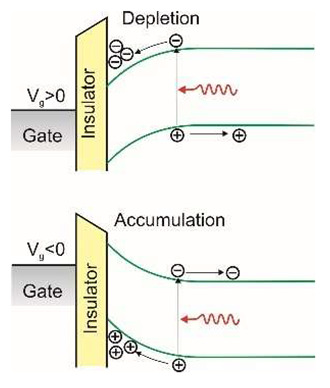

**Table 2 sensors-23-04452-t002:** Physical properties of semiconductors used in the manufacture of UV photodetectors.

Parameters	Si	Diamond	4H-SiC	6H-SiC	GaN	AlN	ZnO	MgO	ZnSe	β-Ga_2_O_3_
Energy gap (eV)	1.12	5.45	3.26	3.02	3.4	6.2	3.37	7.83	2.58	4.85
Density (g/cm^3^)	2.329	3.52	3.21	3.21	6.15	3.32	5.61	3.58	5.42	5.95
Thermal conductivity (W/cmK)	1.45	22.9	3.7	4.9	3.2	4.6	5.4	4.82	0.18	0.20
Melting point (K)	1683	3773	3100	3100	>2000	>2500	2242	3073	1517	1795
Electron saturation velocity (10^5^ m/s)	1	2.3	2.1	2	1.4	1.3				1.1
Electron mobility (cm^2^/Vs)	1240	7300	950	400	1000	420	170	10	540	300
Hole mobility (cm^2^/Vs)	480	5300	120	75	11	14	40	2	30	20
Dielectric constant	11.9	5.7	9.7	9.7	10.4	10.1	9.1	9.8	8.6	10
Breakdown field (10^5^ V/cm)	3	130	31	24	49.5	154				103

**Table 3 sensors-23-04452-t003:** Performance of different AlGaN UV photodetectors.

Structure	Dark Current/(Density)	*I_ph_*/*I_dark_*	Responsivity [A/W]	UV/VIS Rejection Ratio	*t_rise_*/*t_fall_*	Ref.
GaN PIN	8 × 10^−10^ A/cm^2^	8.4 × 10^5^	0.35	1.2 × 10^4^	75 µs/110 µs	[43]
GaN MSM	1 pA	1 × 10^5^	3.096	5 × 10^4^	-	[44]
AlGaN MSM	20 pA	2.6 × 10^5^	-	-	25 ps	[45]
p-GaN/AlGaN/GaN on Si (*λ_c_* = 395 nm)	29 × 10^−9^ mA/mm^2^	1.7 × 10^8^	2 × 10^4^	1.1 × 10^7^	12.2 ms/8.9 ms	[46]
GaN PIN PC mode	40 pA/cm^2^ (<2 fA)	-	0.14	8 × 10^3^	-	[47]
AlGaN/GaN PD (*λ_c_* = 365 nm)	50 nA	1 × 10^5^	7 × 10^4^	10^4^	6 ms *	[48]
AlGaN Schottky (*λ_c_* = 310 nm)	20 pA@1 V	105	0.07	-	10 ns/190 ns	[49]

* Response time.

**Table 4 sensors-23-04452-t004:** Performance of different GaN and AlGaN avalanche photodiodes.

Structure	Substrate	Front/Back Illuminated	Electron/Hole Initiated	Breakdown Voltage(V)	*I_dark_* (A/cm^2^ at 95% BV)	Maximum Responsivity (A/W)	Maximum Gain	Ref.
GaN	GaN	Front	Hole	278	1.5 × 10^−5^	60	10^5^	[62]
GaN	Sapphire	Back	Hole	90	0.044	0.163	1.9 × 10^4^	[64]
AlGaN (*λ_c_* = 340 nm)	Sapphire	Back	Hole	66.5	0.01	0.15	2 × 10^4^	[65]
AlGaN (*λ_p_* = 255 nm)	Sapphire	Back	Hole	90	3.5 × 10^−3^	2	3 × 10^3^	[66]
AlGaN (*λ_p_* = 255 nm)	Sapphire	Back	Hole	80	0.8	NA	5 × 10^4^	[67]
Al_0.05_Ga_0.95_N	GaN	Front	Electron	100	1 × 10^−4^	0.8	8 × 10^5^	[68]
GaN	GaN	Front	Electron	92.3	2 × 10^−4^	0.15	1.4 × 10^4^	[69]
GaN	GaN	Front	Electron	95.4	1 × 10^−2^	0.14	2 × 10^6^	[70]
Al_0.05_Ga_0.95_N	Sapphire	Front	Electron	90	4.4 × 10^−5^	NA	2 × 10^4^	[68]
GaN	Sapphire	Front	Electron	>120	1.27 × 10^−4^	0.23	NA	[71]
GaN	Sapphire	Front	Electron	92	795	NA	300	[72]
GaN	SiC	Front	Electron	160	2 × 10^−8^	4.2	10^5^	[73]

**Table 5 sensors-23-04452-t005:** Summary of reported AlGaN-based focal plane arrays.

Material of Active Region	Pixel Structure	Array Format	Pixel Size (µm^2^)	Wavelength Range (nm)	Year	Ref.
AlGaN	p-i-n	256 × 256	30 × 30	265–285	2001	[80]
GaN	p-i-n	320 × 256	24 × 24	300–365	2002	[81]
Al_0.32_Ga_0.68_N	p-i-n	320 × 256	25 × 25	280	2005	[82]
AlGaN	p-i-n	256 × 256	-	285–365	2005	[83]
Al_0.45_Ga_0.55_N	p-i-n	256 × 256	30 × 30	260–280	2006	[84]
Al_0.45_Ga_0.55_N	Schottky	320 × 256	30 × 30	280	2007	[85]
Al_0.59_Ga_0.41_N	p-i-n	128 × 128	50 × 50	233–258	2008	[86]
Al_0.43_Ga_0.57_N	Schottky	320 × 256	30 × 30	260–290	2009	[87]
Al_0.4_Ga_0.6_N with Si-layer	Schottky	256 × 256	10 × 10	1–33	2011	[88]
Al_0.45_Ga_0.55_N	p-i-n	320 × 256	25 × 25	290	2013	[89]
Al_0.4_Ga_0.6_N	p-i-n	320 × 256	25 × 25	278	2015	[90]
Al_0.43_Ga_0.57_N	p-i-n	640 × 512	15 × 15	280	2020	[91]

**Table 6 sensors-23-04452-t006:** Comparison of the major parameters for Schottky and p-i-n SiC photodetectors.

Device	Active Region	Responsivity Peak (A/W)	Reverse Voltage (V)	Wavelength at Peak (nm)	Quantum Efficiency (%)	Dark Current/ (Density)	Ref.
Schottky	6H-SiC	0.070	0	320	28	0.2 nA/cm^2^	[106]
Schottky	4H-SiC	0.093	−15	270	87.9	<70 fA	[107]
Schottky	4H-SiC	0.073	−10	290	31.2	<1 pA @, 0 V	[108]
Schottky	4H-SiC	0.115	0	285	50	1 pA (20 V, 100 °C)	[109]
p-i-n	4H-SiC	0.15	−5	268	-	1.8 pA @ RT	[105]
p-i-n	4H-SiC	0.13	−5	270	61	2.5 pA/mm^2^	[110]
p-i-n	4H-SiC	0.204	−30	285	88.3	38.6 nA/cm^2^@-10 V	[111]

**Table 7 sensors-23-04452-t007:** Comparison of the main parameters of APDs based on 4H-SiC structures.

*λ_peak_* (nm)	Responsivity Peak (A/W)	V (V)	Gain (*M*)	*k*	QE (%)	Dark Current */(Density)	Ref.
280	0.093	144	1.4 × 10^4^	0.1	41	63 nA/cm^2^ at *M* = 1	[113]
290	0.080	112 (M = 100)	10^3^	0.1	35	9.2 µA/cm^2^ at *M*	[112]
274	0.18	~156	10^6^	-	81.5	0.2 nA/cm^2^	[117]
270	0.203	42	1.8 × 10^4^ @55 V	-	93	<10 pA@27.5 V	[115]
270	0.100	30	10^5^@94 V	-	45	10 nA@30 V	[116]

* Dark current is equal to primary dark current if *M* = 1.

**Table 8 sensors-23-04452-t008:** Comparison of significant parameters based on diamond detectors.

Material /Device *	Responsivity (A/W)@220 nm	Dark Current (A)	Response Time	Rejection Ratio	Ref.
PC/BSCD PC/SCD PC/Polylicryst.	6 at 3 V 21.8 at 50 V 0.63 at 222 nm	10^−12^ @20 V - 0.3 × 10^−9^	<10 ns Rise 310 µs/Decay 330 µs -	10^8^ 8.9 × 10^3^ 10^6^	[127] [129] [137]
MSM MSM	230 at 1 V 0.02 at 6 V	10^−12^@0.4 V Few pA	- -	10^6^ 10^4^	[133] [138]
Schottky BSCD Polycrystalline	18 at −23 V 10 at 100 V	10^−14^@−23 V -	1–10 ms Rise 5 ms/Decay 8 ms	10^8^ 10^4^–10^5^	[135] [139]
PD/pin	27 mA/W@200 nm	4 × 10^−13^	-	10^6^	[136]

* where: PD—photodiode; PC—photoconductive detector; SCD—single-crystal diamond; BSCD—boron-doped SCD.

**Table 9 sensors-23-04452-t009:** Material properties of Ga_2_O_3_ in comparison with major semiconductors.

Parameter	Ga_2_O_3_	Si	4H-SiC	GaN	AlN	Diamond
Energy gap (eV)	4.5–5.3	1.1	3.3	3.4	6.2	5.5
Breakdown field (MV/cm)	5–9	0.3	2.6	3.3	2	10
Electronic mobility (cm^2^/Vs)	300	1400	1000	1200	155	2000
Thermal conductivity (W/cmK)	0.23 (010) 0.13 (100)	1.5	2.7	2.1	13	22
Baliga’s FOM ($\varepsilon \mu E_{b}^{3})$	3443	1	340	870	-	24,664
Johnson’s FOM ${\left(E_{b}V_{s}/\pi \right)}^{2}$	1093	1	177.6	756	-	-
Substrate size (inch)	6	12	8	2	2	1
Response spectrum (nm)	<280	400–1100	200–370	200–300	<200	<225

**Table 10 sensors-23-04452-t010:** Performance parameters of Ga_2_O_3_-based solar-blind UV photodetectors.

Structure	*λ* [µm]	*R* (A/W)	EOE (%)	Dark Current (A)	Rejection Ratio	*D** (Jones)	Response Time (*t_r_*/*t_d_*)
β-Ga_2_O_3_/spiro-MeOTAD	254	6.5 × 10^−2^ (0 V)	32	7.5 × 10^−14^	3.6 × 10^3^	3.95 × 10^11^	0.21 s/0.07 s
β-Ga_2_O_3_/MoS_2_	245	2.05 × 10^−3^ (0 V)	-	0.9 × 10^−12^	1.6 × 10^3^	1.21 × 10^11^	-
Ga_2_O_3_/graphene	200	12.8 (−6 V)	-	1.2 × 10^−8^	-	1.3 × 10^13^	0.0015 s/0.002 s
Ga_2_O_3_/polyaniline	246	2.1 × 10^−2^ (0 V)	-	0.8 × 10^−13^	10^2^	1.5 × 10^11^	0.00034 s/0.00814 s
amorphous-Ga_2_O_3_	250	607.11	3.0 × 10^5^	-	-	5.15 × 10^14^	0.65 s/1.18 s
Ga_2_O_3_ nanobelts	250	851	4.2 × 10^3^	1.0 × 10^−13^	-	-	<0.3 s/<0.3 s
ZnO-Ga_2_O_3_ core shell	254	1.3 × 10^3^ (−6 V)	2.5 × 10^6^	5.35 × 10^−10^	5 × 10^3^	9.91 × 10^14^	2 × 10^−5^ s/4.2 × 10^−5^ s
Au/β-Ga_2_O_3_/ZnO	254	1.1 × 10^4^ (−40 V)	-	>1.0 × 10^−12^	10^3^	9.6 × 10^12^	<0.000563 s/0.000238 s
β-Ga_2_O_3_	254	4.24 × 10^4^	2 × 10^5^	-	-	1.7 × 10^11^	-
β-Ga_2_O_3_ micro flakes	254	1.7 × 10^5^	8.3 × 10^7^	27 × 10^−15^	2.4 × 10^4^	1.1 × 10^18^	0.428 s/0.435 s
β-Ga_2_O_3_ phototransistor	254	1.4 × 10^7^	6.4 × 10^7^	-	6.0 × 10^7^	1.1 × 10^19^	-/0.016 s
β-Ga_2_O_3_/MgO flakes	260	2.4 × 10^7^	-	6.7 × 10^−12^	-	1.7 × 10^15^	6.38 s/0.07 s

**Table 11 sensors-23-04452-t011:** The key performance parameters of selected II-VI semiconductor UV photodetectors.

Material/Structure	*λ_c_* (nm)	*R_i_* (A/W)	Dark Current (pA)	Rejection Ratio	Response Time (µs); *t_r_*/*t_d_*
w-ZnMgO/MSM	310	34.02	7.67 × 10^3^ (5 V)	2660	44/1.2 × 10^5^
w-ZnMgO/MSM	350	2.1 (10 V)	0.73 (5 V)	5.46 × 10^5^	2 × 10^8^/6 × 10^7^
w-Zn_0.6_Mg_0.4_O/p-Si heterojunction	300	0.22 (−2 V)	700 (−3 V)	10^2^	1 × 10^6^
w-Zn_0.5_Mg_0.5_O/p-Si heterojunction	208 301	0.013 (1 V) 0.006 (−2 V)	2 × 10^3^	10^2^	10^5^/10^5^
m-Zn_0.38_Mg_0.62_O/MSM	275	1.66 (10 V)	8.29 (40 V)	10^3^	15/-
m-Zn_0.59_Mg_0.41_O/MSM	320	89.8 (40 V)	11 (40 V)	10^5^	-/3 × 10^4^

**Table 12 sensors-23-04452-t012:** New concepts of ultraviolet photodetector design—selected parameters.

Materials	Wavelength (nm)	Responsivity (A/W)	Response Time (Rise/Delay)	Detectivity (Jones)
2D materials				
Few-layer BP	310–390	9 × 10^4^	1 ms/4 ms	3 × 10^13^
WS_2_	365	53.3	-	1.2 × 10^11^
GaS	254	19.2	-	1 × 10^14^
BiOBr	340	15.0	80 ms/40 ms	5.7 × 10^10^
CuGaS_2_ nanosheets	254	5.1	1.8 s/10.1 s	1.7 × 10^11^
Graphene quantum dots	254	2.1 × 10^−3^	64 ms/43 ms	9.6 × 10^11^
2D heterostructure and hybrid				
MoS_2_/Cs_3_Bi_2_I_9_	325	1.42	11.6 ms/14.9 ms	1.2 × 10^13^
Graphene/GaN	300–360	0.23	2.5 ms/2.5 ms	
BP/MoS_2_	365	77.2	-	6.5 × 10^9^
Graphene/poly-BrNpA	365	149	-	1.6 × 10^11^
Self-powered				
p-GaSe/n-WS_2_	270–740	0.149	37 ms/43 ms	4.3 × 10^12^
SnS_2_/ZnO_1−x_S_x_	365	8.3 × 10^−3^	49.5 ms/25.9 ms	5.1 × 10^10^
Bi_2_Te_3_ nanoshets	365	6.3 × 10^−3^	12 ms/32 ms	-
Graphene/Si	365	0.12	5 ns	4.3 × 10^12^
Flexible				
GaS	254	19.2	30 ms	1 × 10^14^
MAPbI_3_	264	1.2	2.2 ms/4 ms	1 × 10^11^
ITO/ZnO/PI	360	0.0125	3.24 s/108.7 ms	2.71 × 10^11^
ZnS/MoS_2_	365	9.5 × 10^−6^	22 ms	-
Ca_2_Nb_3_O_10_	280	14.9	0.08 ms/5.6 ms	8.7 × 10^13^

**Table 13 sensors-23-04452-t013:** Advantages and disadvantages of ultraviolet photodetectors.

Type of Photodetector	Advantages	Disadvantages
Photoconductive detector	Simple design, easy process of technological control, high photoelectrical gain	Large dark current, slow response time, generally lower detectivity in comparison with p-n photodiode
p-n and p-i-n photodiodes	High impedance, low dark current, fast response time, high-frequency operation, low-bias or self-powered operation	Spectral characteristic depends on the doping level of the active region, higher doping level worsens the spectral sensitivity
Phototransistor	Very high responsivity, very large photogating effect	Complicated technology, slow response time
Schottky barrier	Simple fabrication process, good responsivity, short response time,	Absorption losses, saturation dark current considerably higher than in p–n junction (due to lower built-in voltage)
Metal-semiconductor-metal (MSM)	Simple fabrication process, fast response time, lower dark current compared to a single Schottky diode, easy integration	Lower spectral response and quantum efficiency

## Data Availability

The study did not report any data.

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
