# Peer review of "Ultraviolet Photodetectors: From Photocathodes to Low-Dimensional Solids"

_sensors, 2023, doi:10.3390/s23094452_

Round 1

Reviewer 1 Report

The manuscript presents the long term evolution and recent development of ultraviolet photodetectors. Here are some comments and suggestions. The authors completed described and revealed various type photodetector in the manuscript, including the principle of the photodetector, the properties of responsivity, NEP, detectivity. It could be helped researcher realized the field of the ultraviolet photodetectors. To further help researcher know the properties in the low frequency, it is recommended to add the characteristics of the low frequency noise in the manuscript.

Author Response

Responses to the Reviewer Comments

Manuscript ID: sensors-2319559

We appreciate very much the Reviewer for the comments and time put into the review of the manuscript. Each comment has been carefully considered and responded. In the revised manuscript, we tried to make changes in accordance with the expectations of Reviewer, while presenting our point of view. Changes in the revised manuscript are marked in red. 

Report Form #1

Comments and Suggestions for Authors

The manuscript presents the long term evolution and recent development of ultraviolet photodetectors. Here are some comments and suggestions. The authors completed described and revealed various type photodetector in the manuscript, including the principle of the photodetector, the properties of responsivity, NEP, detectivity. It could be helped researcher realized the field of the ultraviolet photodetectors. To further help researcher know the properties in the low frequency, it is recommended to add the characteristics of the low frequency noise in the manuscript.

Answer. We think, according to the reviewer's suggestion, that it would have been good to include more considerations of 1/f noise. We had already discussed this during the writing of the paper. However, we abandoned it due to the disparate experimental data reported in the literature. Especially in devices with a wide energy gap (and such are used in ultraviolet detectors), the 1/f noise level is affected by the quality of the material in the active area of the detector and the procedure for making contacts. The data in Table 10 and Figure 53 are a particular example of this.

We believe we have addressed all the Reviewer remarks and we have done our best for the manuscript to meet the requirements for publication in Sensors. We also believe that all the changes/corrections introduced in the revised version of the manuscript improved its scientific value.

Yours sincerely,

Authors

Reviewer 2 Report

The authors present the long term evolution and recent development of ultraviolet photodetectors. The paper covers all available technologies in this field and the lead author (Dr Rogalski) is a prominent figure in this field. I believe this paper is excellent in its current form and should be published as it is. This is a valuable reference for researchers in this field.

Author Response

Responses to the Reviewer Comments

Manuscript ID: sensors-2319559

We appreciate very much the Reviewer for the comments and time put into the review of the manuscript. In the revised manuscript, we tried to make changes in accordance with the expectations of Reviewers, while presenting our point of view. Changes in the revised manuscript are marked in red. 

Report Form #2

Comments and Suggestions for Authors

The authors present the long term evolution and recent development of ultraviolet photodetectors. The paper covers all available technologies in this field and the lead author (Dr Rogalski) is a prominent figure in this field. I believe this paper is excellent in its current form and should be published as it is. This is a valuable reference for researchers in this field.

Answer. Thank you very much for the evaluation of our manuscript, high rating and positive recommendation.

We believe that all the changes/corrections introduced in the revised version of the manuscript improved its scientific value.

Yours sincerely,

Authors

Reviewer 3 Report

The review article on UV detectors is well preparred and written. Could be a compendium of knowledge about detectors. I recommend to publish it, after minor revisions:

1. Fig 1, 5, 8, 10, 11... Are these your figures? If yes, probably it was prepared on the basis literature datas. Could you provide references?

2. Chapters are named: 4. Silicon based detectors; 5. Wide bandgap... It will be easier to understand classification, when in Fig 3 will be the same names of detectors groups.

3. I advise to add in conclusions table or plot with collected advantages and disadvantages and parameters of dectors.

Author Response

Responses to the Reviewer Comments

Manuscript ID: sensors-2319559

We appreciate very much the Reviewer for the comments and time put into the review of the manuscript. Each comment has been carefully considered and responded. In the revised manuscript, we tried to make changes in accordance with the expectations of Reviewer, while presenting our point of view. Changes in the revised manuscript are marked in red. 

Report Form #3

Comments and Suggestions for Authors

1. Fig 1, 5, 8, 10, 11... Are these your figures? If yes, probably it was prepared on the basis literature datas. Could you provide references?

Answer. Most of the drawings are developed by the authors. Some of them are modified (compared to the known original), but then this fact is indicated in the text. Figures taken from the papers of other authors are clearly marked and we have permissions to include them in our paper.

2. Chapters are named: 4. Silicon based detectors; 5. Wide bandgap... It will be easier to understand classification, when in Fig 3 will be the same names of detectors groups.

Answer. To take into account the reviewer's suggestion, we have modified Fig. 3.

3. I advise to add in conclusions table or plot with collected advantages and disadvantages and parameters of detectors.

Answer. In accordance with the reviewer’s suggestion, we made the additions highlighted in red in the Conclusion (as below).

Currently, among the various types of UV detectors, photovoltaic detectors are dominant. Table 13 compares their advantages and disadvantages. Among the various materials used in UV detectors, Si-based photodiodes have a superior rank regarding simplicity, high sensitivity, low costs, and integrated circuit compatibility. However, when operating in high-temperature environments, such photodetectors must often be cooled to reduce their leakage current and, consequently, the noise of the detection system. Additionally, an optical filter is usually needed to filter out visible and infrared light, which significantly increases the complexity and cost of photodetectors. To overcome these problems, wide bandgap semiconductors such as AlGaN, SiC, diamond-based, and Ga2O3 have been investigated for UV light detection. Compared to Si-based devices, the application of these materials has many advantages, such as high breakthrough field, filter-less UV light detection, high radiation resistance and high chemical and thermal stability.

Table 13. Advantages and disadvantages of ultraviolet photodetectors

Type of photodetector

Advantages

Disadvantages

Photoconductive detector

Simple design, easy process of technological control, high photoelectrical gain

Large dark current, slow response time, generally lower detectivity in comparison with p-n photodiode

p-n and p-i-n photodiodes

High impedance, low dark current, fast response time, high-frequency operation, low-bias or self-powered operation

Spectral characteristic depends on the doping level of the active region, higher doping level worsens the spectral sensitivity

Phototransistor

Very high responsivity, large photogaiting effect

Complicated technology, slow response time

Schottky barrier

Simple fabrication process, good responsivity, short response time,

Absorption losses, saturation dark current considerably higher than in p-n junction (due to lower built-in voltage)

Metal-semiconductor-metal (MSM)

Simple fabrication process, fast response time, lower dark current compared to a single Schottky diode, easy integration

Lower spectral response and quantum efficiency

In comparison with standard semiconductors, the random stacking and fabrication of 2D material devices is a tremendous advantage due to absence of lattice matching. A weak stack of atomic planes stacked on top of each other, held together by van der Waals (vdW) forces, leaves no dangling bonds, facilitating the fabrication of vertical heterostructures and the integration of 2D materials into silicon chips. In addition, vdW interactions allow large-scale growth of 2D materials regardless of substrates.

We believe we have addressed all the Reviewer remarks and we have done our best for the manuscript to meet the requirements for publication in Sensors. We also believe that all the changes/corrections introduced in the revised version of the manuscript improved its scientific value.

Yours sincerely,

Authors